# Trade-Offs of Diagonal Fisher Information Matrix Estimators

**Alexander Soen**
The Australian National University
RIKEN AIP
alexander.soen@anu.edu.au

**Ke Sun**
CSIRO's Data61
The Australian National University
Ke.Sun@data61.csiro.au

## Abstract

The Fisher information matrix can be used to characterize the local geometry of the parameter space of neural networks. It elucidates insightful theories and useful tools to understand and optimize neural networks. Given its high computational cost, practitioners often use random estimators and evaluate only the diagonal entries. We examine two popular estimators whose accuracy and sample complexity depend on their associated variances. We derive bounds of the variances and instantiate them in neural networks for regression and classification. We navigate trade-offs for both estimators based on analytical and numerical studies. We find that the variance quantities depend on the non-linearity w.r.t. different parameter groups and should not be neglected when estimating the Fisher information.

## 1 Settings

In the parameter space of neural networks (NNs), *i.e.* the *neuromanifold* [1], the network weights and biases play the role of a coordinate system and the local metric tensor can be described by the Fisher Information Matrix (FIM). As a result, empirical estimation of the FIM helps reveal the geometry of the loss landscape and the intrinsic structure of the neuromanifold. Utilizing these insights has lead to efficient optimization algorithms, *e.g.*, the natural gradient [1] and Adam [16].

A NN with inputs $\boldsymbol{x}$ and stochastic outputs $\boldsymbol{y}$ can be specified by a conditional p.d.f. $p(\boldsymbol{y} \mid \boldsymbol{x}; \boldsymbol{\theta})$, where $\boldsymbol{\theta}$ is the NN's weights and biases. This paper considers the general parametric form

$$p(\boldsymbol{y} \mid \boldsymbol{x}; \boldsymbol{\theta}) = \pi(\boldsymbol{y}) \cdot \exp\left(\boldsymbol{t}^\top(\boldsymbol{y})\boldsymbol{h_\theta}(\boldsymbol{x}) - F(\boldsymbol{h_\theta}(\boldsymbol{x}))\right), \tag{1}$$

where $\boldsymbol{h_\theta} \colon \Re^I \to \Re^T$ maps $I$-dimensional inputs $\boldsymbol{x}$ to $T$-dimensional exponential family parameters, $\boldsymbol{t}(\boldsymbol{y})$ is a vector of sufficient statistics, $\pi(\boldsymbol{y})$ is a base measure, and $F(\cdot)$ is the log-partition function (normalizing the exponential). For example, if $\boldsymbol{y}$ denotes class labels and $\boldsymbol{t}(\boldsymbol{y})$ maps to its corresponding one-hot vectors, then Eq. (1) is associated with a multi-class classification network.

Assuming that the marginal distribution $q(\boldsymbol{x})$ is parameter-free, we define parametric joint distributions $p(\boldsymbol{x}, \boldsymbol{y}; \boldsymbol{\theta}) = q(\boldsymbol{x})p(\boldsymbol{y} \mid \boldsymbol{x}; \boldsymbol{\theta})$. The (joint) FIM is defined as $\mathcal{I}(\boldsymbol{\theta}) \doteq \mathbb{E}_{q(\boldsymbol{x})}\left[\mathcal{I}(\boldsymbol{\theta} \mid \boldsymbol{x})\right]$, where

$$\mathcal{I}(\boldsymbol{\theta} \mid \boldsymbol{x}) \doteq \mathop{\mathbb{E}}_{p(\boldsymbol{y} \mid \boldsymbol{x}; \boldsymbol{\theta})}\left[\frac{\partial \log p(\boldsymbol{y} \mid \boldsymbol{x}; \boldsymbol{\theta})}{\partial \boldsymbol{\theta}}\frac{\partial \log p(\boldsymbol{y} \mid \boldsymbol{x}; \boldsymbol{\theta})}{\partial \boldsymbol{\theta}^\top}\right] \overset{(*)}{=} - \mathop{\mathbb{E}}_{p(\boldsymbol{y} \mid \boldsymbol{x}; \boldsymbol{\theta})}\left[\frac{\partial^2 \log p(\boldsymbol{y} \mid \boldsymbol{x}; \boldsymbol{\theta})}{\partial \boldsymbol{\theta}\partial \boldsymbol{\theta}^\top}\right] \tag{2}$$

is the 'conditional FIM'. The second equality (*) holds if $\boldsymbol{h_\theta}$'s activation functions are in $C^2(\Re)$ (*i.e.*, $\boldsymbol{h_\theta}$ is a sufficiently smooth NN). $\mathcal{I}(\boldsymbol{\theta} \mid \boldsymbol{x})$ does *not* have this equivalent expression (*) for NNs with ReLU activation functions [37]. Both $\mathcal{I}(\boldsymbol{\theta})$ and $\mathcal{I}(\boldsymbol{\theta} \mid \boldsymbol{x})$ define $\dim(\boldsymbol{\theta}) \times \dim(\boldsymbol{\theta})$ positive semi-definite (PSD) matrices. The distinction in notation is to emphasize that the joint FIM $\mathcal{I}(\boldsymbol{\theta})$ (depending only on $\boldsymbol{\theta}$) is simply the average over individual conditional FIMs $\mathcal{I}(\boldsymbol{\theta} \mid \boldsymbol{x})$ (depending on both $\boldsymbol{\theta}$ and $\boldsymbol{x}$).

38th Conference on Neural Information Processing Systems (NeurIPS 2024).

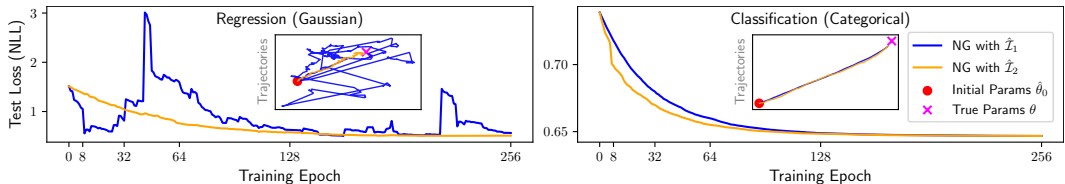

Figure 1: Natural gradient (NG) descent using $\hat{\mathcal{I}}_1(\boldsymbol{\theta})$ / $\hat{\mathcal{I}}_2(\boldsymbol{\theta})$ on a 2D toy dataset for regression (linear regression) and classification (logistic regression) (details in Appendix A). Inset plot shows the parameter updates throughout training. Here, the variance of $\hat{\mathcal{I}}_2(\boldsymbol{\theta})$ is generally lower than $\hat{\mathcal{I}}_1(\boldsymbol{\theta})$.

In practice, the FIM is typically computationally expensive and needs to be estimated. Given $q(\boldsymbol{x})$ and a NN with weights and biases $\boldsymbol{\theta}$ parameterizing $p(\boldsymbol{y} \mid \boldsymbol{x}; \boldsymbol{\theta})$, as per Eq. (1), we consider two commonly used estimators of the FIM [11, 37] given by

$$\hat{\mathcal{I}}_1(\boldsymbol{\theta}) \doteq \frac{1}{N} \sum_{k=1}^{N} \left[ \frac{\partial \log p(\boldsymbol{y}_k \mid \boldsymbol{x}_k)}{\partial \boldsymbol{\theta}} \frac{\partial \log p(\boldsymbol{y}_k \mid \boldsymbol{x}_k)}{\partial \boldsymbol{\theta}^\top} \right]; \quad \text{and} \quad \hat{\mathcal{I}}_2(\boldsymbol{\theta}) \doteq \frac{1}{N} \sum_{k=1}^{N} \left[ -\frac{\partial^2 \log p(\boldsymbol{y}_k \mid \boldsymbol{x}_k)}{\partial \boldsymbol{\theta} \partial \boldsymbol{\theta}^\top} \right], \quad (3)$$

where $p(\boldsymbol{y}_k \mid \boldsymbol{x}_k) \doteq p(\boldsymbol{y}_k \mid \boldsymbol{x}_k; \boldsymbol{\theta})$ and $(\boldsymbol{x}_1, \boldsymbol{y}_1), \ldots, (\boldsymbol{x}_N, \boldsymbol{y}_N)$ are i.i.d. sampled from $p(\boldsymbol{x}, \boldsymbol{y}; \boldsymbol{\theta})$. A conditional variant of the estimators, denoted as $\hat{\mathcal{I}}_1(\boldsymbol{\theta} \mid \boldsymbol{x})$ and $\hat{\mathcal{I}}_2(\boldsymbol{\theta} \mid \boldsymbol{x})$, can be defined by fixing $\boldsymbol{x} = \boldsymbol{x}_1 = \cdots = \boldsymbol{x}_N$ and sampling $\boldsymbol{y}_1, \ldots, \boldsymbol{y}_N$ independently from $p(\boldsymbol{y} \mid \boldsymbol{x})$ in Eq. (3) — details omitted for brevity.

Both estimators, $\hat{\mathcal{I}}_1(\boldsymbol{\theta})$ and $\hat{\mathcal{I}}_2(\boldsymbol{\theta})$, are random matrices with the same shape as $\mathcal{I}(\boldsymbol{\theta})$. By Eq. (2), they are *unbiased* — for $\hat{\mathcal{I}}_2(\boldsymbol{\theta})$, this only holds if activations functions are in $C^2(\Re)$. Following Eq. (1)'s setting, the estimation variances of $\hat{\mathcal{I}}_1(\boldsymbol{\theta})$ and $\hat{\mathcal{I}}_2(\boldsymbol{\theta})$ can be expressed in closed form and upper bounded [37]. This provides an important, yet not widely discussed, tool for quantifying the estimators' accuracy [11] and hence insights for where / when different estimators should be used. Despite this, for deep NNs, neither these variances nor their bounds can be computed efficiently due to the huge dimensionality of $\boldsymbol{\theta}$.

This work focuses on estimating the *diagonal entries* of the FIM and their associated variances. Our results — including estimators of the FIM, their variances, and their variance bounds — can be implemented through automatic differentiation. These computational tools empower us to practically explore the trade-offs between the two estimators. For example, Fig. 1 shows natural gradient descent [1] for generalized linear models on a toy dataset, where $\hat{\mathcal{I}}_2(\boldsymbol{\theta})$ is preferable (especially for regression) and $\hat{\mathcal{I}}_1(\boldsymbol{\theta})$ suffers from high variance and an unstable learning curve. Our analytical results reveal how moments of the output exponential family and gradients of the NN in Eq. (1) affects the FIM estimators. We discover a general decomposition of the estimators' variances corresponding to the samples of $\boldsymbol{x}$ and $\boldsymbol{y}$. We investigate different scenarios where each FIM estimator is the preferred one and then connect our analysis to the empirical FIM.

## 2 Related Work

Prior efforts aim to analyze the structure of the FIM of NNs with random weights [34, 14, 15, 3, 31]. This body of work hinges on utilizing tools from random matrix theory and spectral analysis, characterizing the behavior and statistics of the FIM. One insight is that randomly weighted NNs have FIMs with a majority of eigenvalues close to zero; with the other eigenvalues taking large values [14, 15]. In our work, the randomness stems from sampling from data distributions $p(\boldsymbol{x}, \boldsymbol{y})$ — which follows the principle of Monte Carlo (MC) information geometry [29] that approximates information geometric quantities via MC estimation. We examine a different subject on how the distribution of the FIM on a matrix manifold is affected by finite sampling of the data distribution.

In the literature of NN optimization, a main focus is on deriving a computationally friendly proxy for the FIM. One can consider the *unit-wise* FIM [30, 20, 39, 3] (also known as quasi-diagonal FIM [30]), where a block-diagonal approximation of the FIM is taken to capture intra-neuron curvature information. Or one can consider the block-diagonal *layer-wise* FIM where each block corresponds to parameters within a layer [19, 27, 32, 26, 12, 35, 13]. NN optimizers can approximate the inverse FIM [36] or approximate the product of the inverse FIM and the gradient vector [35].

Much less attention is paid to how related approximations deviate from the true FIM [11, 37] or how optimization is affected by such deviation [41]. For the univariate case, one can study the asymptotic variance of the Fisher information [11] with the central limit theorem. In deep NNs, the estimation variance of the FIM can be derived in closed form and bounded [37]. However, our former analysis [37] has two limitations: (1) the variance tensors are 4D and can not be easily computed; (2) only the norm of these tensors are bounded, and it is not clear how the variance is distributed among individual parameters. The current work tackles these limitations by focusing on the diagonal elements of the FIM. Our results can be computed numerically at a reasonable cost in typical learning settings. We provide novel bounds so that one can quantify the accuracy of the FIM computation w.r.t. individual parameters or subgroup of parameters.

Issues of utilizing the empirical FIM to approximate the FIM have been highlighted [32, 25]. For example, estimators of the FIM do not in general capture any second-order information about the log-likelihood [18]. The empirical FIM is a biased estimator and can be connected with our unbiased estimators via a generalized definition of the Fisher matrix in Section 6.

Alternative to the FIM, the Generalized Gauss-Newton (GGN) matrix — a Hessian approximator — was originally motivated through the squared loss for non-linear models [25]. The GGN is equivalent to the FIM when a loss function is taken to be the empirical expectation of the negative log-likelihood of Eq. (1) [12, 32, 25].

## 3   Variance of Diagonal FIM Estimators

In our notations, all vectors such as $\boldsymbol{x}$, $\boldsymbol{y}$, and $\boldsymbol{\theta}$ are column vectors. We use $k$ to index random samples $\boldsymbol{x}$ and $\boldsymbol{y}$ and use $i$ and $j$ to index the NN weights and biases $\boldsymbol{\theta}$. We shorthand $\boldsymbol{h} \doteq \boldsymbol{h_\theta}$, $p(\boldsymbol{y} \,|\, \boldsymbol{x}) \doteq p(\boldsymbol{y} \,|\, \boldsymbol{x}; \boldsymbol{\theta})$, and $p(\boldsymbol{y}, \boldsymbol{x}) \doteq p(\boldsymbol{y}, \boldsymbol{x}; \boldsymbol{\theta})$ whenever the parameters $\boldsymbol{\theta}$ is clear from context. To be consistent, we use '$|\,\boldsymbol{x}$ conditioning' to distinguish between jointly calculated values versus conditioned values with fixed $\boldsymbol{x}$. By default, the derivatives are w.r.t. $\boldsymbol{\theta}$. For example, $\partial_i \boldsymbol{h} \doteq \partial \boldsymbol{h}/\partial\theta_i$ and $\partial_i^2 \boldsymbol{h} \doteq \partial^2 \boldsymbol{h}/\partial\theta_i^2$. We adopt Einstein notation to express tensor summations, so that an index appearing as both a subscript and a superscript in the same term indicates a summation. For example, $x^a y_a$ denotes $\sum_a x^a y_a$. For clarity, we mix standard $\Sigma$-sum and Einstein notation. We denote the variance and covariance of random variables by $\mathrm{Var}(\cdot)$ and $\mathrm{Cov}(\cdot)$, respectively.

Based on the parametric form of the model in Eq. (1), the diagonal entries of the FIM estimators in Eq. (3) can be written as[1]:

$$\hat{\mathcal{I}}_1(\theta_i) \doteq \left(\hat{\mathcal{I}}_1(\boldsymbol{\theta})\right)_{ii} = \frac{1}{N} \sum_{k=1}^{N} \left( \frac{\partial F(\boldsymbol{h}(\boldsymbol{x}_k))}{\partial\theta_i} - \frac{\partial \boldsymbol{h}^a(\boldsymbol{x}_k)}{\partial\theta_i} \cdot \boldsymbol{t}_a(\boldsymbol{y}_k) \right)^2 ;$$

$$\hat{\mathcal{I}}_2(\theta_i) \doteq \left(\hat{\mathcal{I}}_2(\boldsymbol{\theta})\right)_{ii} = \frac{1}{N} \sum_{k=1}^{N} \left( \frac{\partial^2 F(\boldsymbol{h}(\boldsymbol{x}_k))}{\partial^2\theta_i} - \frac{\partial^2 \boldsymbol{h}^a(\boldsymbol{x}_k)}{\partial^2\theta_i} \cdot \boldsymbol{t}_a(\boldsymbol{y}_k) \right) .$$

Correspondingly, the $i$'th diagonal entry of the FIM $\mathcal{I}(\boldsymbol{\theta})$, which is the expected value of $\hat{\mathcal{I}}_1(\theta_i)$ and $\hat{\mathcal{I}}_2(\theta_i)$, is denoted as $\mathcal{I}(\theta_i)$. Notation is abused in $\mathcal{I}(\theta_i)$, $\hat{\mathcal{I}}_1(\theta_i)$, and $\hat{\mathcal{I}}_2(\theta_i)$ as they depend on the whole $\boldsymbol{\theta}$ vector rather than solely on $\theta_i$. Clearly $\hat{\mathcal{I}}_1(\theta_i) \geq 0$, while there is no guarantee for $\hat{\mathcal{I}}_2(\theta_i)$ which can be negative. Our results will be expressed in terms of the (central) moments of $\boldsymbol{t}(\boldsymbol{y})$:

$$\boldsymbol{\eta}_a(\boldsymbol{x}) \doteq \mathop{\mathbb{E}}_{p(\boldsymbol{y}\,|\,\boldsymbol{x})}[\boldsymbol{t}_a(\boldsymbol{y})]; \qquad \mathcal{I}(\boldsymbol{h}\,|\,\boldsymbol{x}) \doteq \mathop{\mathbb{E}}_{p(\boldsymbol{y}\,|\,\boldsymbol{x})}[(\boldsymbol{t}(\boldsymbol{y}) - \boldsymbol{\eta}(\boldsymbol{x}))(\boldsymbol{t}(\boldsymbol{y}) - \boldsymbol{\eta}(\boldsymbol{x}))^\top];$$

$$\mathcal{K}^p(\boldsymbol{t}\,|\,\boldsymbol{x}) \doteq \mathop{\mathbb{E}}_{p(\boldsymbol{y}\,|\,\boldsymbol{x})}[(\boldsymbol{t}(\boldsymbol{y}) - \boldsymbol{\eta}(\boldsymbol{x})) \otimes (\boldsymbol{t}(\boldsymbol{y}) - \boldsymbol{\eta}(\boldsymbol{x})) \otimes (\boldsymbol{t}(\boldsymbol{y}) - \boldsymbol{\eta}(\boldsymbol{x})) \otimes (\boldsymbol{t}(\boldsymbol{y}) - \boldsymbol{\eta}(\boldsymbol{x}))],$$

where "$\otimes$" denotes the tensor product. We denote the covariance of $\boldsymbol{t}$ w.r.t. to $p(\boldsymbol{y}\,|\,\boldsymbol{x})$ as $\mathrm{Cov}^p(\boldsymbol{t}\,|\,\boldsymbol{x})$ — noting that $\mathcal{I}(\boldsymbol{h}\,|\,\boldsymbol{x}) = \mathrm{Cov}^p(\boldsymbol{t}\,|\,\boldsymbol{x})$. The 4D tensor $\mathcal{K}^p(\boldsymbol{t}\,|\,\boldsymbol{x})$ denotes the $4^{\mathrm{th}}$ central moment of $\boldsymbol{t}(\boldsymbol{y})$ w.r.t. $p(\boldsymbol{y}\,|\,\boldsymbol{x})$. These central moments correspond to the cumulants of $\boldsymbol{t}(\boldsymbol{y})$, $i.e.$ the derivatives of $F$ w.r.t. the natural parameters $\boldsymbol{h}(\boldsymbol{x})$ of the exponential family. Therefore, the derivatives of $F$ in $\hat{\mathcal{I}}_1(\theta_i)$ and $\hat{\mathcal{I}}_2(\theta_i)$ can further be written in terms of $\boldsymbol{\eta}(\boldsymbol{x})$ and $\mathcal{I}(\boldsymbol{h}\,|\,\boldsymbol{x})$ following the chain rule. Practically, $\hat{\mathcal{I}}_1$ and $\hat{\mathcal{I}}_2$ involves computing the Jacobian $\partial \boldsymbol{h}(\boldsymbol{x})/\partial\theta_i$ and the Hessian $\partial^2 \boldsymbol{h}(\boldsymbol{x})/\partial^2\theta_i$, respectively.

---

[1]This and subsequent derivations can be found in the appendix.

Table 1: Exponential family statistics with eigenvalue upper bounds for moments. For classification, $\sigma(\boldsymbol{x})$ denotes the softmax of logit $\boldsymbol{h}(\boldsymbol{x})$. † denotes exact eigenvalues rather than upper bounds.

| Setting | Exp. Family | Output $\mathcal{Y}$ | Sufficient Statistic $\boldsymbol{t}(\boldsymbol{y})$ | UB $\lambda_{\max}(\mathcal{I}(\boldsymbol{h}\,|\,\boldsymbol{x}))$ | UB $\tilde{\lambda}_{\max}(\mathcal{K}(\boldsymbol{t}\,|\,\boldsymbol{x}))$ |
|---|---|---|---|---|---|
| Regression | (Iso.) Gaussian | $\Re^T$ | $\boldsymbol{y}$ | 1† | 3† |
| Classification | Categorical | $[C] \subset \Re$ | $(\llbracket y=0 \rrbracket, \ldots, \llbracket y=C \rrbracket)$ | $\min\{\sigma_{\max}(\boldsymbol{x}), 1 - \|\sigma(\boldsymbol{x})\|_2^2\}$ | $2 \cdot \min\{\sigma_{\max}(\boldsymbol{x}), 1 - \|\sigma(\boldsymbol{x})\|_2^2\}$ |

In practice, both estimators can be computed via automatic differentiation [33, 6]. In terms of complexity, by restricting to just the diagonal elements $\mathcal{I}(\theta_i)$, we need to calculate $\mathcal{O}(\dim(\boldsymbol{\theta}))$ elements (originally $\mathcal{O}(\dim(\boldsymbol{\theta}) \times \dim(\boldsymbol{\theta}))$ for the full FIM). Although the log-partition function for general exponential family distributions can be complicated, for the ones used in NNs (determined by the loss functions used in optimization) [37] the log-partition function $F$ is usually in closed-form; and thus the cumulants $\boldsymbol{\eta}(\boldsymbol{x})$ and $\mathcal{I}(\boldsymbol{h}\,|\,\boldsymbol{x})$ can be calculated efficiently.

Indeed, the primary cost of the estimators comes from evaluating the gradient information of the NN, given by $\partial \boldsymbol{h}(\boldsymbol{x})/\partial \theta_i$ and $\partial^2 \boldsymbol{h}(\boldsymbol{x})/\partial^2 \theta_i$. The former can be calculated easily. The latter is costly even when restricted to the diagonal elements of the FIM. With the Hessian's quadratic complexity, in practice approximations are used to reduce the computational overhead [4, 45, 46, 8]. In this case, additional error and (potentially) variance may be introduced as a result of the Hessian approximation. Note, the computational cost of the Hessian can still be manageable for the last few layers close to the output. By the chain rule, we only require a sub-computational graph from the output layer to a certain layer to compute the Hessian of that layer. Despite this, there is still a memory cost that scales quadratically with the number of parameters for non-linear activation functions [6].

The high cost of Hessian computation does not justify refraining from using $\hat{\mathcal{I}}_2$. Depending on the setting (chosen loss function), an estimator's variance can outweigh the benefits of lower computational costs [37]. This is especially true when the FIM is used in an offline setting — where the Hessian's cost can be tolerated — to study, *e.g.*, the singular structure of the neuromanifold [2, 40], the curvature of the loss [7], to quantify model sensitivity [28], and to evaluate the quality of the local optimum [14, 15], *etc.*

To study the quality of $\hat{\mathcal{I}}_1(\boldsymbol{\theta})$ and $\hat{\mathcal{I}}_2(\boldsymbol{\theta})$, it is natural to examine the variance of the estimators [37]: $\mathcal{V}_j(\theta_i\,|\,\boldsymbol{x}) \doteq \mathrm{Var}(\hat{\mathcal{I}}_j(\theta_i\,|\,\boldsymbol{x}))$, where $\hat{\mathcal{I}}_j(\theta_i\,|\,\boldsymbol{x}) \doteq \left(\hat{\mathcal{I}}_j(\boldsymbol{\theta}\,|\,\boldsymbol{x})\right)_{ii}$ $(j \in \{1,2\})$ is the $i$'th diagonal element of $\hat{\mathcal{I}}_j(\boldsymbol{\theta}\,|\,\boldsymbol{x})$. Similar to $\hat{\mathcal{I}}_1(\theta_i)$ and $\hat{\mathcal{I}}_2(\theta_i)$, $\mathcal{V}_j(\theta_i\,|\,\boldsymbol{x})$ and $\hat{\mathcal{I}}_j(\theta_i\,|\,\boldsymbol{x})$ depend on the vector $\boldsymbol{\theta}$ and are abuses of notation. An estimator with a smaller variance indicates that it is more accurate and more likely to be close to the true FIM. Based on the variance, one can derive sample complexity bounds of the diagonal FIM via Chebyshev's inequality, see for instance [37, Section 3.4].

By its definition, $\mathcal{V}_j(\theta_i\,|\,\boldsymbol{x})$ has a simple closed form, which was proved in [37] and is restated below.

**Lemma 3.1.** $\forall \boldsymbol{x} \in \Re^I$, $\forall i = 1, \ldots, \dim(\boldsymbol{\theta})$,

$$\mathcal{I}(\theta_i\,|\,\boldsymbol{x}) = \partial_i \boldsymbol{h}^a(\boldsymbol{x}) \partial_i \boldsymbol{h}^b(\boldsymbol{x}) \cdot \mathcal{I}_{ab}(\boldsymbol{h}\,|\,\boldsymbol{x}), \tag{4}$$

$$\mathcal{V}_1(\theta_i\,|\,\boldsymbol{x}) = \frac{1}{N} \cdot \partial_i \boldsymbol{h}^a(\boldsymbol{x}) \partial_i \boldsymbol{h}^b(\boldsymbol{x}) \partial_i \boldsymbol{h}^c(\boldsymbol{x}) \partial_i \boldsymbol{h}^d(\boldsymbol{x}) \cdot \left[ \mathcal{K}^p_{abcd}(\boldsymbol{t}\,|\,\boldsymbol{x}) - \mathcal{I}_{ab}(\boldsymbol{h}\,|\,\boldsymbol{x}) \cdot \mathcal{I}_{cd}(\boldsymbol{h}\,|\,\boldsymbol{x}) \right], \tag{5}$$

$$\mathcal{V}_2(\theta_i\,|\,\boldsymbol{x}) = \frac{1}{N} \cdot \partial_i^2 \boldsymbol{h}^a(\boldsymbol{x}) \partial_i^2 \boldsymbol{h}^b(\boldsymbol{x}) \cdot \mathcal{I}_{ab}(\boldsymbol{h}\,|\,\boldsymbol{x}). \tag{6}$$

Given a fixed $\boldsymbol{x} \in \Re^I$, both $\mathcal{V}_1(\theta_i\,|\,\boldsymbol{x})$ and $\mathcal{V}_2(\theta_i\,|\,\boldsymbol{x})$ have an order of $\mathcal{O}(1/N)$, with $N$ denoting the number of samples of $\boldsymbol{y}_k$. They further depend on two factors: ① the derivatives of the parameter-output mapping $\boldsymbol{\theta} \to \boldsymbol{h}$ stored in a $T \times \dim(\boldsymbol{\theta})$ matrix, either $\partial_i \boldsymbol{h}^a(\boldsymbol{x})$ or $\partial_i^2 \boldsymbol{h}^a(\boldsymbol{x})$, where the latter can be expensive to calculate; and ② the central moments of $\boldsymbol{t}(\boldsymbol{y})$, whose computation only scales with $T$ (the number of output units) and is independent to $\dim(\boldsymbol{\theta})$.

From an information geometry [1] perspective, $\mathcal{I}(\boldsymbol{\theta})$, $\mathcal{V}_1(\boldsymbol{\theta})$, and $\mathcal{V}_2(\boldsymbol{\theta})$ are all pullback tensors of different orders. For example, $\mathcal{I}(\boldsymbol{\theta})$ is the pullback tensor of $\mathcal{I}(\boldsymbol{h})$ and the singular semi-Riemannian metric [40]. They induce the geometric structures of the neuromanifold (parameterized by $\boldsymbol{\theta}$) based on the corresponding low dimensional structures of the exponential family (parameterized by $\boldsymbol{h}$).

## 4 Practical Variance Estimation

To further understand the dependencies of the derivative and central moment terms, the FIM $\mathcal{I}(\theta_i \,|\, \boldsymbol{x})$ and variances of estimators $\hat{\mathcal{I}}_j(\theta_i \,|\, \boldsymbol{x})$ can be bounded to strengthen intuition and to provide a computationally convenient proxy of the interested quantities.

**Theorem 4.1.** $\forall \boldsymbol{x} \in \Re^I$,

$$\|\partial_i \boldsymbol{h}(\boldsymbol{x})\|_2^2 \cdot \lambda_{\min}(\mathcal{I}(\boldsymbol{h} \,|\, \boldsymbol{x})) \leq \mathcal{I}(\theta_i \,|\, \boldsymbol{x}) \ \leq \|\partial_i \boldsymbol{h}(\boldsymbol{x})\|_2^2 \cdot \lambda_{\max}(\mathcal{I}(\boldsymbol{h} \,|\, \boldsymbol{x})), \tag{7}$$

$$\frac{1}{N} \cdot \|\partial_i \boldsymbol{h}(\boldsymbol{x})\|_2^4 \cdot \tilde{\lambda}_{\min}(\mathcal{M}) \leq \mathcal{V}_1(\theta_i \,|\, \boldsymbol{x}) \leq \frac{1}{N} \cdot \|\partial_i \boldsymbol{h}(\boldsymbol{x})\|_2^4 \cdot \tilde{\lambda}_{\max}(\mathcal{M}), \tag{8}$$

$$\frac{1}{N} \cdot \|\partial_i^2 \boldsymbol{h}(\boldsymbol{x})\|_2^2 \cdot \lambda_{\min}(\mathcal{I}(\boldsymbol{h} \,|\, \boldsymbol{x})) \leq \mathcal{V}_2(\theta_i \,|\, \boldsymbol{x}) \leq \frac{1}{N} \cdot \|\partial_i^2 \boldsymbol{h}(\boldsymbol{x})\|_2^2 \cdot \lambda_{\max}(\mathcal{I}(\boldsymbol{h} \,|\, \boldsymbol{x})), \tag{9}$$

*where $\mathcal{M} = \mathcal{K}^p(\boldsymbol{t} \,|\, \boldsymbol{x}) - \mathcal{I}(\boldsymbol{h} \,|\, \boldsymbol{x}) \otimes \mathcal{I}(\boldsymbol{h} \,|\, \boldsymbol{x})$; $\lambda_{\min} / \lambda_{\max}$ denotes the minimum / maximum matrix eigenvalue; and $\tilde{\lambda}_{\min}, \tilde{\lambda}_{\max} \colon \Re^{T \times T \times T \times T} \to \Re$ are defined as*

$$\tilde{\lambda}_{\min}(\mathcal{T}) \doteq \inf_{\boldsymbol{u} \colon \|\boldsymbol{u}\|_2 = 1} \boldsymbol{u}^a \boldsymbol{u}^b \boldsymbol{u}^c \boldsymbol{u}^d \mathcal{T}_{abcd}; \quad and \quad \tilde{\lambda}_{\max}(\mathcal{T}) \doteq \sup_{\boldsymbol{u} \colon \|\boldsymbol{u}\|_2 = 1} \boldsymbol{u}^a \boldsymbol{u}^b \boldsymbol{u}^c \boldsymbol{u}^d \mathcal{T}_{abcd}. \tag{10}$$

To help ground Theorem 4.1, we summarize different sufficient statistics quantities for common learning settings in Table 1 — with further learning setting implications presented in Section 5. Note that Eqs. (8) and (9) (and many subsequent results) can be further generalized for off-diagonal elements. See Appendix C for details. Compared to prior work [37], Theorem 4.1 provides bounds for individual elements of the variance tensors, where the NN weights (the derivatives) and sufficient statistics (the eigenvalues) are neatly disentangled into a product. From a technical point of view, this comes from a difference in proof technique: we utilize variational definitions and computations of eigenvalues to establish bounds whereas [37] primarily applies Hölder's inequality.

We stress that $\tilde{\lambda}_{\min}(\mathcal{T})$ and $\tilde{\lambda}_{\max}(\mathcal{T})$ in Eq. (10) correspond to tensor eigenvalues iff $\mathcal{T}$ is a super-symmetric tensor [23] (a.k.a. totally symmetric tensor), *i.e.*, indices are permutation invariant. In this case, Eq. (10) is exactly the maximum and minimum Z-eigenvalues. These variational forms mirror the Courant-Fischer min-max theorem for symmetric matrices [42]. In the case of Eq. (8), with $\mathcal{M} = \mathcal{K}^p(\boldsymbol{t} \,|\, \boldsymbol{x}) - \mathcal{I}(\boldsymbol{h} \,|\, \boldsymbol{x}) \otimes \mathcal{I}(\boldsymbol{h} \,|\, \boldsymbol{x})$, the tensor is not a supersymmetric tensor in general. Despite this, we note that the lower bound of Eq. (8) is non-trivial. A weaker bound than Eq. (8) can be established based on the Z-eigenvalue of the supersymmetric tensor $\mathcal{K}^p(\boldsymbol{t} \,|\, \boldsymbol{x})$.

**Corollary 4.2.** $\forall \boldsymbol{x} \in \Re^I$,

$$\tilde{\lambda}_{\min}(\mathcal{K}^p(\boldsymbol{t} \,|\, \boldsymbol{x}) - \mathcal{I}(\boldsymbol{h} \,|\, \boldsymbol{x}) \otimes \mathcal{I}(\boldsymbol{h} \,|\, \boldsymbol{x})) \geq \max\left\{0, \tilde{\lambda}_{\min}(\mathcal{K}^p(\boldsymbol{t} \,|\, \boldsymbol{x})) - \lambda_{\max}^2(\mathcal{I}(\boldsymbol{h} \,|\, \boldsymbol{x}))\right\}; \tag{11}$$

$$\tilde{\lambda}_{\max}(\mathcal{K}^p(\boldsymbol{t} \,|\, \boldsymbol{x}) - \mathcal{I}(\boldsymbol{h} \,|\, \boldsymbol{x}) \otimes \mathcal{I}(\boldsymbol{h} \,|\, \boldsymbol{x})) \leq \tilde{\lambda}_{\max}(\mathcal{K}^p(\boldsymbol{t} \,|\, \boldsymbol{x})) - \lambda_{\min}^2(\mathcal{I}(\boldsymbol{h} \,|\, \boldsymbol{x})). \tag{12}$$

The tensor eigenvalue is typically expensive to calculate. However in our case, the eigenvalues $\tilde{\lambda}_{\min}(\mathcal{K}^p(\boldsymbol{t} \,|\, \boldsymbol{x}))$ and $\tilde{\lambda}_{\max}(\mathcal{K}^p(\boldsymbol{t} \,|\, \boldsymbol{x}))$ on the RHS of Eqs. (11) and (12) can be calculated via [17]'s method with $\mathcal{O}(T^4/4!)$ complexity. In this paper, we assume $T$ is reasonably bounded and are mainly concerned with the complexity w.r.t. $\dim(\boldsymbol{\theta})$. From this perspective, all our bounds scale linearly w.r.t. $\dim(\boldsymbol{\theta})$, and thus can be computed efficiently.

When $\boldsymbol{t}(\boldsymbol{y}) - \boldsymbol{\eta}(\boldsymbol{x})$ is bounded (*e.g.* in classification), we can upper bound $\tilde{\lambda}_{\max}(\mathcal{K}^p(\boldsymbol{t} \,|\, \boldsymbol{x}))$ with $\lambda_{\max}(\mathcal{I}(\boldsymbol{h} \,|\, \boldsymbol{x}))$, which is easier to calculate.

**Proposition 4.3.** *Suppose $\|\boldsymbol{t}(\boldsymbol{y}) - \boldsymbol{\eta}(\boldsymbol{x})\|_2^2 \leq B$. Then, $\tilde{\lambda}_{\max}(\mathcal{K}^p(\boldsymbol{t} \,|\, \boldsymbol{x})) \leq B\lambda_{\max}(\mathcal{I}(\boldsymbol{h} \,|\, \boldsymbol{x})) \leq B^2$.*

As long as the sufficient statistics $\boldsymbol{t}(\boldsymbol{y})$ has bounded norm $\|\boldsymbol{t}\|_2$, we have that $\|\boldsymbol{t}(\boldsymbol{y}) - \boldsymbol{\eta}(\boldsymbol{x})\|_2^2 \leq 4\|\boldsymbol{t}\|_2^2 < \infty$. A similar lower bound can be established for the minimum tensor eigenvalue $\tilde{\lambda}_{\min}(\mathcal{K}^p(\boldsymbol{t} \,|\, \boldsymbol{x})) \geq \lambda_{\min}^2(\mathcal{I}(\boldsymbol{h} \,|\, \boldsymbol{x}))$, but this ends up being trivial when applying Corollary 4.2's lower bound, Eq. (11).

Examining Theorem 4.1 reveals several trade-offs. An immediate observation is that the first order gradients of $\boldsymbol{h}(\boldsymbol{x})$ correspond to the robustness of $\boldsymbol{h}$ to parameter misspecification (w.r.t. an input

$\boldsymbol{x}$). As such, from the bounds in Eqs. (7) and (8), the scale of $\mathcal{I}(\theta_i \,|\, \boldsymbol{x})$ and $\mathcal{V}_1(\theta_i \,|\, \boldsymbol{x})$ will be large when small shifts in parameter space yield large changes in the output $\boldsymbol{h}(\boldsymbol{x})$. Another observation is how the spectrum of $\mathcal{I}(\boldsymbol{h} \,|\, \boldsymbol{x})$ affects the scale of $\mathcal{I}(\theta_i \,|\, \boldsymbol{x})$ and the estimator variances. In particular, when $\lambda_{\min}(\mathcal{I}(\boldsymbol{h} \,|\, \boldsymbol{x}))$ increases, the scale of $\mathcal{V}_1(\theta_i \,|\, \boldsymbol{x})$ decreases but the scale of $\mathcal{I}(\theta_i \,|\, \boldsymbol{x})$ and $\mathcal{V}_2(\theta_i \,|\, \boldsymbol{x})$ increases. When $\lambda_{\max}(\mathcal{I}(\boldsymbol{h} \,|\, \boldsymbol{x}))$ decreases, then the opposite scaling occurs. With these two observations, there is a tension in how the scale of $\mathcal{I}(\theta_i \,|\, \boldsymbol{x})$ follows the different variances $\mathcal{V}_1(\theta_i \,|\, \boldsymbol{x})$ and $\mathcal{V}_2(\theta_i \,|\, \boldsymbol{x})$. The element-wise FIM $\mathcal{I}(\theta_i \,|\, \boldsymbol{x})$ follows $\mathcal{V}_1(\theta_i \,|\, \boldsymbol{x})$ in terms of the scale of NN derivatives $\|\partial_i \boldsymbol{h}(\boldsymbol{x})\|_2$; at the same time, $\mathcal{I}(\theta_i \,|\, \boldsymbol{x})$ follows $\mathcal{V}_2(\theta_i \,|\, \boldsymbol{x})$ in terms of the spectrum of sufficient statistics moment $\mathcal{I}(\boldsymbol{h} \,|\, \boldsymbol{x})$.

**Remark 4.4.** Typically, $\boldsymbol{h}$ is the linear output units: $\boldsymbol{h}(\boldsymbol{x}) = \boldsymbol{W}_{-1}\boldsymbol{h}_{-1}(\boldsymbol{x})$, where $\boldsymbol{W}_{-1}$ is the weights of the last layer, and $\boldsymbol{h}_{-1}(\boldsymbol{x})$ is the second last layer's output. We have $\mathcal{V}_2(\theta_i \,|\, \boldsymbol{x}) = 0 \leq \mathcal{V}_1(\theta_i \,|\, \boldsymbol{x})$ for any $\theta_i$ in $\boldsymbol{W}_{-1}$. A smaller variance $\mathcal{V}_2(\theta_i \,|\, \boldsymbol{x})$ is guaranteed for the last layer regardless of the choice of the exponential family in Eq. (1).

**Remark 4.5.** $\boldsymbol{h}(\boldsymbol{x}) = \boldsymbol{w}^j_{-1}\phi(\boldsymbol{h}^\top_{-2}(\boldsymbol{x})\boldsymbol{w}^j_{-2} + C_{-2}) + \boldsymbol{c}_{-1}$ defines the NN mapping w.r.t. the $j$'th neuron in the second last layer, where $\boldsymbol{w}^j_{-2}$ and $\boldsymbol{w}^j_{-1}$ are incoming and outgoing links of the interested neuron, respectively; $\boldsymbol{h}_{-2}(\boldsymbol{x})$ is the output of the third last layer; and $\phi$ is the activation function. The 'constants' $C_{-2}$ and $\boldsymbol{c}_{-1}$ denote an aggregation of all terms which are independent of $\boldsymbol{w}^j_{-2}$ and $\boldsymbol{w}^j_{-1}$ in their respective layers. The Hessian of $\boldsymbol{h}_k(\boldsymbol{x})$ w.r.t. $\boldsymbol{w}^j_{-2}$ is $\partial^2 \boldsymbol{h}_k(\boldsymbol{x}) = (\boldsymbol{w}^j_{-1})_k \cdot \phi''(\boldsymbol{h}^\top_{-2}(\boldsymbol{x})\boldsymbol{w}^j_{-2} + C_{-2}) \cdot (\boldsymbol{h}_{-2}(\boldsymbol{x})\boldsymbol{h}^\top_{-2}(\boldsymbol{x}))$. By Theorem 4.1, $\mathcal{V}_2(\theta_i \,|\, \boldsymbol{x})$ can be arbitrarily small depending on $\phi''(\boldsymbol{h}^\top_{-2}(\boldsymbol{x})\boldsymbol{w}^j_{-2} + C_{-2})$. For example, if $\phi(t) = 1/(1 + \exp(-t))$, then $\phi''(t) = \phi(t)(1 - \phi(t))(1 - 2\phi(t))$. In this case, for a neuron in the second last layer, a sufficient condition for $\mathcal{V}_2(\theta_i \,|\, \boldsymbol{x}) = 0$ (and having $\hat{\mathcal{I}}_2$ favored against $\hat{\mathcal{I}}_1$) is $\boldsymbol{h}^\top_{-2}(\boldsymbol{x})\boldsymbol{w}^j_{-2} + C_{-2} = 0$ for the neuron's pre-activation. When the pre-activation value is saturated ($-\infty$ or $\infty$), we also have that $\mathcal{V}_1(\theta_i \,|\, \boldsymbol{x}) = \mathcal{V}_2(\theta_i \,|\, \boldsymbol{x}) = 0$. Alternatively, suppose that $\phi(t) = \text{SoftPlus}(t) \doteq \log(1 + \exp(t))$, a continuous relaxation of ReLU, then $\phi''(t) = \phi'(t)(1 - \phi'(t))$ where $\phi'(t) = 1/(1 + \exp(-t))$. Then a sufficient condition for $\mathcal{V}_2(\theta_i \,|\, \boldsymbol{x}) = 0$ with $\mathcal{V}_1(\theta_i \,|\, \boldsymbol{x}) \neq 0$ for a neuron in the second last layer is $\boldsymbol{h}^\top_{-2}(\boldsymbol{x})\boldsymbol{w}^j_{-2} + C_{-2} \to +\infty$.

These observations are further clarified by looking at related quantities over multiple parameters. So far we have only examined the variance of the FIM element-wise w.r.t. parameters $\theta_i$. To study all parameters $\boldsymbol{\theta}$ jointly, we consider the *trace variances* of the FIM estimators: for any $j \in \{1, 2\}$, $\mathcal{V}_j(\boldsymbol{\theta} \,|\, \boldsymbol{x})$ denotes the trace of the covariance matrix of $\text{diag}(\hat{\mathcal{I}}_j(\boldsymbol{\theta} \,|\, \boldsymbol{x}))$, where $\text{diag}(\cdot)$ extracts a matrix's diagonal elements into a column vector. We present upper bounds of these joint quantities.

**Corollary 4.6.** *For any $\boldsymbol{x} \in \Re^I$,*

$$\text{tr}\,(\mathcal{I}(\boldsymbol{\theta} \,|\, \boldsymbol{x})) \leq \|\partial \boldsymbol{h}(\boldsymbol{x})\|_F \cdot \min\{\text{tr}\,(\mathcal{I}(\boldsymbol{h} \,|\, \boldsymbol{x})), \|\partial \boldsymbol{h}(\boldsymbol{x})\|_F \cdot \lambda_{\max}(\mathcal{I}(\boldsymbol{h} \,|\, \boldsymbol{x}))\}; \tag{13}$$

$$\mathcal{V}_1(\boldsymbol{\theta} \,|\, \boldsymbol{x}) \leq \frac{1}{N} \cdot \|\partial \boldsymbol{h}(\boldsymbol{x})\|_F^2 \cdot \min\left\{ \sum_{t,u=1}^T \mathcal{K}^p_{ttuu}(\boldsymbol{t} \,|\, \boldsymbol{x}) - \|\mathcal{I}(\boldsymbol{h} \,|\, \boldsymbol{x})\|_F^2, \|\partial \boldsymbol{h}(\boldsymbol{x})\|_F^2 \cdot \tilde{\lambda}_{\max}(\mathcal{M}) \right\}; \tag{14}$$

$$\mathcal{V}_2(\boldsymbol{\theta} \,|\, \boldsymbol{x}) \leq \frac{1}{N} \cdot \|\text{dHes}(\boldsymbol{h} \,|\, \boldsymbol{x})\|_F \cdot \min\{\text{tr}(\mathcal{I}(\boldsymbol{h} \,|\, \boldsymbol{x})), \|\text{dHes}(\boldsymbol{h} \,|\, \boldsymbol{x})\|_F \cdot \lambda_{\max}(\mathcal{I}(\boldsymbol{h} \,|\, \boldsymbol{x}))\}, \tag{15}$$

*where* $\text{dHes}(\boldsymbol{h} \,|\, \boldsymbol{x}) \doteq (\text{diag}(\text{Hes}(\boldsymbol{h}_1 \,|\, \boldsymbol{x})), \dots, \text{diag}(\text{Hes}(\boldsymbol{h}_T \,|\, \boldsymbol{x})))$ *and* $\|\cdot\|_F$ *is the Frobenius norm.*

This upper bound comes from integrating the parameter-wise variances in Theorem 4.1 and incorporating a trace variance bound which utilizes the full spectrum of the NN derivatives and sufficient statistics quantities. This is fully depicted in Theorem D.1. Lower bounds can also be derived in terms of singular values (deferred to the Appendix). Note the upper bounds in Corollary 4.6 can be improved by expressing the $\min$ function's first term with singular value quantities.

Having the $\min$ function in Corollary 4.6 is helpful as it shows a trade-off between two upper bounds: the scale of NN derivatives $\partial \boldsymbol{h}(\boldsymbol{x})$ and $\partial^2 \boldsymbol{h}(\boldsymbol{x})$ versus the spectrum of the sufficient statistic terms. In the case of Eqs. (13) and (15), the trace of $\mathcal{I}(\boldsymbol{h} \,|\, \boldsymbol{x})$ is exactly the sum of all eigenvalues, including $\lambda_{\max}(\mathcal{I}(\boldsymbol{h} \,|\, \boldsymbol{x}))$. This can be helpful when the scale of the NN derivatives are not bounded by a small value. It should be noted that, by the chain rule, these NN derivatives scale with the overall sharpness / flatness [22] of the landscape of the loss, *i.e.*, the log-likelihood of Eq. (1). For NNs with large derivatives, the first term of the $\min$ could yield tight bounds of the variance, and one can therefore avoid dealing with the quadratic scaling of $\|\partial \boldsymbol{h}(\boldsymbol{x})\|$ in the second term. On the other hand, if the sharpness of the NN $\boldsymbol{h}$ can be controlled, *e.g.* via sharpness aware minimization [10], then one can

benefit from the second term of the $\min$ and avoid computing the full spectrum of $\mathcal{I}(\boldsymbol{h} \,|\, \boldsymbol{x})$ in the first term.

**Joint FIM Estimators**   In the above, we considered the variance of conditional FIMs, which can scale differently depending on the input $\boldsymbol{x}$. Prior work's analysis was limited to that of conditional FIMs (and their estimators) [37]. Nevertheless, the 'joint FIM' estimators $\hat{\mathcal{I}}_j(\theta_i)$ depend on sampling of $\boldsymbol{x}$ w.r.t. the data distribution $q(\boldsymbol{x})$. The bounds in Eq. (7) can be extended to the joint FIM $\mathcal{I}(\theta_i) \doteq \mathbb{E}_{q(\boldsymbol{x})} \mathcal{I}(\theta_i \,|\, \boldsymbol{x})$ by simply taking an expectation $\mathbb{E}_{q(\boldsymbol{x})}$ over the bounds. To analyze the variances of the joint FIM estimators $\mathcal{V}_j(\theta_i)$, we present the following theorem which connects the prior results established for $\mathcal{V}_j(\theta_i \,|\, \boldsymbol{x})$, *e.g.* Theorem 4.1, via the law of total variance.

**Theorem 4.7.** *For any $j \in \{1, 2\}$, given $N_x$ samples of $\boldsymbol{x} \sim q(\boldsymbol{x})$ and $N$ samples of $\boldsymbol{y}_{\,|\, \boldsymbol{x}} \sim p(\boldsymbol{y} \,|\, \boldsymbol{x})$ for each $\boldsymbol{x}$ sampled,*

$$\mathcal{V}_j(\theta_i) = \frac{1}{N_x} \cdot \mathrm{Var}\left(\mathcal{I}(\theta_i \,|\, \boldsymbol{x})\right) + \frac{1}{N_x} \cdot \underset{q(\boldsymbol{x})}{\mathbb{E}}\left[\mathcal{V}_j(\theta_i \,|\, \boldsymbol{x})\right], \tag{16}$$

*where $\mathrm{Var}\left(\mathcal{I}(\theta_i \,|\, \boldsymbol{x})\right)$ is the variance of $\mathcal{I}(\theta_i \,|\, \boldsymbol{x})$ w.r.t. $q(\boldsymbol{x})$.*

The dependence on $N$, the number of samples of $\boldsymbol{y}$ for each fixed $\boldsymbol{x}$, is hidden in $\mathcal{V}_j(\theta_i \,|\, \boldsymbol{x})$. When $N = 1$, the hierarchical sampling described in the Theorem corresponds to an i.i.d. sampling of the joint distribution $p(\boldsymbol{x}, \boldsymbol{y})$.

The variance incurred when estimating the FIM has two components. The first term on the RHS of Eq. (16) characterizes the randomness of the FIM w.r.t. $q(\boldsymbol{x})$, *i.e.*, the input randomness. It vanishes when the FIM is estimated by taking the expectation w.r.t. $q(\boldsymbol{x})$, or the number of samples $\boldsymbol{x}$ is large enough. The second term (although also depending on $N_x$) comes from the sampling of $\boldsymbol{y}_{\,|\, \boldsymbol{x}}$ according to $p(\boldsymbol{y} \,|\, \boldsymbol{x})$, *i.e.*, the output randomness, which scales with the central moments of $\boldsymbol{t}(\boldsymbol{y})$. If the NN is trained so that $p(\boldsymbol{y} \,|\, \boldsymbol{x})$ tends to be deterministic, this term will disappear leaving the first term to dominate. Eq. (16) can be further generalized using the law of total covariance to extend prior work considering conditional FIM covariances [37] to joint FIM covariances. Theorem 4.7 connects the variance of assuming a fixed input $\boldsymbol{x}$ with multiple samples $\boldsymbol{y}_k$ with the variance of pairs of samples $(\boldsymbol{x}_k, \boldsymbol{y}_k)$. The $\mathcal{V}_j(\theta_i \,|\, \boldsymbol{x})$ bounds in this section can thus be applied to the corresponding joint variance $\mathcal{V}_j(\theta_i)$ by using this theorem. This is straightforward and omitted.

The first variance term in Theorem 4.7 is difficult to compute in practice: it relies on how the closed-form FIM varies w.r.t. $q(\boldsymbol{x})$. As such, it is useful to bound the first term into computable quantities.

**Lemma 4.8.** $\mathrm{Var}\left(\mathcal{I}(\theta_i \,|\, \boldsymbol{x})\right) \leq \mathbb{E}_{q(\boldsymbol{x})}\left[\|\partial_i \boldsymbol{h}(\boldsymbol{x})\|_2^4 \cdot \lambda_{\max}^2(\mathcal{I}(\boldsymbol{h} \,|\, \boldsymbol{x}))\right].$

This upper bound is very similar to the 4th central moment term $\tilde{\lambda}_{\max}(\mathcal{K}^p(\boldsymbol{t} \,|\, \boldsymbol{x}))$ when considering the variance upper bound $\mathcal{V}_2(\theta_i \,|\, \boldsymbol{x})$ in Theorem 4.1 and Corollary 4.2. In general, the eigenvalue terms of Lemma 4.8 and Theorem 4.1 are distinct, *i.e.*, $\lambda_{\max}^2(\mathcal{I}(\boldsymbol{h} \,|\, \boldsymbol{x})) \neq \tilde{\lambda}_{\max}(\mathcal{K}^p(\boldsymbol{t} \,|\, \boldsymbol{x}))$. This is especially true for the classification and regression problems explored in this paper (see Table 1). However, the maximum eigenvalues can be related for exponential families with bounded sufficient statistic via Proposition 4.3, making both bounds depend only on $\lambda_{\max}(\mathcal{I}(\boldsymbol{h} \,|\, \boldsymbol{x}))$.

The total number of samples of $(\boldsymbol{x}_k, \boldsymbol{y}_k)$ is $N_x \cdot N$. In terms of sample complexity of the joint variance, using Theorem 4.7 and Lemma 4.8, the bound's rate is given by $\mathcal{O}(1/N_x + 1/(N_x \cdot N))$.

## 5   Case Studies

To make our theoretic results more concrete, we consider regression and classification settings, which correspond to specifying the exponential family in Eq. (1) to an isotropic Gaussian distribution and a categorical distribution, respectively. We include an empirical analysis of NNs trained on MNIST. Notably, our analysis considers general multi-dimensional NN output. This extends the case studies of [37] which was limited to 1D distributions due to the limitations of their bounds (and their associated computational costs of dealing with a 4D tensor of the full covariance).

**Regression: Isotropic Gaussian Distribution**   To characterize regression, we consider Gaussian distributions. As per Eq. (1), we have $\boldsymbol{t}(\boldsymbol{y}) = \boldsymbol{y} \in \Re^T$ and base measure $\pi(\boldsymbol{y}) \propto \exp(-\frac{1}{2}\boldsymbol{y}^\top \boldsymbol{y})$.

This corresponds to the case where $\boldsymbol{h}(\boldsymbol{x})$ is trained via the squared loss. In this case, $\mathcal{I}(\boldsymbol{h} \,|\, \boldsymbol{x}) = \boldsymbol{I}$ and $\mathcal{K}^p_{abcd}(\boldsymbol{t} \,|\, \boldsymbol{x}) = \mathcal{I}_{ab}(\boldsymbol{h} \,|\, \boldsymbol{x}) \cdot \mathcal{I}_{cd}(\boldsymbol{h} \,|\, \boldsymbol{x}) + \mathcal{I}_{ac}(\boldsymbol{h} \,|\, \boldsymbol{x}) \cdot \mathcal{I}_{bd}(\boldsymbol{h} \,|\, \boldsymbol{x}) + \mathcal{I}_{ad}(\boldsymbol{h} \,|\, \boldsymbol{x}) \cdot \mathcal{I}_{bc}(\boldsymbol{h} \,|\, \boldsymbol{x})$. The derivatives of the log-partition function $F(\boldsymbol{h})$ yields these central moments [37, Lemma 5]. To apply Theorem 4.1, we examine the extreme eigenvalues of $\mathcal{I}(\boldsymbol{h} \,|\, \boldsymbol{x})$ and $\mathcal{K}^p(\boldsymbol{t} \,|\, \boldsymbol{x}) - \mathcal{I}(\boldsymbol{h} \,|\, \boldsymbol{x}) \otimes \mathcal{I}(\boldsymbol{h} \,|\, \boldsymbol{x})$.

**Proposition 5.1.** *Suppose that Eq.* (1) *is an isotropic Gaussian distribution. Then:*

$$\lambda_{\min}(\mathcal{I}(\boldsymbol{h} \,|\, \boldsymbol{x})) = \lambda_{\max}(\mathcal{I}(\boldsymbol{h} \,|\, \boldsymbol{x})) = 1;$$

$$\tilde{\lambda}_{\min}(\mathcal{K}^p(\boldsymbol{t} \,|\, \boldsymbol{x}) - \mathcal{I}(\boldsymbol{h} \,|\, \boldsymbol{x}) \otimes \mathcal{I}(\boldsymbol{h} \,|\, \boldsymbol{x})) = \tilde{\lambda}_{\max}(\mathcal{K}^p(\boldsymbol{t} \,|\, \boldsymbol{x}) - \mathcal{I}(\boldsymbol{h} \,|\, \boldsymbol{x}) \otimes \mathcal{I}(\boldsymbol{h} \,|\, \boldsymbol{x})) = 2.$$

Hence, for regression the eigenvalues of sufficient statistics quantities in our bounds are equal. As such, in this case, the bound for $\mathcal{I}(\theta_i \,|\, \boldsymbol{x})$, $\mathcal{V}_1(\theta_i \,|\, \boldsymbol{x})$, and $\mathcal{V}_2(\theta_i \,|\, \boldsymbol{x})$ in Theorem 4.1 are all equalities.

As $\mathcal{I}(\theta_i \,|\, \boldsymbol{x})$ can be written exactly in terms of the gradients of $\boldsymbol{h}$, in practice one does not need to utilize a random estimator when computing the conditional FIM, which simplifies to a Gauss-Newton matrix for the squared loss [25]. When computing the FIM over a random sample $\boldsymbol{x}$, a variance still appears due to Theorem 4.7. By Lemma 4.8 and Proposition 5.1, the variance over a joint distribution is bounded by a function of the derivative $\partial_i \boldsymbol{h}(\boldsymbol{x})$ over the marginal input distributions: $\mathcal{V}(\theta_i) \leq \mathbb{E}_{q(\boldsymbol{x})}[\|\partial_i \boldsymbol{h}(\boldsymbol{x})\|_2^4] \leq \max_{\boldsymbol{x} \in \mathrm{Supp}(q)} \|\partial_i \boldsymbol{h}(\boldsymbol{x})\|_2^4$. In other words, the overall variance to approximate the joint FIM is bounded by the gradients of the NN outputs.

**Classification: Categorical Distribution**   For multi-class classification, we instantiate our exponential family with a categorical distribution (with $\pi(y) = 1$). This corresponds to training a classifier NN with the log-loss. Let $\mathcal{Y} = [C]$ for $T = C$ classes defining the possible labels. Let $\boldsymbol{t}(y) = ([\![y = 1]\!], \ldots [\![y = C]\!])$, where $[\![r]\!] = 1$ when the predicate $r$ is true and $[\![r]\!] = 0$ otherwise. Noting our results do not depend on minimal sufficiency, this $\boldsymbol{t}(y)$ is sufficient but *not minimal* sufficient. In this setting, the NN outputs $\boldsymbol{h}$ correspond to the logits of the label probabilities. The resulting probabilities $p(y \,|\, \boldsymbol{x})$ are the softmax values of $\boldsymbol{h}$ denoted by $\sigma(\boldsymbol{x}) \doteq \mathrm{SoftMax}(\boldsymbol{h}(\boldsymbol{x})) \in [0, 1]^T$.

Under this setting, we have $\mathcal{I}(\boldsymbol{h} \,|\, \boldsymbol{x}) = \mathrm{Diag}(\sigma(\boldsymbol{x})) - \sigma(\boldsymbol{x})\sigma(\boldsymbol{x})^\top$ (where $\mathrm{Diag}(\sigma(\boldsymbol{x}))$ is the diagonal matrix with its diagonal entries set to $\sigma(\boldsymbol{x})$), whose eigenvalues do not follow a convenient pattern as $C$ increases [44]. Likewise, the maximum eigenvalue of $\mathcal{K}^p(\boldsymbol{t} \,|\, \boldsymbol{x})$ is not available in simple closed form. As such, we provide upper bounds for the maximum eigenvalues of $\mathcal{I}(\boldsymbol{h} \,|\, \boldsymbol{x})$ and $\mathcal{K}^p(\boldsymbol{t} \,|\, \boldsymbol{x}) - \mathcal{I}(\boldsymbol{h} \,|\, \boldsymbol{x}) \otimes \mathcal{I}(\boldsymbol{h} \,|\, \boldsymbol{x})$ using Corollary 4.2 and Proposition 4.3.

**Theorem 5.2.** *Suppose that Eq.* (1) *is a categorical distribution. With* $\sigma_{\max}(\boldsymbol{x}) \doteq \max_k \sigma_k(\boldsymbol{x})$:

$$\lambda_{\max}(\mathcal{I}(\boldsymbol{h} \,|\, \boldsymbol{x})) \leq m(\boldsymbol{x}); \quad and \quad \tilde{\lambda}_{\max}(\mathcal{K}^p(\boldsymbol{t} \,|\, \boldsymbol{x}) - \mathcal{I}(\boldsymbol{h} \,|\, \boldsymbol{x}) \otimes \mathcal{I}(\boldsymbol{h} \,|\, \boldsymbol{x})) \leq 2 \cdot m(\boldsymbol{x}),$$

*where* $m(\boldsymbol{x}) \doteq \min\left(\sigma_{\max}(\boldsymbol{x}), 1 - \|\sigma(\boldsymbol{x})\|_2^2\right)$.

This upper bounds provides a tension. When the first term of $m(\boldsymbol{x})$ is maximized, the second is minimized, and vice-versa. In particular, the dominating term depends on the uncertainty of the NN's output. When the NN's output is near random, *e.g.* at initialization, the first term will dominate with $\sigma_{\max}(\boldsymbol{x}) \approx 1/C$. However, as the NN becomes more certain with its prediction, the second term will start dominating: a more deterministic output $p(y \,|\, \boldsymbol{x}) \to 1$ implies that $\lambda_{\max}(\mathcal{I}(\boldsymbol{h} \,|\, \boldsymbol{x})) \to 0$.

**Empirical Verification: Classification**   We examine the MNIST classification task [21] (CC BY-SA 3.0) using multilayer perceptrons (MLP) with four densely connected layers, sigmoid activations, and a dropout layer. For classification, we consider a categorical distribution with $C = 10$ class labels. For a random $\boldsymbol{x}$ from the test set, we compute both estimators $\hat{\mathcal{I}}_1(\theta_i \,|\, \boldsymbol{x})$ and $\hat{\mathcal{I}}_2(\theta_i \,|\, \boldsymbol{x})$ using $N = 5,000$ samples. We record the variances of each estimator and compute their bounds based on Theorem 4.1. For all 20 training epochs, the Fisher information (FI) and their variances of individual parameters are aggregated via arithmetic averages over four parameter groups (corresponding to the four layers).

In Fig. 2, we present the variance scale of the estimators $\hat{\mathcal{I}}_1(\theta_i \,|\, \boldsymbol{x})$ and $\hat{\mathcal{I}}_2(\theta_i \,|\, \boldsymbol{x})$ in log-space; and the tightness of the bounds in Theorem 4.1 by consider the log-ratio $\log \frac{\mathrm{UB}}{\mathcal{V}_1(\theta_i \,|\, \boldsymbol{x})}$, where UB is the upper bounds in Theorem 4.1. In this experiment, the UB is much tighter than the lower bound (LB), which is omitted in the figures for clarity. More experimental results are given in Appendix F.

We varied the NN's architecture and activation function. Across different settings, the proposed UB and LB are always valid. In Fig. 2, one can observe that the diagonal FIM and the associated

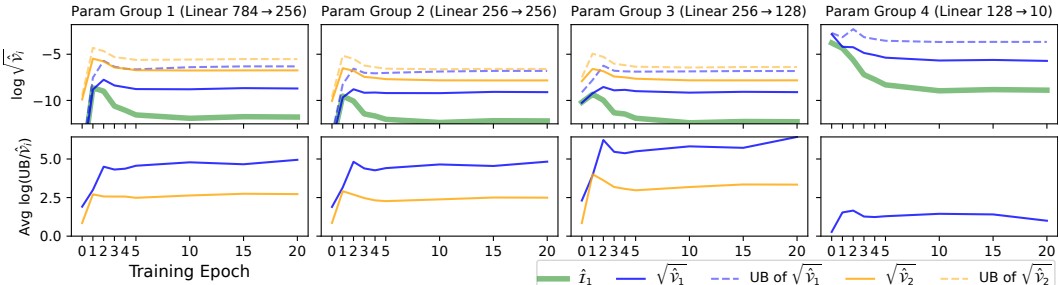

Figure 2: MNIST for a 4-layer MLP with sigmoid activations. Top: The estimated Fisher information (FI), variances, and variance bounds across 4 parameter groups and 20 training epochs. The FI (green line) is estimated using $\hat{\mathcal{I}}_1$ ($\hat{\mathcal{I}}_2$ is almost identical and not shown for clarity). The s.t.d. (square root of variance) is shown for variances and their bounds. Bottom: the log-ratio of Theorem 4.1's upper bounds (UBs) and the true variances. The closer to 0, the better the UB. In the right most column, the variance of $\hat{\mathcal{I}}_2$ vanishes: $\mathcal{V}_2(\theta_i \,|\, \boldsymbol{x}) = 0 \leq \mathcal{V}_1(\theta_i \,|\, \boldsymbol{x})$. Thus related curves of $\hat{\mathcal{I}}_2$ are not shown.

variances have a small magnitude. For example, in the first layer, $\mathcal{V}_1(\theta_i \,|\, \boldsymbol{x})$ and $\mathcal{V}_2(\theta_i \,|\, \boldsymbol{x})$ are roughly $e^{-10} \approx 5 \times 10^{-5}$. The log-ratio $\log \frac{\text{UB}}{\mathcal{V}_1(\theta_i \,|\, \boldsymbol{x})} \approx 4$ means that the UB is roughly 50 times larger than $\mathcal{V}_1(\theta_i \,|\, \boldsymbol{x})$. Comparatively, $\mathcal{V}_2(\theta_i \,|\, \boldsymbol{x})$ has a tighter UB which is approximately 10 times larger than itself. The UB serves as a useful hint on the *order of magnitude* of the variances. In Appendix D, we present tighter bounds which are more expensive to compute.

In the first three layers of the MLP, $\mathcal{V}_1(\theta_i \,|\, \boldsymbol{x})$ presents a smaller value than $\mathcal{V}_2(\theta_i \,|\, \boldsymbol{x})$, meaning that $\hat{\mathcal{I}}_1$ can more accurately estimate the diagonal FIM. Interestingly, this is not true for the last layer: $\mathcal{V}_2(\theta_i \,|\, \boldsymbol{x})$ becomes zero while $\hat{\mathcal{I}}_1$ presents the largest variance across all parameter groups. Due to this, one should always prefer $\hat{\mathcal{I}}_2$ over $\hat{\mathcal{I}}_1$ for the last layer. In the last two layers, $\hat{\mathcal{I}}_2$ is in simple closed form and, hence, does not need automatic differentiation to calculate (see Remarks 4.4 and 4.5). The shape of the variance curves are sensitive to the choices of activation functions $\phi$ and inputs $\boldsymbol{x}$. In general, the variance in the first few epochs presents more dynamics than the rest of the training process. If one uses log-sigmoid activations $\phi(t) = -\log(1 + \exp(-t))$ (which is equivalent to $\phi(t) = -\text{SoftPlus}(-t)$, as per Remark 4.4), the variances of $\hat{\mathcal{I}}_1$ and $\hat{\mathcal{I}}_2$ only appear in the randomly initialized NN and quickly vanish once training starts, as shown in Appendix F. In this case, the learner more easily approaches a nearly linear region of the loss landscape where local optima lie. In practice, one should estimate and examine the scale of variances — which should not be neglected as per Fig. 2 — before choosing a preferred diagonal FIM estimator.

## 6 Relationship with the "Empirical Fisher"

In some scenarios, even the estimators of the diagonal FIM $\hat{\mathcal{I}}_1(\boldsymbol{\theta})$ and $\hat{\mathcal{I}}_2(\boldsymbol{\theta})$ can be prohibitively expensive. Part of the cost comes from requiring label samples $\boldsymbol{y}_k$ for each $\boldsymbol{x}_k$, as per Eq. (3). For example, when the FIM is used in an iterative optimization procedure, $\boldsymbol{y}_k$'s need to be re-sampled at each learning step w.r.t. the current $\boldsymbol{h}$ alongside their backpropagation (accounting for sampling).

As such, alternative 'FIM-like' objects have been explored which replace the samples from $p(\boldsymbol{y} \,|\, \boldsymbol{x})$ with samples from an underlying true (but unknown) data distribution $q(\boldsymbol{y} \,|\, \boldsymbol{x})$ [20, 27]. We define the data's joint distribution as $q(\boldsymbol{x}, \boldsymbol{y}) \doteq q(\boldsymbol{x})q(\boldsymbol{y} \,|\, \boldsymbol{x})$. Analogous to the FIM, the *data Fisher information matrix* (DFIM) can be defined as the PSD tensor $\mathrm{I}(\boldsymbol{\theta}) \doteq \mathbb{E}_{q(\boldsymbol{x})}[\mathrm{I}(\boldsymbol{\theta} \,|\, \boldsymbol{x})]$, with

$$\mathrm{I}(\boldsymbol{\theta} \,|\, \boldsymbol{x}) = \underset{q(\hat{\mathbf{y}} \,|\, \boldsymbol{x})}{\mathbb{E}} \left[ \frac{\partial \log p(\hat{\mathbf{y}} \,|\, \boldsymbol{x})}{\partial \boldsymbol{\theta}} \frac{\partial \log p(\hat{\mathbf{y}} \,|\, \boldsymbol{x})}{\partial \boldsymbol{\theta}^\top} \right] = \left( \frac{\partial \boldsymbol{h}}{\partial \boldsymbol{\theta}} \right)^\top \mathrm{I}(\boldsymbol{h} \,|\, \boldsymbol{x}) \left( \frac{\partial \boldsymbol{h}}{\partial \boldsymbol{\theta}} \right), \qquad (17)$$

where $\mathrm{I}(\boldsymbol{h} \,|\, \boldsymbol{x}) = \mathbb{E}_{q(\hat{\mathbf{y}} \,|\, \boldsymbol{x})} \left[ (\boldsymbol{t}(\hat{\mathbf{y}}) - \boldsymbol{\eta}(\boldsymbol{x}))(\boldsymbol{t}(\hat{\mathbf{y}}) - \boldsymbol{\eta}(\boldsymbol{x}))^\top \right]$ denotes the 2nd (non-central) moment of $(\boldsymbol{t}(\hat{\mathbf{y}}) - \boldsymbol{\eta}(\boldsymbol{x}))$ w.r.t. $q(\hat{\mathbf{y}} \,|\, \boldsymbol{x})$, and $\partial \boldsymbol{h}/\partial \boldsymbol{\theta}$ is the Jacobian of the map $\boldsymbol{\theta} \to \boldsymbol{h}$. In the special case that $q(\boldsymbol{y} \,|\, \boldsymbol{x}) = p(\boldsymbol{y} \,|\, \boldsymbol{x}; \boldsymbol{\theta})$, then $\mathrm{I}(\boldsymbol{\theta} \,|\, \boldsymbol{x})$ becomes exactly $\mathcal{I}(\boldsymbol{\theta} \,|\, \boldsymbol{x})$.

The DFIM $\mathrm{I}(\boldsymbol{\theta} \,|\, \boldsymbol{x})$ in Eq. (17) is a more general definition. Compared to the FIM $\mathcal{I}(\boldsymbol{\theta} \,|\, \boldsymbol{x})$, it yields a different PSD tensor on the $\boldsymbol{\theta}$ parameter space (the neuromanifold) depending on a dis-

tribution $q(\boldsymbol{x}, \boldsymbol{y})$, which is neither necessarily on the same neuromanifold nor necessarily parametric at all. The asymmetry in the true data distribution and the empirical one results in different geometric structures [5]. By definition, we have $\mathrm{I}(\boldsymbol{\theta} \,|\, \boldsymbol{x}) \succeq (\partial \mathrm{KL}/\partial \boldsymbol{\theta})(\partial \mathrm{KL}/\partial \boldsymbol{\theta})^\top$, where $\mathrm{KL}(\boldsymbol{\theta}) \doteq \int q(\hat{\boldsymbol{y}} \,|\, \boldsymbol{x}) \log \frac{q(\hat{\boldsymbol{y}} \,|\, \boldsymbol{x})}{p(\hat{\boldsymbol{y}} \,|\, \boldsymbol{x};\boldsymbol{\theta})} \, \mathrm{d}\hat{\boldsymbol{y}}$ is the Kullback-Leibler (KL) divergence, or the loss in a parameter learning scenario. The DFIM can be regarded as a surrogate function of the squared gradient of the KL divergence. It is a symmetric covariant tensor and satisfies the same rule w.r.t. reparameterization as the FIM. Consider the reparameterization $\boldsymbol{\theta} \to \boldsymbol{\zeta}$, the DFIM becomes $\mathrm{I}(\boldsymbol{\zeta} \,|\, \boldsymbol{x}) = (\partial \boldsymbol{\theta}/\partial \boldsymbol{\zeta})^\top \mathrm{I}(\boldsymbol{\theta} \,|\, \boldsymbol{x})(\partial \boldsymbol{\theta}/\partial \boldsymbol{\zeta})$.

Notice that $\hat{\boldsymbol{\eta}}(\boldsymbol{x}) \doteq \mathbb{E}_{q(\hat{\boldsymbol{y}} \,|\, \boldsymbol{x})}[\boldsymbol{t}(\hat{\boldsymbol{y}})] \neq \boldsymbol{\eta}(\boldsymbol{x})$ in general. As such, there will be a miss-match when utilizing $\mathrm{I}(\boldsymbol{h} \,|\, \boldsymbol{x})$ as a substitute for $\mathcal{I}(\boldsymbol{h} \,|\, \boldsymbol{x})$. However, as learning progresses and $p(\hat{\boldsymbol{y}} \,|\, \boldsymbol{x})$ becomes more similar to the data's true labeling posterior $q(\hat{\boldsymbol{y}} \,|\, \boldsymbol{x})$, the DFIM will become closer to the FIM.

If $q(\boldsymbol{x}, \boldsymbol{y}) = \frac{1}{N} \sum_{k=1}^N \delta(\boldsymbol{x} - \boldsymbol{x}_k) \cdot \delta(\boldsymbol{y} - \boldsymbol{y}_k)$ is defined by the observed samples, DFIM gives the widely used "*Empirical Fisher*" [25], whose diagonal entries are

$$\hat{\mathrm{I}}(\theta_i) = \frac{1}{N} \sum_{k=1}^N \left( \partial \boldsymbol{h}_i^a(\boldsymbol{x}_k) \cdot (\boldsymbol{t}_a(\hat{\boldsymbol{y}}_k) - \boldsymbol{\eta}_a(\boldsymbol{x}_k)) \right)^2,$$

where $(\boldsymbol{x}_1, \hat{\boldsymbol{y}}_1), \ldots, (\boldsymbol{x}_N, \hat{\boldsymbol{y}}_N)$ are i.i.d. sampled from $q(\boldsymbol{x}, \hat{\boldsymbol{y}})$. Similar to $\hat{\mathcal{I}}_1(\theta_i \,|\, \boldsymbol{x})$, an estimator with a fixed input $\boldsymbol{x}$ can be considered, denoted as $\hat{\mathrm{I}}(\theta_i \,|\, \boldsymbol{x})$.

Given the computational benefits of using the data directly — bypassing a separate sampling routine — many popular optimization methods employ the empirical Fisher or its approximation. For instance, the Adam optimizer [16] uses the empirical Fisher to approximate the diagonal FIM. However, switching from sampling $\boldsymbol{y}_k$ to $\hat{\boldsymbol{y}}_k$ is anything but superficial [25, Chapter 11] — $\hat{\mathrm{I}}(\boldsymbol{\theta})$ is *not* an unbiased estimator of $\mathcal{I}(\boldsymbol{\theta})$ as $\mathrm{I}(\boldsymbol{h} \,|\, \boldsymbol{x})$ is different from $\mathcal{I}(\boldsymbol{h} \,|\, \boldsymbol{x})$.

The biased nature of the empirical Fisher affects the other moments as well. In particular, we do not have the same equivalence of covariance and the metric being pulled back by $\boldsymbol{\theta} \to \boldsymbol{h}$ [38].

**Lemma 6.1.** *Given the conditional data distribution $q(\hat{\boldsymbol{y}} \,|\, \boldsymbol{x})$, the covariance of $\boldsymbol{t}$ given $\boldsymbol{x}$ is given by*

$$\mathrm{Cov}^q(\boldsymbol{t} \,|\, \boldsymbol{x}) = \mathrm{I}(\boldsymbol{h} \,|\, \boldsymbol{x}) - \Delta \mathsf{H}(\boldsymbol{x}), \tag{18}$$

*where $\Delta \mathsf{H}(\boldsymbol{x}) = (\boldsymbol{\eta}(\boldsymbol{x}) - \hat{\boldsymbol{\eta}}(\boldsymbol{x}))(\boldsymbol{\eta}(\boldsymbol{x}) - \hat{\boldsymbol{\eta}}(\boldsymbol{x}))^\top$.*

As a result, although the variance of the estimator $\hat{\mathrm{I}}(\theta_i \,|\, \boldsymbol{x})$ takes a similar form to $\mathcal{V}_1(\theta_i \,|\, \boldsymbol{x})$ (*i.e.*, Eq. (8)), its sufficient statistic terms do not exclusively consist of central moments. Noting the miss-match in $\hat{\boldsymbol{\eta}}(\boldsymbol{x}) \neq \boldsymbol{\eta}(\boldsymbol{x})$, Lemma 6.1 reveals an additional term which shifts $\mathrm{I}(\boldsymbol{h} \,|\, \boldsymbol{x})$ away from the 2nd central moment of $\boldsymbol{t}(\hat{\boldsymbol{y}})$ (w.r.t. $q(\hat{\boldsymbol{y}} \,|\, \boldsymbol{x})$). Instead, these sufficient statistic terms correspond to non-central moments of $\boldsymbol{t}(\hat{\boldsymbol{y}}) - \boldsymbol{\eta}(\boldsymbol{x})$. Some corresponding empirical Fisher / DFIM bounds are characterized in Appendix G.

# 7   Conclusion

We have analyzed two different estimators $\hat{\mathcal{I}}_1(\boldsymbol{\theta})$ and $\hat{\mathcal{I}}_2(\boldsymbol{\theta})$ for the diagonal entries of the FIM. The variances of these estimators are determined by both the non-linearly of the neural network and the moments of the exponential family. We have identified distinct scenarios on which estimator is preferable. For example, ReLU networks can only apply $\hat{\mathcal{I}}_1(\boldsymbol{\theta})$ due to a lack of smoothness. As another example, $\hat{\mathcal{I}}_2(\boldsymbol{\theta})$ has zero variance in the last layer and thus is always preferable than $\hat{\mathcal{I}}_1(\boldsymbol{\theta})$. Similarly, in the second last layer, $\hat{\mathcal{I}}_2(\boldsymbol{\theta})$ has a simple closed form and potentially preferable for neurons in their linear regions (see Remark 4.5). In general, one has to apply Theorem 4.1 based on their specific neural network and settings and choose the estimator with the smaller variance. Our results suggest that, from a variance perspective, uniformly utilizing one of the FIM estimators $\hat{\mathcal{I}}_j(\boldsymbol{\theta})$ is often suboptimal in NNs. Our work has further extended from analyzing the conditional FIM estimators $\hat{\mathcal{I}}_j(\boldsymbol{\theta} \,|\, \boldsymbol{x})$ to the joint FIM estimators $\hat{\mathcal{I}}_j(\boldsymbol{\theta})$; and we have examined the relationship between the investigated estimators and the empirical Fisher. Future directions include extending the analysis of the variance of FIM estimators to block diagonals (*e.g.* [26, 35]) and adapting current NN optimizers (*e.g.* [16]) to incorporate the variance of FIM estimators.

## Acknowledgments and Disclosure of Funding

The authors thank Frank Nielsen, James C. Spall, and the anonymous reviewers for their insightful feedback and many constructive comments.

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

# Supplementary Material

## Abstract

This is the Supplementary Material to Paper "Trade-Offs of Diagonal Fisher Information Matrix Estimators". To differentiate with the numberings in the main file, the numbering of Theorems is letter-based (A, B, ...).

## Table of Contents

# A    Natural Gradient Toy Data Example

The following section describes the data and models of Fig. 1. In general, the toy data and models constructed consists of taking 1D output setting presented by Section 5, where the NN $h_{\boldsymbol{\theta}}(\boldsymbol{x})$ is a linear function.

## A.I    Data

The 2D input data $\boldsymbol{x} \in \Re^2$ is sampled from a simple isotropic centered Gaussian $\mathcal{N}(\boldsymbol{0}, \boldsymbol{I})$. A linear response variable $a \in \Re$ is defined by the following:

$$a = \boldsymbol{\theta}_{\mathrm{true}}^{\top} \boldsymbol{x}; \quad \text{where } \boldsymbol{\theta}_{\mathrm{true}} = (1, 1).$$

The outputs of $y$ for the cases of regression and classification are differentiated by how $a$ is used in sampling:

$$y_{\mathrm{regression}} \sim \mathcal{N}(\mu = 1, \sigma = 1)$$
$$y_{\mathrm{classification}} \sim \mathrm{Bern}(p = \sigma(a)),$$

where $\sigma(z) = (1 + \exp(-z))$ is the logistic function.

## A.II    Model

The model $h_{\boldsymbol{\theta}}(\boldsymbol{x}) = \boldsymbol{\theta}^{\top} \boldsymbol{x}$ consists of a linear function; and the exponential family Eq. (1) is chosen to be a 1D isotropic Gaussian and binary multinomial distribution (Bernoulli) for regression and classification, respectively. This corresponds to Section 5 for 1D outputs. Notice that the model exactly matches the data generating function.

## A.III    Training

Natural gradient descent (NGD) is taken using both $\hat{\mathcal{I}}_1(\boldsymbol{\theta})$ and $\hat{\mathcal{I}}_2(\boldsymbol{\theta})$. The estimated FIM utilize only a single $y \mid \boldsymbol{x}$ sample for each input $\boldsymbol{x}$. We use a learning rate of $\eta = 0.01$ over 256 epochs. A training set of 256 data points are sampled. At each iteration of NGD, we sample 4 random points from the training set for the update. The test loss is evaluated on a test set of 4096 data points sampled.

## A.IV    Variance Plot of Example

Larger version of Fig. 1 with additional variance sum plotted over time is given by Fig. I. Note that variance sum is including off diagonals. Further note that the variance is calculated over joint sample in $(\boldsymbol{x}, y)$.

## A.V    Other Seeds

We further present other random seed of the teaser plot in Figs. II to IV.

# B    The Conditional Variances in Closed Form

We consider the diagonal entries of the conditional FIM $\mathcal{I}(\theta_i \mid \boldsymbol{x})$ and the conditional variances $\mathcal{V}_j(\theta_i \mid \boldsymbol{x})$ of its estimators in closed form.

*Proof of Lemma 3.1.*  The proof directly follows from [37, Equation 6], [37, Theorem 4], and [37, Theorem 6]. In what follows, we provide a proof of the Lemma utilizing the notation of this paper for completeness. We prove the statement one equation at a time.

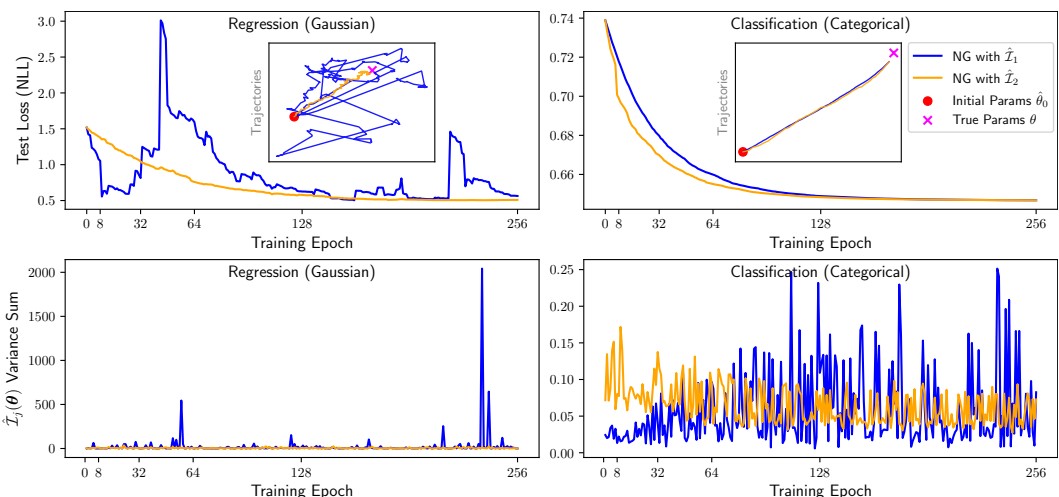

Figure I: Extended version of Fig. I with the sum of variance of FIM estimators over epochs.

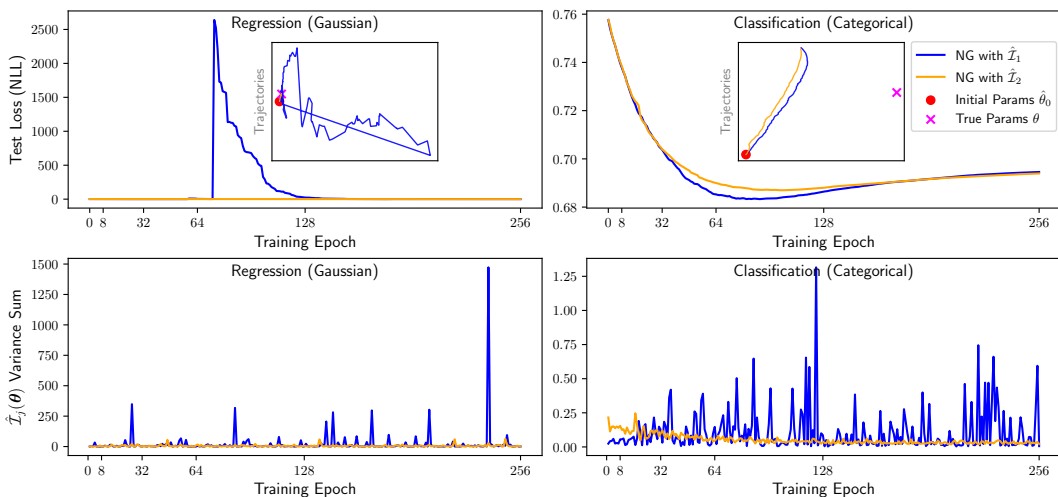

Figure II: Fig. I over different randomizations (a).

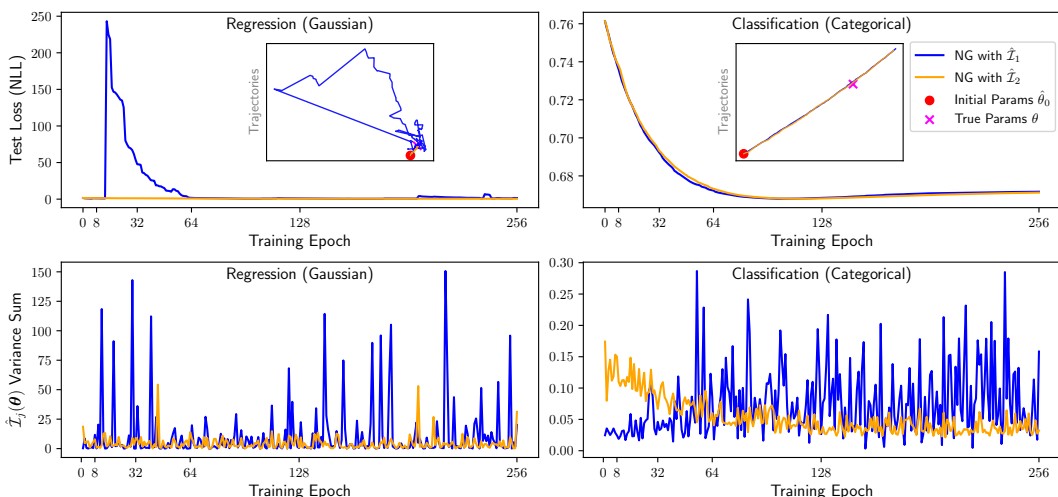

Figure III: Fig. I over different randomizations (b).

For Eq. (4), we consider the following computation.

$$\mathcal{I}(\theta_i \mid \boldsymbol{x})$$

$$= \mathop{\mathbb{E}}_{p(\boldsymbol{y} \mid \boldsymbol{x};\boldsymbol{\theta})} \left[ \frac{\partial \log p(\boldsymbol{y} \mid \boldsymbol{x};\boldsymbol{\theta})}{\partial \boldsymbol{\theta}} \frac{\partial \log p(\boldsymbol{y} \mid \boldsymbol{x};\boldsymbol{\theta})}{\partial \boldsymbol{\theta}^\top} \right]$$

$$= \mathop{\mathbb{E}}_{p(\boldsymbol{y} \mid \boldsymbol{x};\boldsymbol{\theta})} \left[ \frac{\partial \left( \boldsymbol{t}^\top(\boldsymbol{y})\boldsymbol{h}_{\boldsymbol{\theta}}(\boldsymbol{x}) - F(\boldsymbol{h}_{\boldsymbol{\theta}}(\boldsymbol{x})) \right)}{\partial \boldsymbol{\theta}} \frac{\partial \left( \boldsymbol{t}^\top(\boldsymbol{y})\boldsymbol{h}_{\boldsymbol{\theta}}(\boldsymbol{x}) - F(\boldsymbol{h}_{\boldsymbol{\theta}}(\boldsymbol{x})) \right)}{\partial \boldsymbol{\theta}^\top} \right]$$

$$= \mathop{\mathbb{E}}_{p(\boldsymbol{y} \mid \boldsymbol{x};\boldsymbol{\theta})} \left[ \left( \frac{\partial \boldsymbol{h}_{\boldsymbol{\theta}}(\boldsymbol{x})}{\partial \boldsymbol{\theta}} \right)^\top \left( \boldsymbol{t}(\boldsymbol{y}) - \frac{\partial F(\boldsymbol{h})}{\partial \boldsymbol{h}} \bigg|_{\boldsymbol{h}=\boldsymbol{h}_{\boldsymbol{\theta}}(\boldsymbol{x})} \right) \left( \boldsymbol{t}(\boldsymbol{y}) - \frac{\partial F(\boldsymbol{h})}{\partial \boldsymbol{h}} \bigg|_{\boldsymbol{h}=\boldsymbol{h}_{\boldsymbol{\theta}}(\boldsymbol{x})} \right)^\top \left( \frac{\partial \boldsymbol{h}_{\boldsymbol{\theta}}(\boldsymbol{x})}{\partial \boldsymbol{\theta}^\top} \right) \right]$$

$$= \mathop{\mathbb{E}}_{p(\boldsymbol{y} \mid \boldsymbol{x};\boldsymbol{\theta})} \left[ \left( \frac{\partial \boldsymbol{h}_{\boldsymbol{\theta}}(\boldsymbol{x})}{\partial \boldsymbol{\theta}} \right)^\top (\boldsymbol{t}(\boldsymbol{y}) - \boldsymbol{\eta}(\boldsymbol{x})) (\boldsymbol{t}(\boldsymbol{y}) - \boldsymbol{\eta}(\boldsymbol{x}))^\top \left( \frac{\partial \boldsymbol{h}_{\boldsymbol{\theta}}(\boldsymbol{x})}{\partial \boldsymbol{\theta}^\top} \right) \right]$$

$$= \left( \frac{\partial \boldsymbol{h}_{\boldsymbol{\theta}}(\boldsymbol{x})}{\partial \boldsymbol{\theta}} \right)^\top \left( \mathop{\mathbb{E}}_{p(\boldsymbol{y} \mid \boldsymbol{x};\boldsymbol{\theta})} \left[ (\boldsymbol{t}(\boldsymbol{y}) - \boldsymbol{\eta}(\boldsymbol{x})) (\boldsymbol{t}(\boldsymbol{y}) - \boldsymbol{\eta}(\boldsymbol{x}))^\top \right] \right) \left( \frac{\partial \boldsymbol{h}_{\boldsymbol{\theta}}(\boldsymbol{x})}{\partial \boldsymbol{\theta}^\top} \right)$$

$$= \left( \frac{\partial \boldsymbol{h}_{\boldsymbol{\theta}}(\boldsymbol{x})}{\partial \boldsymbol{\theta}} \right)^\top \mathcal{I}(\boldsymbol{h} \mid \boldsymbol{x}) \left( \frac{\partial \boldsymbol{h}_{\boldsymbol{\theta}}(\boldsymbol{x})}{\partial \boldsymbol{\theta}^\top} \right) .$$

Using Einstein notation and restricting the partial derivative to a component of $\boldsymbol{\theta}$ yields the desired result.

For Eq. (5), we shorthand $\boldsymbol{\delta}(\boldsymbol{y}) = \boldsymbol{t}(\boldsymbol{y}) - \boldsymbol{\eta}(\boldsymbol{x})$. Note that the $\hat{\mathcal{I}}_1(\theta_i \mid \boldsymbol{x})$ estimator can be written as follows:

$$\hat{\mathcal{I}}_1(\theta_i \mid \boldsymbol{x}) = \frac{1}{N} \sum_{k=1}^{N} \left( \frac{\partial F(\boldsymbol{h}(\boldsymbol{x}))}{\partial \theta_i} - \frac{\partial \boldsymbol{h}^a(\boldsymbol{x})}{\partial \theta_i} \cdot \boldsymbol{t}_a(\boldsymbol{y}_k) \right)^2$$

$$= \frac{1}{N} \sum_{k=1}^{N} \left( \frac{\partial \boldsymbol{h}^a(\boldsymbol{x})}{\partial \theta_i} \cdot \boldsymbol{\eta}_a(\boldsymbol{x}) - \frac{\partial \boldsymbol{h}^a(\boldsymbol{x})}{\partial \theta_i} \cdot \boldsymbol{t}_a(\boldsymbol{y}_k) \right)^2$$

$$= \frac{1}{N} \sum_{k=1}^{N} \partial \boldsymbol{h}^a(\boldsymbol{x}) \partial \boldsymbol{h}^b(\boldsymbol{x}) \boldsymbol{\delta}_a(\boldsymbol{y}_k) \boldsymbol{\delta}_b(\boldsymbol{y}_k).$$

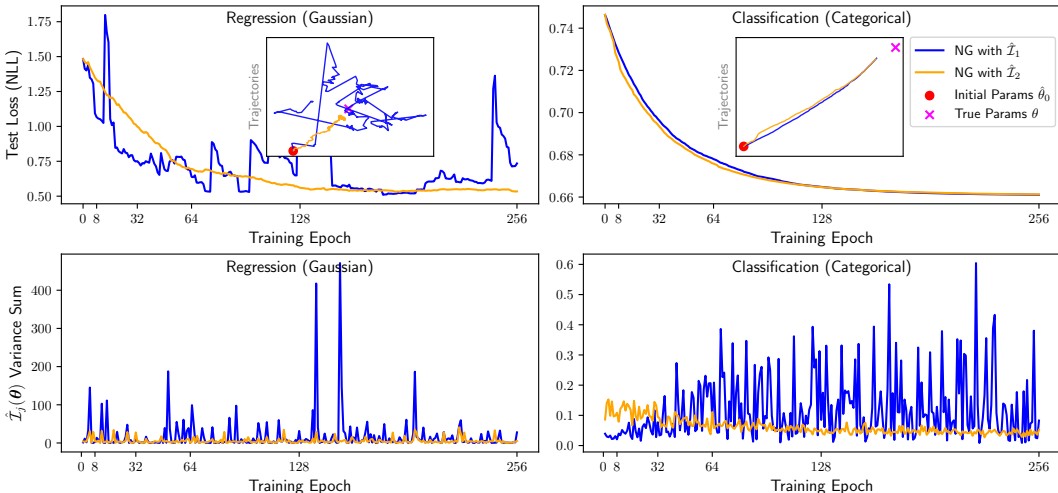

Figure IV: Fig. I over different randomizations (c).

Thus, we have

$$\mathcal{V}_1(\theta_i \mid \boldsymbol{x})$$

$$= \text{Var}\left(\hat{\mathcal{I}}_1(\theta_i \mid \boldsymbol{x})\right)$$

$$= \text{Var}\left(\frac{1}{N}\sum_{k=1}^{N}\partial\boldsymbol{h}^a(\boldsymbol{x})\partial\boldsymbol{h}^b(\boldsymbol{x})\boldsymbol{\delta}_a(\boldsymbol{y}_k)\boldsymbol{\delta}_b(\boldsymbol{y}_k)\right)$$

$$= \frac{1}{N}\cdot\text{Var}\left(\partial\boldsymbol{h}^a(\boldsymbol{x})\partial\boldsymbol{h}^b(\boldsymbol{x})\boldsymbol{\delta}_a(\boldsymbol{y}_k)\boldsymbol{\delta}_b(\boldsymbol{y}_k)\right)$$

$$= \frac{1}{N}\cdot\left(\mathop{\mathbb{E}}_{p(\boldsymbol{y}\mid\boldsymbol{x};\boldsymbol{\theta})}\left[\left(\partial\boldsymbol{h}^a(\boldsymbol{x})\partial\boldsymbol{h}^b(\boldsymbol{x})\boldsymbol{\delta}_a(\boldsymbol{y})\boldsymbol{\delta}_b(\boldsymbol{y})\right)^2\right] - \mathop{\mathbb{E}}_{p(\boldsymbol{y}\mid\boldsymbol{x};\boldsymbol{\theta})}\left[\partial\boldsymbol{h}^a(\boldsymbol{x})\partial\boldsymbol{h}^b(\boldsymbol{x})\boldsymbol{\delta}_a(\boldsymbol{y})\boldsymbol{\delta}_b(\boldsymbol{y})\right]^2\right).$$

Let us compute each of these terms.

$$\mathop{\mathbb{E}}_{p(\boldsymbol{y}\mid\boldsymbol{x};\boldsymbol{\theta})}\left[\left(\partial\boldsymbol{h}^a(\boldsymbol{x})\partial\boldsymbol{h}^b(\boldsymbol{x})\boldsymbol{\delta}_a(\boldsymbol{y})\boldsymbol{\delta}_b(\boldsymbol{y})\right)^2\right]$$

$$= \mathop{\mathbb{E}}_{p(\boldsymbol{y}\mid\boldsymbol{x};\boldsymbol{\theta})}\left[\partial\boldsymbol{h}^a(\boldsymbol{x})\partial\boldsymbol{h}^b(\boldsymbol{x})\partial\boldsymbol{h}^c(\boldsymbol{x})\partial\boldsymbol{h}^d(\boldsymbol{x})\boldsymbol{\delta}_a(\boldsymbol{y})\boldsymbol{\delta}_b(\boldsymbol{y})\boldsymbol{\delta}_c(\boldsymbol{y})\boldsymbol{\delta}_d(\boldsymbol{y})\right]$$

$$= \partial\boldsymbol{h}^a(\boldsymbol{x})\partial\boldsymbol{h}^b(\boldsymbol{x})\partial\boldsymbol{h}^c(\boldsymbol{x})\partial\boldsymbol{h}^d(\boldsymbol{x})\mathop{\mathbb{E}}_{p(\boldsymbol{y}\mid\boldsymbol{x};\boldsymbol{\theta})}\left[\boldsymbol{\delta}_a(\boldsymbol{y})\boldsymbol{\delta}_b(\boldsymbol{y})\boldsymbol{\delta}_c(\boldsymbol{y})\boldsymbol{\delta}_d(\boldsymbol{y})\right]$$

$$= \partial\boldsymbol{h}^a(\boldsymbol{x})\partial\boldsymbol{h}^b(\boldsymbol{x})\partial\boldsymbol{h}^c(\boldsymbol{x})\partial\boldsymbol{h}^d(\boldsymbol{x})\mathcal{K}^p_{abcd}(\boldsymbol{t}\mid\boldsymbol{x}).$$

And,

$$\left(\mathop{\mathbb{E}}_{p(\boldsymbol{y}\mid\boldsymbol{x};\boldsymbol{\theta})}\left[\partial\boldsymbol{h}^a(\boldsymbol{x})\partial\boldsymbol{h}^b(\boldsymbol{x})\boldsymbol{\delta}_a(\boldsymbol{y})\boldsymbol{\delta}_b(\boldsymbol{y})\right]\right)^2$$

$$= \left(\partial\boldsymbol{h}^a(\boldsymbol{x})\partial\boldsymbol{h}^b(\boldsymbol{x})\mathop{\mathbb{E}}_{p(\boldsymbol{y}\mid\boldsymbol{x};\boldsymbol{\theta})}\left[\boldsymbol{\delta}_a(\boldsymbol{y})\boldsymbol{\delta}_b(\boldsymbol{y})\right]\right)^2$$

$$= \left(\partial\boldsymbol{h}^a(\boldsymbol{x})\partial\boldsymbol{h}^b(\boldsymbol{x})\mathcal{I}_{ab}(\boldsymbol{h}\mid\boldsymbol{x})\right)^2$$

$$= \partial\boldsymbol{h}^a(\boldsymbol{x})\partial\boldsymbol{h}^b(\boldsymbol{x})\partial\boldsymbol{h}^c(\boldsymbol{x})\partial\boldsymbol{h}^d(\boldsymbol{x})\mathcal{I}_{ab}(\boldsymbol{h}\mid\boldsymbol{x})\mathcal{I}_{cd}(\boldsymbol{h}\mid\boldsymbol{x})$$

$$= \partial\boldsymbol{h}^a(\boldsymbol{x})\partial\boldsymbol{h}^b(\boldsymbol{x})\partial\boldsymbol{h}^c(\boldsymbol{x})\partial\boldsymbol{h}^d(\boldsymbol{x})\left(\mathcal{I}(\boldsymbol{h}\mid\boldsymbol{x})\otimes\mathcal{I}(\boldsymbol{h}\mid\boldsymbol{x})\right)_{abcd}$$

Simplifying all term yields the result as required.

Finally, for Eq. (6) we consider the following simplification of the estimator.

$$\hat{\mathcal{I}}_2(\theta_i \mid \boldsymbol{x}) = \frac{1}{N}\sum_{k=1}^{N}\left(\frac{\partial^2 F(\boldsymbol{h}(\boldsymbol{x}))}{\partial^2\theta_i} - \frac{\partial^2\boldsymbol{h}^a(\boldsymbol{x})}{\partial^2\theta_i}\cdot\boldsymbol{t}_a(\boldsymbol{y}_k)\right)$$

$$= \frac{1}{N}\sum_{k=1}^{N}\left(\frac{\partial}{\partial\theta_i}\left(\frac{\partial\boldsymbol{h}^a(\boldsymbol{x})}{\partial\theta_i}\cdot\boldsymbol{\eta}_a(\boldsymbol{x})\right) - \frac{\partial^2\boldsymbol{h}^a(\boldsymbol{x})}{\partial^2\theta_i}\cdot\boldsymbol{t}_a(\boldsymbol{y}_k)\right)$$

$$= \frac{1}{N}\sum_{k=1}^{n}\left(\frac{\partial\boldsymbol{h}^a(\boldsymbol{x})}{\partial\theta_i}\cdot\frac{\partial\boldsymbol{\eta}_a(\boldsymbol{x})}{\partial\theta_i} + \frac{\partial^2\boldsymbol{h}^a(\boldsymbol{x})}{\partial^2\theta_i}\cdot\boldsymbol{\eta}_a(\boldsymbol{x}) - \frac{\partial^2\boldsymbol{h}^a(\boldsymbol{x})}{\partial^2\theta_i}\cdot\boldsymbol{t}_a(\boldsymbol{y}_k)\right)$$

$$= \frac{1}{N}\sum_{k=1}^{n}\left(\partial_i\boldsymbol{h}^a(\boldsymbol{x})\cdot\frac{\partial\boldsymbol{\eta}_a(\boldsymbol{x})}{\partial\theta_i} - \partial_i^2\boldsymbol{h}^a(\boldsymbol{x})\cdot\boldsymbol{\delta}_a(\boldsymbol{y}_k)\right)$$

$$= \frac{1}{N}\sum_{k=1}^{n}\left(\partial_i\boldsymbol{h}^a(\boldsymbol{x})\cdot\partial_i\boldsymbol{h}^b(\boldsymbol{x})\cdot\mathcal{I}_{ab}(\boldsymbol{h}\mid\boldsymbol{x}) - \partial_i^2\boldsymbol{h}^a(\boldsymbol{x})\cdot\boldsymbol{\delta}_a(\boldsymbol{y}_k)\right)$$

$$= \partial_i\boldsymbol{h}^a(\boldsymbol{x})\cdot\partial_i\boldsymbol{h}^b(\boldsymbol{x})\cdot\mathcal{I}_{ab}(\boldsymbol{h}\mid\boldsymbol{x}) - \frac{1}{N}\sum_{k=1}^{n}\left(\partial_i^2\boldsymbol{h}^a(\boldsymbol{x})\cdot\boldsymbol{\delta}_a(\boldsymbol{y}_k)\right),$$

where the last line follows from [37, Lemma 2] (a result of $p(\boldsymbol{y}\mid\boldsymbol{x};\boldsymbol{\theta})$ following an exponential family, see [1]).

Notice that the first quantity is a constant w.r.t. the randomness of $\boldsymbol{y}_k$. As such, we can simplify the variance calculation as follows.

$$\mathcal{V}_2(\theta_i \,|\, \boldsymbol{x}) = \mathrm{Var}\left(\hat{\mathcal{I}}_2(\theta_i \,|\, \boldsymbol{x})\right)$$

$$= \mathrm{Var}\left(\partial_i \boldsymbol{h}^a(\boldsymbol{x}) \cdot \partial_i \boldsymbol{h}^b(\boldsymbol{x}) \cdot \mathcal{I}_{ab}(\boldsymbol{h} \,|\, \boldsymbol{x}) - \frac{1}{N}\sum_{k=1}^{n}\left(\partial_i^2 \boldsymbol{h}^a(\boldsymbol{x}) \cdot \boldsymbol{\delta}_a(\boldsymbol{y}_k)\right)\right)$$

$$= \mathrm{Var}\left(\frac{1}{N}\sum_{k=1}^{n}\left(\partial_i^2 \boldsymbol{h}^a(\boldsymbol{x}) \cdot \boldsymbol{\delta}_a(\boldsymbol{y}_k)\right)\right)$$

$$= \frac{1}{N}\mathrm{Var}\left(\partial_i^2 \boldsymbol{h}^a(\boldsymbol{x}) \cdot \boldsymbol{\delta}_a(\boldsymbol{y})\right)$$

$$= \frac{1}{N} \cdot \left(\mathbb{E}_{p(\boldsymbol{y} \,|\, \boldsymbol{x};\boldsymbol{\theta})}\left[\left(\partial_i^2 \boldsymbol{h}^a(\boldsymbol{x}) \cdot \boldsymbol{\delta}_a(\boldsymbol{y})\right)^2\right] - \mathbb{E}_{p(\boldsymbol{y} \,|\, \boldsymbol{x};\boldsymbol{\theta})}\left[\partial_i^2 \boldsymbol{h}^a(\boldsymbol{x}) \cdot \boldsymbol{\delta}_a(\boldsymbol{y})\right]^2\right)$$

$$= \frac{1}{N} \cdot \left(\mathbb{E}_{p(\boldsymbol{y} \,|\, \boldsymbol{x};\boldsymbol{\theta})}\left[\partial_i^2 \boldsymbol{h}^a(\boldsymbol{x}) \cdot \partial_i^2 \boldsymbol{h}^b(\boldsymbol{x}) \cdot \boldsymbol{\delta}_a(\boldsymbol{y}) \cdot \boldsymbol{\delta}_b(\boldsymbol{y})\right] - \partial_i^2 \boldsymbol{h}^a(\boldsymbol{x}) \cdot \mathbb{E}_{p(\boldsymbol{y} \,|\, \boldsymbol{x};\boldsymbol{\theta})}\left[\boldsymbol{\delta}_a(\boldsymbol{y})\right]^2\right)$$

$$= \frac{1}{N} \cdot \left(\partial_i^2 \boldsymbol{h}^a(\boldsymbol{x}) \cdot \partial_i^2 \boldsymbol{h}^b(\boldsymbol{x}) \cdot \mathbb{E}_{p(\boldsymbol{y} \,|\, \boldsymbol{x};\boldsymbol{\theta})}\left[\boldsymbol{\delta}_a(\boldsymbol{y}) \cdot \boldsymbol{\delta}_b(\boldsymbol{y})\right]\right)$$

$$= \frac{1}{N} \cdot \partial_i^2 \boldsymbol{h}^a(\boldsymbol{x}) \cdot \partial_i^2 \boldsymbol{h}^b(\boldsymbol{x}) \cdot \mathcal{I}_{ab}(\boldsymbol{h} \,|\, \boldsymbol{x}),$$

where the second last line follows from the fact that $\boldsymbol{\eta}(\boldsymbol{x}) = \mathbb{E}_{p(\boldsymbol{y} \,|\, \boldsymbol{x};\boldsymbol{\theta})}[\boldsymbol{t}(\boldsymbol{y})]$ and thus $\mathbb{E}_{p(\boldsymbol{y} \,|\, \boldsymbol{x};\boldsymbol{\theta})}[\boldsymbol{\delta}(\boldsymbol{y})] = 0$. This yields the desired result. $\qquad\square$

Lemma 3.1 shows that, for the former, $\mathcal{V}_1(\theta_i \,|\, \boldsymbol{x})$ only depends on 1st order derivatives; while $\mathcal{V}_2(\theta_i \,|\, \boldsymbol{x})$ only depends on the 2nd order derivatives. For the latter, $\mathcal{V}_1(\theta_i \,|\, \boldsymbol{x})$ depends on both the 2nd and 4th central moments of $\boldsymbol{t}(\boldsymbol{y})$; while $\mathcal{V}_2(\theta_i \,|\, \boldsymbol{x})$ only depends on the 2nd central moments.

Given $\mathcal{I}_{ab}(\boldsymbol{h} \,|\, \boldsymbol{x})$ and $\partial_i \boldsymbol{h}^a(\boldsymbol{x})$, the computational complexity of all diagonal entries $\mathcal{I}(\theta_i \,|\, \boldsymbol{x})$ is $\mathcal{O}(T^2 \dim(\boldsymbol{\theta}))$. If $\mathcal{K}^p_{abcd}(\boldsymbol{t} \,|\, \boldsymbol{x})$ and $\partial_i^2 \boldsymbol{h}^a(\boldsymbol{x})$ are given, then the computational complexity of the variances in Eqs. (5) and (6) is respectively $\mathcal{O}(T^4 \dim(\boldsymbol{\theta}))$ and $\mathcal{O}(T^2 \dim(\boldsymbol{\theta}))$. Each requires to evaluate a $T \times \dim(\boldsymbol{\theta})$ matrix, either $\partial_i \boldsymbol{h}^a(\boldsymbol{x})$ or $\partial_i^2 \boldsymbol{h}^a(\boldsymbol{x})$ — which can be expensive to calculate for the latter. This is why we need efficient estimators and / or bounds for the tensors on the LHS of Eqs. (4) to (6).

## C  Off-Diagonal Variance

We consider an off-diagonal version of the bound given by Theorem 4.1. Notice that in terms of the dependence on neural network weights, the only change is splitting the "responsibility" of the $i$'th and $j$'th parameter norms.

**Theorem C.1.** $\forall \boldsymbol{x} \in \Re^I$,

$$\mathrm{Var}\left(\hat{\mathcal{I}}_1(\boldsymbol{\theta} \,|\, \boldsymbol{x})_{ij}\right) \leq \frac{1}{N} \cdot \|\partial_i \boldsymbol{h}(\boldsymbol{x})\|_2^2 \cdot \|\partial_j \boldsymbol{h}(\boldsymbol{x})\|_2^2 \cdot \tilde{\gamma}_{\max}\left(\mathcal{M}\right), \tag{19}$$

$$\mathrm{Var}\left(\hat{\mathcal{I}}_2(\boldsymbol{\theta} \,|\, \boldsymbol{x})_{ij}\right) \leq \frac{1}{N} \cdot \|\partial_{ij}^2 \boldsymbol{h}(\boldsymbol{x})\|_2^2 \cdot \gamma_{\max}(\mathcal{I}(\boldsymbol{h} \,|\, \boldsymbol{x})), \tag{20}$$

*where*

$$\tilde{\gamma}_{\max}\left(\mathcal{M}\right) = \sup_{\boldsymbol{u}:\|\boldsymbol{u}\|_2=1, \boldsymbol{v}:\|\boldsymbol{v}\|_2=1} \boldsymbol{u}^a \boldsymbol{v}^b \boldsymbol{u}^c \boldsymbol{v}^d \mathcal{M}_{abcd}$$

$$\gamma_{\max}(M) = \sup_{\boldsymbol{u}:\|\boldsymbol{u}\|_2=1, \boldsymbol{v}:\|\boldsymbol{v}\|_2=1} \boldsymbol{u}^a \boldsymbol{v}^b M_{ab}.$$

*Proof.* The proof follows similarly to Appendices J and K, where the primary difference is just swapping the regular eigenvalue-like quantities with the $\gamma$ variational forms. $\qquad\square$

It should be noted that the corresponding lower bounds become trivial as the additional degree of freedom of having an $\inf$ over both $\boldsymbol{u}$ and $\boldsymbol{v}$ causes the corresponding $\gamma_{\min}$ definition to have negative quantities. Although it is unclear what the "tensor-like" variational quantity $\tilde{\gamma}_{\max}(\mathcal{M})$ will be, for a matrix, we have the following equivalence.

**Lemma C.2.** $\gamma_{\max}(A) = s_{\max}(A)$, where $s_{\max}(A)$ is the maximum singular value of $A$.

*Proof.* The proof follows from optimizing over $\boldsymbol{u}$ and $\boldsymbol{v}$ separately:

$$
\begin{aligned}
\gamma_{\max}(A) &= \sup_{\boldsymbol{u}:\|\boldsymbol{u}\|_2=1} \sup_{\boldsymbol{v}:\|\boldsymbol{v}\|_2=1} \boldsymbol{u}^a \boldsymbol{v}^b A_{ab} \\
&= \sup_{\boldsymbol{u}:\|\boldsymbol{u}\|_2=1} \sup_{\boldsymbol{v}:\|\boldsymbol{v}\|_2=1} \boldsymbol{u}^\top A \boldsymbol{v} \\
&= \sup_{\boldsymbol{v}:\|\boldsymbol{v}\|_2=1} \frac{(A\boldsymbol{v})^\top A\boldsymbol{v}}{\|A\boldsymbol{v}\|_2} \\
&= \sup_{\boldsymbol{v}:\|\boldsymbol{v}\|_2=1} \sqrt{\boldsymbol{v}^T (A^\top A)\boldsymbol{v}}.
\end{aligned}
$$

This is equivalent to the square root of the maximal eigenvalue of $A^\top A$, which is exactly the maximum singular value. $\qquad\square$

Hence for the $\hat{\mathcal{I}}_2$ we have the following.

**Corollary C.3.** $\forall \boldsymbol{x} \in \Re^I$,

$$
\operatorname{Var}\left(\hat{\mathcal{I}}_2(\boldsymbol{\theta} \mid \boldsymbol{x})_{ij}\right) \leq \frac{1}{N} \cdot \|\partial_{ij}^2 \boldsymbol{h}(\boldsymbol{x})\|_2^2 \cdot s_{\max}(\mathcal{I}(\boldsymbol{h} \mid \boldsymbol{x})). \tag{21}
$$

# D  Bounding the Trace Variance by Full Spectrum

**Theorem D.1.** *For any* $\boldsymbol{x} \in \Re^I$,

$$
\sum_{t=1}^{T} s_t^2(\partial \boldsymbol{h}(\boldsymbol{x})) \cdot \lambda_{T-t+1}\left(\mathcal{I}(\boldsymbol{h} \mid \boldsymbol{x})\right) \leq \operatorname{tr}\left(\mathcal{I}(\boldsymbol{\theta} \mid \boldsymbol{x})\right)
$$

$$
\leq \sum_{t=1}^{T} s_t^2(\partial \boldsymbol{h}(\boldsymbol{x})) \cdot \lambda_t\left(\mathcal{I}(\boldsymbol{h} \mid \boldsymbol{x})\right), \tag{22}
$$

$$
\frac{1}{N} \cdot \sum_{t=1}^{T} s_t^2\left(\operatorname{vJac}(\boldsymbol{h} \mid \boldsymbol{x})\right) \cdot \lambda_{T-t+1}\left(\overline{\mathcal{M}}\right) \leq \mathcal{V}_1(\boldsymbol{\theta} \mid \boldsymbol{x})
$$

$$
\frac{1}{N} \cdot \sum_{t=1}^{T} s_t^2\left(\operatorname{vJac}(\boldsymbol{h} \mid \boldsymbol{x})\right) \cdot \lambda_t\left(\overline{\mathcal{M}}\right), \tag{23}
$$

$$
\frac{1}{N} \cdot \sum_{t=1}^{T} s_t^2(\operatorname{dHes}(\boldsymbol{h} \mid \boldsymbol{x})) \cdot \lambda_{T-t+1}\left(\mathcal{I}(\boldsymbol{h} \mid \boldsymbol{x})\right) \leq \mathcal{V}_2(\boldsymbol{\theta} \mid \boldsymbol{x})
$$

$$
\leq \frac{1}{N} \cdot \sum_{t=1}^{T} s_t^2(\operatorname{dHes}(\boldsymbol{h} \mid \boldsymbol{x})) \cdot \lambda_t\left(\mathcal{I}(\boldsymbol{h} \mid \boldsymbol{x})\right), \tag{24}
$$

*where* $s_i^2(A) = \lambda_i(A^\top A)$ *denotes the $i$-th singular values,* $\overline{\mathcal{M}}$ *is the "reshaped" matrix of* $\mathcal{M}$ *defined in Theorem 4.1 — i.e. there exists $j, k$ such that* $\overline{\mathcal{M}}_{jk} = \mathcal{M}_{abcd}$ *for all $a, b, c, d$,*

$$
\operatorname{dHes}(\boldsymbol{h} \mid \boldsymbol{x}) = (\operatorname{diag}(\operatorname{Hes}(\boldsymbol{h}_1 \mid \boldsymbol{x})), \dots, \operatorname{diag}(\operatorname{Hes}(\boldsymbol{h}_T \mid \boldsymbol{x}))),
$$

*and*

$$
\operatorname{vJac}(\boldsymbol{h} \mid \boldsymbol{x}) = (\operatorname{vec}(\partial_1 \boldsymbol{h}(\boldsymbol{x}) \partial_1 \boldsymbol{h}(\boldsymbol{x})^\top), \dots, \operatorname{vec}(\partial_T \boldsymbol{h}(\boldsymbol{x}) \partial_T \boldsymbol{h}(\boldsymbol{x})^\top)).
$$

*Proof.* The proof follows from a generalized Ruhe's trace inequality [24]:

**Theorem D.2.** *For $A, B \in \Re^{n \times n}$ Hermitian matrices, we have that*

$$\sum_{i=1}^{n} \lambda_i(A) \cdot \lambda_{n-i+1}(B) \leq \operatorname{tr}(AB) \leq \sum_{i=1}^{n} \lambda_i(A) \cdot \lambda_i(B).$$

We prove the result for each equations.

For readability, we let $J^{ia} = \partial_i h^a(\boldsymbol{x})$.

**For Eq. (22):**

One can notice that the trace of the FIM can exactly be expressed as the trace of two matrices.

$$\operatorname{tr}(\mathcal{I}(\boldsymbol{\theta} \,|\, \boldsymbol{x})) = \sum_{i=1}^{\dim(\boldsymbol{\theta})} \partial_i \boldsymbol{h}^a(\boldsymbol{x}) \partial_i \boldsymbol{h}^b(\boldsymbol{x}) \mathcal{I}_{ab}(\boldsymbol{h} \,|\, \boldsymbol{x})$$

$$= \mathcal{I}_{ab}(\boldsymbol{h} \,|\, \boldsymbol{x}) \sum_{i=1}^{\dim(\boldsymbol{\theta})} J^{ia} J^{ib}$$

$$= \mathcal{I}_{ab}(\boldsymbol{h} \,|\, \boldsymbol{x}) \sum_{i=1}^{\dim(\boldsymbol{\theta})} (J^{\top})^{ai} J^{ib}$$

$$= \mathcal{I}_{ab}(\boldsymbol{h} \,|\, \boldsymbol{x})(J^{\top} J)^{ab}$$

$$= \operatorname{tr}\left((J^{\top} J)\mathcal{I}(\boldsymbol{h} \,|\, \boldsymbol{x})\right).$$

Thus, noting that the eigenvalue of the "squared" matrix is the matrix's singular value $\lambda_t(J^{\top} J) = s_t^2(J)$, with Theorem D.2, we have that:

$$\sum_{t=1}^{T} s_t^2(\partial \boldsymbol{h}(\boldsymbol{x})) \cdot \lambda_{T-t+1}(\mathcal{I}(\boldsymbol{h} \,|\, \boldsymbol{x})) \leq \operatorname{tr}(\mathcal{I}(\boldsymbol{\theta} \,|\, \boldsymbol{x})) \leq \sum_{t=1}^{T} s_t^2(\partial \boldsymbol{h}(\boldsymbol{x})) \cdot \lambda_t(\mathcal{I}(\boldsymbol{h} \,|\, \boldsymbol{x})).$$

**For Eq. (23):**

Noting that $\mathcal{M}_{abcd} = \mathcal{K}_{abcd}^p(\boldsymbol{t} \,|\, \boldsymbol{x}) - \mathcal{I}_{ab}(\boldsymbol{h} \,|\, \boldsymbol{x}) \cdot \mathcal{I}_{cd}(\boldsymbol{h} \,|\, \boldsymbol{x})$. Furthermore, we have that

$$\operatorname{vJac}(\boldsymbol{h} \,|\, \boldsymbol{x}) = (\operatorname{vec}(\partial_1 \boldsymbol{h}(\boldsymbol{x}) \partial_1 \boldsymbol{h}(\boldsymbol{x})^{\top}), \dots, \operatorname{vec}(\partial_T \boldsymbol{h}(\boldsymbol{x}) \partial_T \boldsymbol{h}(\boldsymbol{x})^{\top})).$$

Let us define the following 3D tensor with $\mathcal{J}^{iab} = \partial_i \boldsymbol{h}^a(\boldsymbol{x}) \partial_i \boldsymbol{h}^b(\boldsymbol{x}) = (\partial_i \boldsymbol{h}(\boldsymbol{x}) \partial_i^{\top} \boldsymbol{h}(\boldsymbol{x}))^{ab}$.

$$\mathcal{V}_1(\boldsymbol{\theta} \,|\, \boldsymbol{x}) = \frac{1}{N} \sum_{i=1}^{\dim(\boldsymbol{\theta})} \partial_i \boldsymbol{h}^a(\boldsymbol{x}) \partial_i \boldsymbol{h}^b(\boldsymbol{x}) \partial_i \boldsymbol{h}^c(\boldsymbol{x}) \partial_i \boldsymbol{h}^d(\boldsymbol{x}) \mathcal{M}_{abcd}$$

$$= \frac{1}{N} \mathcal{M}_{abcd} \sum_{i=1}^{\dim(\boldsymbol{\theta})} \mathcal{J}^{iab} \mathcal{J}^{icd}$$

$$= \frac{1}{N} \sum_{a,b=1}^{T} \sum_{c,d=1}^{T} \mathcal{M}_{abcd} \sum_{i=1}^{\dim(\boldsymbol{\theta})} \mathcal{J}^{iab} \mathcal{J}^{icd}$$

$$= \frac{1}{N} \sum_{j=1}^{T^2} \sum_{k=1}^{T^2} \overline{\mathcal{M}}_{jk} \sum_{i=1}^{\dim(\boldsymbol{\theta})} \operatorname{vJac}^{ij}(\boldsymbol{h} \,|\, \boldsymbol{x}) \operatorname{vJac}^{ik}(\boldsymbol{h} \,|\, \boldsymbol{x})$$

$$= \frac{1}{N} \sum_{j=1}^{T^2} \sum_{k=1}^{T^2} \overline{\mathcal{M}}_{jk} (\operatorname{vJac}^{\top}(\boldsymbol{h} \,|\, \boldsymbol{x}) \operatorname{vJac}(\boldsymbol{h} \,|\, \boldsymbol{x}))^{jk}$$

$$= \frac{1}{N} \operatorname{tr}\left(\overline{\mathcal{M}}(\operatorname{vJac}^{\top}(\boldsymbol{h} \,|\, \boldsymbol{x}) \operatorname{vJac}(\boldsymbol{h} \,|\, \boldsymbol{x}))\right).$$

Thus, again simplifying the eigenvalue of the "squared" matrix, with Theorem D.2, we have that:

$$\frac{1}{N}\sum_{t=1}^{T} s_t^2(\text{vJac}(\boldsymbol{h}\,|\,\boldsymbol{x}))\cdot\lambda_{T-t+1}\left(\overline{\mathcal{M}}\right) \leq \text{tr}\left(\hat{\mathcal{I}}_1(\boldsymbol{\theta}\,|\,\boldsymbol{x})\right) \leq \frac{1}{N}\sum_{t=1}^{T} s_t^2(\text{vJac}(\boldsymbol{h}\,|\,\boldsymbol{x}))\cdot\lambda_t\left(\overline{\mathcal{M}}\right).$$

**For Eq. (24):**

Similar to Eq. (22), we only need to rearrange the summation. Notice that
$$\text{dHes}(\boldsymbol{h}\,|\,\boldsymbol{x}) = (\text{diag}(\text{Hes}(\boldsymbol{h}_1\,|\,\boldsymbol{x})),\ldots,\text{diag}(\text{Hes}(\boldsymbol{h}_T\,|\,\boldsymbol{x}))),$$
thus $\text{dHes}^{ia}(\boldsymbol{h}\,|\,\boldsymbol{x}) = \partial_i^2(\boldsymbol{h}_a\,|\,\boldsymbol{x})$.

$$\begin{aligned}
\mathcal{V}_2(\boldsymbol{\theta}\,|\,\boldsymbol{x}) &= \frac{1}{N}\sum_{i=1}^{\dim(\boldsymbol{\theta})}\partial_i^2\boldsymbol{h}^a(\boldsymbol{x})\partial_i^2\boldsymbol{h}^b(\boldsymbol{x})\mathcal{I}_{ab}(\boldsymbol{h}\,|\,\boldsymbol{x}) \\
&= \frac{1}{N}\mathcal{I}_{ab}(\boldsymbol{h}\,|\,\boldsymbol{x})\sum_{i=1}^{\dim(\boldsymbol{\theta})}\partial_i^2\boldsymbol{h}^a(\boldsymbol{x})\partial_i^2\boldsymbol{h}^b(\boldsymbol{x}) \\
&= \frac{1}{N}\mathcal{I}_{ab}(\boldsymbol{h}\,|\,\boldsymbol{x})\sum_{i=1}^{\dim(\boldsymbol{\theta})}\text{dHes}^{ia}(\boldsymbol{h}\,|\,\boldsymbol{x})\text{dHes}^{ib}(\boldsymbol{h}\,|\,\boldsymbol{x}) \\
&= \frac{1}{N}\mathcal{I}_{ab}(\boldsymbol{h}\,|\,\boldsymbol{x})\sum_{i=1}^{\dim(\boldsymbol{\theta})}(\text{dHes}^\top)^{ai}(\boldsymbol{h}\,|\,\boldsymbol{x})\text{dHes}^{ib}(\boldsymbol{h}\,|\,\boldsymbol{x}) \\
&= \frac{1}{N}\mathcal{I}_{ab}(\boldsymbol{h}\,|\,\boldsymbol{x})(\text{dHes}^\top(\boldsymbol{h}\,|\,\boldsymbol{x})\text{dHes}(\boldsymbol{h}\,|\,\boldsymbol{x}))^{ab} \\
&= \frac{1}{N}\text{tr}\left(\mathcal{I}(\boldsymbol{h}\,|\,\boldsymbol{x})(\text{dHes}^\top(\boldsymbol{h}\,|\,\boldsymbol{x})\text{dHes}(\boldsymbol{h}\,|\,\boldsymbol{x}))\right).
\end{aligned}$$

Thus, again simplifying the eigenvalue of the "squared" matrix, with Theorem D.2, we have that:

$$\frac{1}{N}\sum_{t=1}^{T} s_t^2(\text{dHes}(\boldsymbol{h}\,|\,\boldsymbol{x}))\cdot\lambda_{T-t+1}\left(\mathcal{I}(\boldsymbol{h}\,|\,\boldsymbol{x})\right) \leq \text{tr}\left(\hat{\mathcal{I}}_2(\boldsymbol{\theta}\,|\,\boldsymbol{x})\right) \leq \frac{1}{N}\sum_{t=1}^{T} s_t^2(\text{dHes}(\boldsymbol{h}\,|\,\boldsymbol{x}))\cdot\lambda_t\left(\mathcal{I}(\boldsymbol{h}\,|\,\boldsymbol{x})\right).$$

$\square$

# E   Second Central Moment of Categorical Distribution

*Proof.* We first notice that the exponential family density is given by,
$$p(y\,|\,\boldsymbol{x}) = \exp(\boldsymbol{h}_y(\boldsymbol{x}) - F(\boldsymbol{h}(\boldsymbol{x})))$$
and thus also have
$$F(\boldsymbol{h}(\boldsymbol{x})) = \log\sum_{t=1}^{T}\exp(\boldsymbol{h}_t(\boldsymbol{x}))$$

The first order derivative follows as,
$$\left.\frac{\partial F(\boldsymbol{h})}{\partial\boldsymbol{h}_i}\right|_{\boldsymbol{h}=\boldsymbol{h}(\boldsymbol{x})} = \frac{\exp(\boldsymbol{h}_i(\boldsymbol{x}))}{\sum_{t=1}^{T}\exp(\boldsymbol{h}_t(\boldsymbol{x}))} = \sigma_i(\boldsymbol{h}(\boldsymbol{x}))$$

As such, the second order derivatives also follow,
$$\begin{aligned}
\left.\frac{\partial^2 F(\boldsymbol{h})}{\partial\boldsymbol{h}_i\partial\boldsymbol{h}_j}\right|_{\boldsymbol{h}=\boldsymbol{h}(\boldsymbol{x})} &= \frac{\exp(\boldsymbol{h}_i(\boldsymbol{x}))\delta_{ij}\cdot\sum_{t=1}^{T}\exp(\boldsymbol{h}_t(\boldsymbol{x})) - \exp(\boldsymbol{h}_i(\boldsymbol{x}))\exp(\boldsymbol{h}_j(\boldsymbol{x}))}{\left(\sum_{t=1}^{T}\exp(\boldsymbol{h}_t(\boldsymbol{x}))\right)^2} \\
&= \sigma_i(\boldsymbol{h}(\boldsymbol{x}))\cdot\delta_{ij} - \sigma_i(\boldsymbol{h}(\boldsymbol{x}))\sigma_j(\boldsymbol{h}(\boldsymbol{x})).
\end{aligned}$$

As such, we have that
$$\mathcal{I}(\boldsymbol{h}\,|\,\boldsymbol{x}) = \text{Diag}(\sigma(\boldsymbol{x})) - \sigma(\boldsymbol{x})\sigma(\boldsymbol{x})^\top.$$

$\square$

# F  Empirical Results Continued

In the following section we present additional details and results for the experimental verification we conduct in Section 5.

## F.I  Additional Details

We note that to calculate the diagonal Hessians required for the bounds and empirical FIM calculations, we utilize the `BackPACK` [6] for `PyTorch`. Additionally, to calculate the sufficient statistics moment's spectrum, we explicitly solve the minimum and maximum eigenvalues via their optimization problems. For 2D tensors / matrices, we utilize `numpy.linalg.eig`. For 4D tensors, we utilize PyTorch Minimize [9], a wrapper for `SciPy`'s optimize function.

## F.II  Additional Plots

We present Figs. V to VIII which are the exact same experiment run in Section 5, but with different initial NN weights and random inputs.

Figures IX to XIII show the experimental results on a 5-layer MLP and log-sigmoid activation function. In most of the cases, the FIM and its associated variances quickly go to zero in the first few epochs.

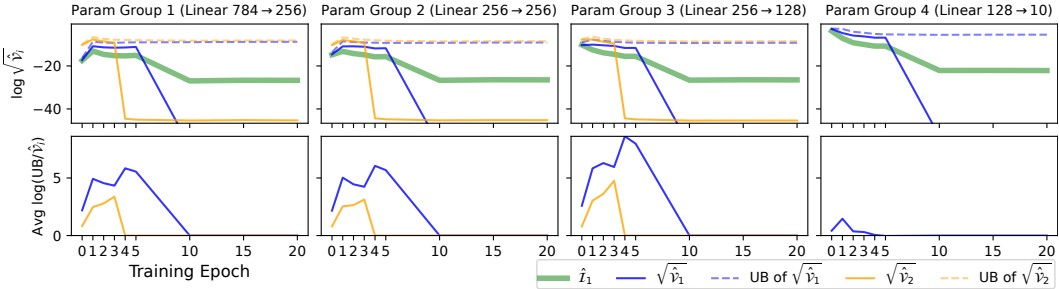

Figure V: The Fisher information, its variances and bounds of the variances w.r.t. a MLP trained with different initialization and a different input $\boldsymbol{x}$ (a)

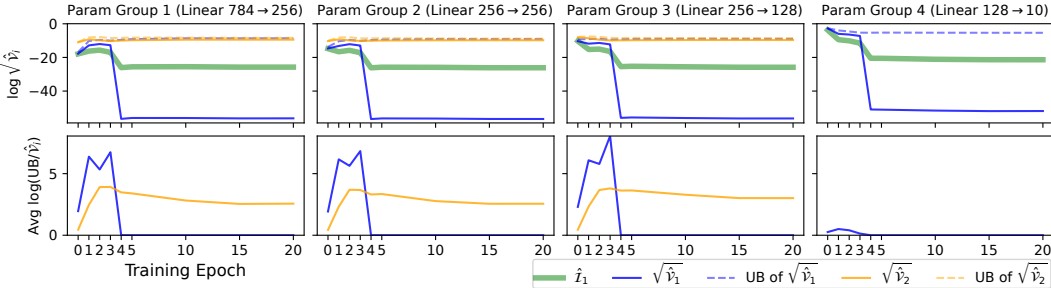

Figure VI: The Fisher information, its variances and bounds of the variances w.r.t. a MLP trained with different initialization and a different input $\boldsymbol{x}$ (b)

# G  "Empirical Fisher" Continued

Noting Lemma 6.1's characterization of the covariance, we are able to characterize the variance of the diagonal elements of $\hat{\mathrm{I}}(\boldsymbol{\theta} \,|\, \boldsymbol{x})$, denoted as $\mathrm{V}(\theta_i \,|\, \boldsymbol{x}) \doteq \mathrm{Var}(\hat{\mathrm{I}}(\theta_i \,|\, \boldsymbol{x}))$.

**Corollary G.1.** *For any* $\boldsymbol{x} \in \Re^I$,

$$\mathrm{V}(\theta_i \,|\, \boldsymbol{x}) = \frac{1}{N} \partial_i \boldsymbol{h}^a(\boldsymbol{x}) \partial_i \boldsymbol{h}^b(\boldsymbol{x}) \partial_i \boldsymbol{h}^c(\boldsymbol{x}) \partial_i \boldsymbol{h}^d(\boldsymbol{x}) \left( \mathrm{K}_{abcd}(\boldsymbol{t} \,|\, \boldsymbol{x}) - \mathrm{I}_{ab}(\boldsymbol{h} \,|\, \boldsymbol{x}) \otimes \mathrm{I}_{cd}(\boldsymbol{h} \,|\, \boldsymbol{x}) \right)$$

$$= \frac{1}{N} \partial_i \boldsymbol{h}^a(\boldsymbol{x}) \partial_i \boldsymbol{h}^b(\boldsymbol{x}) \partial_i \boldsymbol{h}^c(\boldsymbol{x}) \partial_i \boldsymbol{h}^d(\boldsymbol{x}) \mathrm{K}_{abcd}(\boldsymbol{t} \,|\, \boldsymbol{x}) - \frac{1}{N} \left( \partial_i \boldsymbol{h}^\top(\boldsymbol{x}) \left( \mathrm{Cov}^q(\boldsymbol{t} \,|\, \boldsymbol{x}) + \Delta\mathsf{H}(\boldsymbol{x}) \right) \partial_i \boldsymbol{h}(\boldsymbol{x}) \right)^2,$$

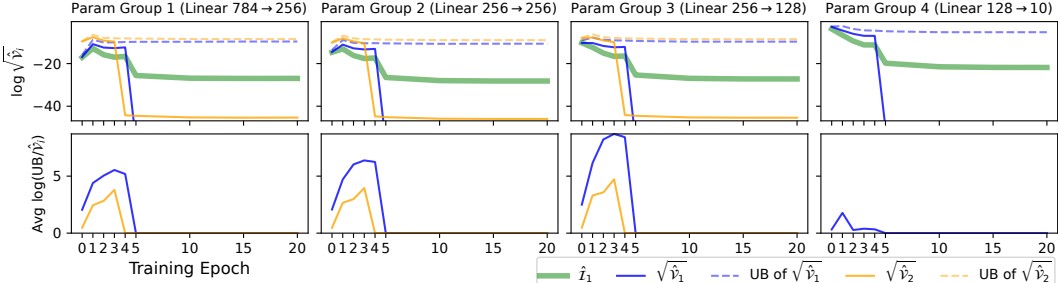

Figure VII: The Fisher information, its variances and bounds of the variances w.r.t. a MLP trained with different initialization and a different input $\boldsymbol{x}$ (c)

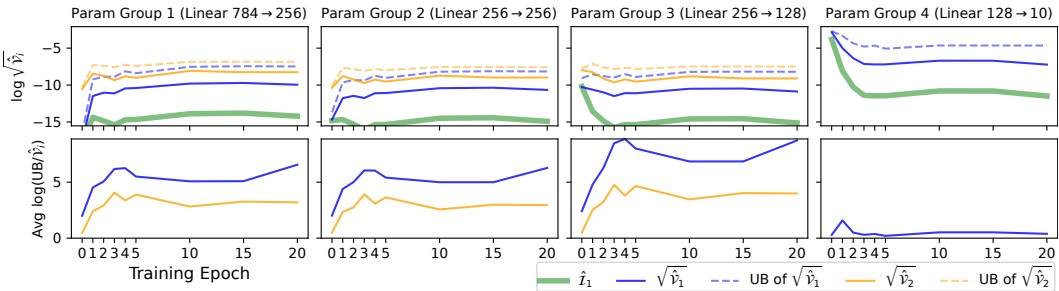

Figure VIII: The Fisher information, its variances and bounds of the variances w.r.t. a MLP trained with different initialization and a different input $\boldsymbol{x}$ (d)

*where* $\mathrm{K}(\boldsymbol{h} \,|\, \boldsymbol{x})$ *the 4th (non-central) moment of* $(\boldsymbol{t}(\hat{\boldsymbol{y}}) - \boldsymbol{\eta}(\boldsymbol{x}))$ *w.r.t.* $q(\hat{\mathbf{y}} \,|\, \boldsymbol{x})$.

As a result of the similarity of the functional forms of the empirical Fisher $\hat{\mathrm{I}}(\boldsymbol{\theta})$ and the FIM estimator $\hat{\mathcal{I}}_1(\boldsymbol{\theta})$, it is not surprising that Corollary G.1 is similar to the variance of $\hat{\mathcal{I}}_1(\theta_i \,|\, \boldsymbol{x})$. Indeed, applying Lemma 6.1 will give the exact same functional form with the 2nd central moments of $\boldsymbol{t}(\boldsymbol{y})$ w.r.t. $p(\boldsymbol{y} \,|\, \boldsymbol{x})$ exchanged with 2nd non-central moments of $(\boldsymbol{t}(\hat{\boldsymbol{y}}) - \boldsymbol{\eta}(\boldsymbol{x}))$ w.r.t. $q(\hat{\mathbf{y}} \,|\, \boldsymbol{x})$. $\mathrm{V}(\theta_i \,|\, \boldsymbol{x})$ is therefore determined by the 2nd and the 4th moment of $(\boldsymbol{t}(\hat{\boldsymbol{y}}) - \boldsymbol{\eta}(\boldsymbol{x}))$ up to the parameter transformation $\boldsymbol{\theta} \to \boldsymbol{h}$. Subsequently, the bounds presented for $\mathcal{V}_1(\theta_i \,|\, \boldsymbol{x})$ (Eq. (8) and Corollary 4.6) can be similarly adapted for $\mathrm{V}(\theta_i \,|\, \boldsymbol{x})$.

The extension of $\mathrm{V}(\theta_i \,|\, \boldsymbol{x})$ to $\mathrm{V}(\theta_i)$ can also be proven in a similar manner to Theorem 4.7.

**Corollary G.2.** *Given* $N_x$ *samples of* $\boldsymbol{x} \sim q(\boldsymbol{x})$ *and* $N$ *samples of* $\boldsymbol{y}_{\,|\,\boldsymbol{x}} \sim q(\boldsymbol{y} \,|\, \boldsymbol{x})$ *for each* $\boldsymbol{x}$ *sampled,*

$$\mathrm{V}(\theta_i) = \frac{1}{N_x} \cdot \mathrm{Var}\left(\mathrm{I}(\theta_i \,|\, \boldsymbol{x})\right) + \frac{1}{N_x} \cdot \mathop{\mathbb{E}}_{q(\boldsymbol{x})}\left[\mathrm{V}(\theta_i \,|\, \boldsymbol{x})\right]. \tag{25}$$

*where* $\mathrm{Var}\left(\mathrm{I}(\theta_i \,|\, \boldsymbol{x})\right)$ *is the variance of* $\mathrm{I}(\theta_i \,|\, \boldsymbol{x})$ *w.r.t.* $q(\boldsymbol{x})$.

If $q(\boldsymbol{x}, \boldsymbol{y}) = \frac{1}{N} \sum_{k=1}^{N} \delta(\boldsymbol{x} - \boldsymbol{x}_k) \cdot \delta(\boldsymbol{y} - \hat{\boldsymbol{y}}_k)$ for a set of observations $\{(\boldsymbol{x}_k, \hat{\boldsymbol{y}}_k)\}_{k=1}^{N}$, then one can directly evaluate the DFIM without sampling and achieve zero variance, *i.e.*, $\hat{\mathrm{I}}(\boldsymbol{\theta}) = \mathrm{I}(\boldsymbol{\theta})$. In this scenario, there is a clear trade-off between the estimators of the FIM in Eq. (3) and the DFIM. The estimators of the FIM are unbiased, but have a variance; while the DFIM has zero variance, but is a biased approximation of the FIM.

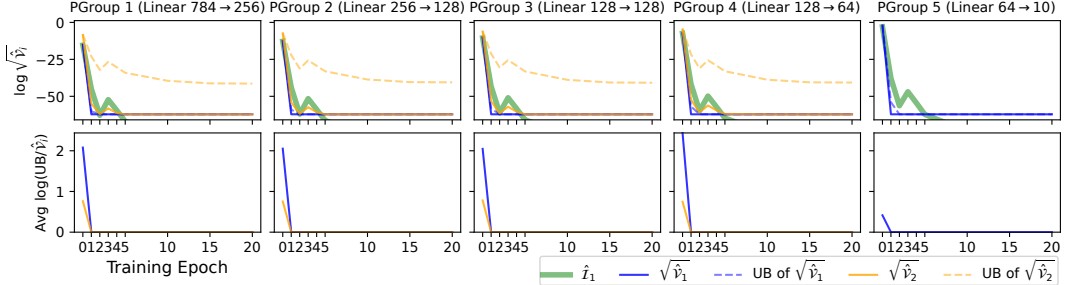

Figure IX: The Fisher information, its variances and bounds of the variances w.r.t. a 5-layer MLP with log-sigmoid activation.

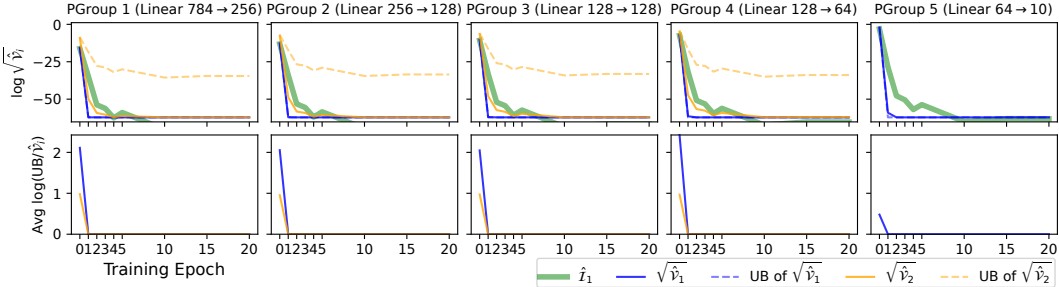

Figure X: The Fisher information, its variances and bounds of the variances w.r.t. a 5-layer MLP with log-sigmoid activation.

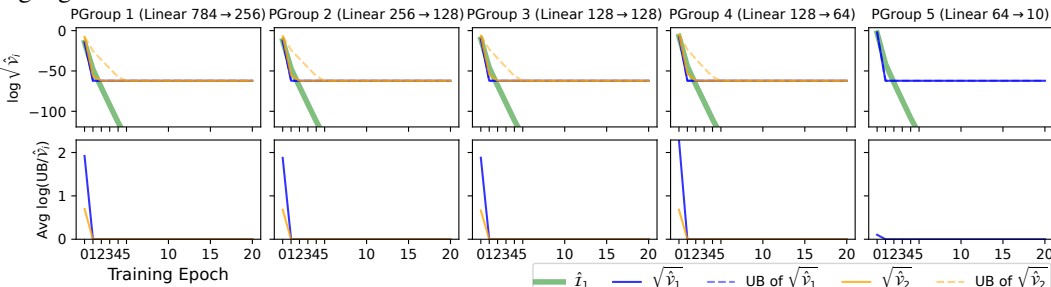

Figure XI: The Fisher information, its variances and bounds of the variances w.r.t. a 5-layer MLP with log-sigmoid activation.

# H  Derivation of Eq. (3) Using Log-Partition Function Derivatives

In what follows, we derive the alternative equations for $\hat{\mathcal{I}}_1(\theta_i)$ and $\hat{\mathcal{I}}_2(\theta_i)$ presented in Section 3. That is, we seek to derive the following equations:

$$\hat{\mathcal{I}}_1(\theta_i) = \frac{1}{N} \sum_{k=1}^{N} \left( \frac{\partial F(\boldsymbol{h}(\boldsymbol{x}_k))}{\partial \theta_i} - \frac{\partial \boldsymbol{h}^a(\boldsymbol{x}_k)}{\partial \theta_i} \cdot \boldsymbol{t}_a(\boldsymbol{y}_k) \right)^2 ; \tag{26}$$

$$\hat{\mathcal{I}}_2(\theta_i) = \frac{1}{N} \sum_{k=1}^{N} \left( \frac{\partial^2 F(\boldsymbol{h}(\boldsymbol{x}_k))}{\partial^2 \theta_i} - \frac{\partial^2 \boldsymbol{h}^a(\boldsymbol{x}_k)}{\partial^2 \theta_i} \cdot \boldsymbol{t}_a(\boldsymbol{y}_k) \right) . \tag{27}$$

We calculate the equations separately.

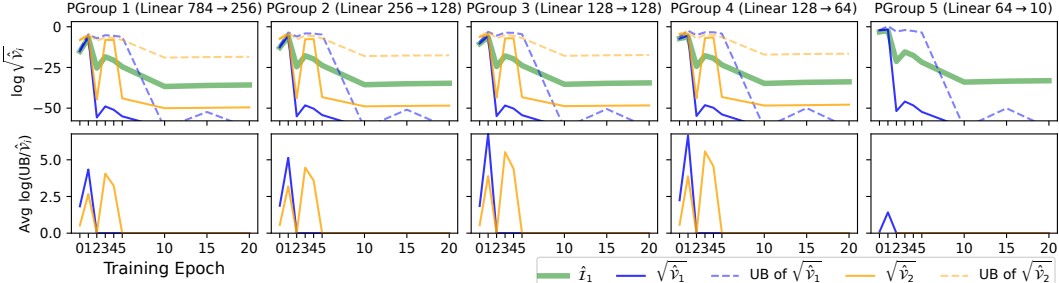

Figure XII: The Fisher information, its variances and bounds of the variances w.r.t. a 5-layer MLP with log-sigmoid activation.

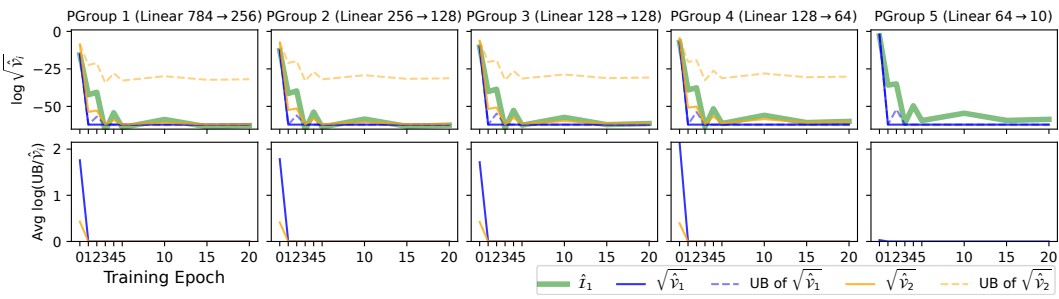

Figure XIII: The Fisher information, its variances and bounds of the variances w.r.t. a 5-layer MLP with log-sigmoid activation.

## H.I  Eq. (26)

*Proof.* For Eq. (26), we note that

$$
\frac{\partial \log p(\boldsymbol{y}_k \mid \boldsymbol{x}_k)}{\partial \theta_i} = \frac{\partial}{\partial \theta_i} \left( \boldsymbol{t}^\top(\boldsymbol{y}_k) \boldsymbol{h}(\boldsymbol{x}_k) - F(\boldsymbol{h}(\boldsymbol{x}_k)) \right)
$$

$$
= \boldsymbol{t}_a(\boldsymbol{y}_k) \frac{\partial \boldsymbol{h}^a(\boldsymbol{x}_k)}{\partial \theta_i} - F_a'(\boldsymbol{h}(\boldsymbol{x}_k)) \frac{\partial \boldsymbol{h}^a(\boldsymbol{x}_k)}{\partial \theta_i}
$$

$$
= \boldsymbol{t}_a(\boldsymbol{y}_k) \frac{\partial \boldsymbol{h}^a(\boldsymbol{x}_k)}{\partial \theta_i} - \boldsymbol{\eta}_a(\boldsymbol{x}_k) \frac{\partial \boldsymbol{h}^a(\boldsymbol{x}_k)}{\partial \theta_i}
$$

$$
= (\boldsymbol{t}_a(\boldsymbol{y}_k) - \boldsymbol{\eta}_a(\boldsymbol{x}_k)) \cdot \frac{\partial \boldsymbol{h}^a(\boldsymbol{x}_k)}{\partial \theta_i},
$$

where we note that $F_a'(\boldsymbol{h}(\boldsymbol{x}_k)) = \boldsymbol{\eta}_a(\boldsymbol{x}_k)$ which follows from the connection to expected parameters and partition functions of exponential families, see *e.g.* [37].

Then Eq. (26) follows immediately. □

## H.II  Eq. (27)

*Proof.* For Eq. (27), we also calculate the derivative:

$$
\frac{\partial \log p(\boldsymbol{y}_k \mid \boldsymbol{x}_k)}{\partial \theta_i} = \frac{\partial}{\partial \theta_i} \left( \boldsymbol{t}^\top(\boldsymbol{y}_k) \boldsymbol{h}(\boldsymbol{x}_k) - F(\boldsymbol{h}(\boldsymbol{x}_k)) \right)
$$

$$
= \boldsymbol{t}_a(\boldsymbol{y}_k) \cdot \frac{\partial \boldsymbol{h}^a(\boldsymbol{x}_k)}{\partial \theta_i} - \frac{\partial F(\boldsymbol{h}(\boldsymbol{x}_k))}{\partial \theta_i}.
$$

Then

$$
\frac{\partial^2 \log p(\boldsymbol{y}_k \mid \boldsymbol{x}_k)}{\partial^2 \theta_i} = \frac{\partial}{\partial \theta_i} \left( \boldsymbol{t}_a(\boldsymbol{y}_k) \cdot \frac{\partial \boldsymbol{h}^a(\boldsymbol{x}_k)}{\partial \theta_i} - \frac{\partial F(\boldsymbol{h}(\boldsymbol{x}_k))}{\partial \theta_i} \right)
$$

$$
= \boldsymbol{t}_a(\boldsymbol{y}_k) \cdot \frac{\partial^2 \boldsymbol{h}^a(\boldsymbol{x}_k)}{\partial^2 \theta_i} - \frac{\partial^2 F(\boldsymbol{h}(\boldsymbol{x}_k))}{\partial^2 \theta_i}.
$$

Then Eq. (27) follows immediately. □

**Remark H.1.** Although Eq. (27) is useful in practice, *i.e.*, it states an equation which can be calculated via automatic differentiation, in the appendix and proofs we use an alternative equation. In particular, we use

$$\hat{\mathcal{I}}_2(\theta_i) = \frac{1}{N} \sum_{k=1}^{N} \left( (\boldsymbol{\eta}_a(\boldsymbol{x}_k) - \boldsymbol{t}_a(\boldsymbol{y}_k)) \cdot \frac{\partial^2 \boldsymbol{h}^a(\boldsymbol{x}_k)}{\partial^2 \theta_i} \right.$$

$$\left. + \frac{\partial \boldsymbol{h}^a(\boldsymbol{x}_k)}{\partial \theta_i} \cdot \mathcal{I}_{ab}(\boldsymbol{h} \,|\, \boldsymbol{x}_k) \cdot \frac{\partial \boldsymbol{h}^b(\boldsymbol{x}_k)}{\partial \theta_i} \right),$$

which follows from taking the derivative of $\partial_i \log p(\boldsymbol{y}_k \,|\, \boldsymbol{x}_k)$ in the proof of Eq. (26) (above).

# I   Proof of Eq. (7)

We first begin by proving the follow lemma to bound an $\Re^{n \times n}$ matrix.

**Lemma I.1.** *Let $A \in \Re^{n \times n}$ and $\boldsymbol{v} \in \Re^n$, then*

$$\|\boldsymbol{v}\|_2^2 \cdot \lambda_{\min}(A) \leq \boldsymbol{v}^a \boldsymbol{v}^b A_{ab} \leq \|\boldsymbol{v}\|_2^2 \cdot \lambda_{\max}(A).$$

*Proof.* The proof follows immediately from the Courant-Fischer min-max theorem [42]. That is,

$$\lambda_{\min}(A) = \inf_{\boldsymbol{u}:\|\boldsymbol{u}\|=1} \boldsymbol{u}^a \boldsymbol{u}^b A_{ab};$$

$$\lambda_{\max}(A) = \sup_{\boldsymbol{u}:\|\boldsymbol{u}\|=1} \boldsymbol{u}^a \boldsymbol{u}^b A_{ab}.$$

Thus it follows that:

$$\boldsymbol{v}^a \boldsymbol{v}^b A_{ab} = \|v\|_2^2 \cdot (\boldsymbol{v}/\|v\|_2)^a (\boldsymbol{v}/\|v\|_2)^b A_{ab}$$

$$\leq \|v\|_2^2 \cdot \lambda_{\max}(A).$$

The lower bound follows identically.

We note that this can be similarly proven via trace bounds, *e.g.*, [43]. □

Now we can prove Eq. (7).

*Proof.* The proof follows from Lemma 3.1, Eq. (4), and directly applying Lemma I.1. □

# J   Proof of Eq. (8)

Let us first define the maximum and minimum Z-eigenvalues of a 4-dimensional tensor $\mathcal{K}$.

$$\tilde{\lambda}_{\min}(\mathcal{K}) = \inf_{\boldsymbol{u}:\|\boldsymbol{u}\|_2=1} \boldsymbol{u}^a \boldsymbol{u}^b \boldsymbol{u}^c \boldsymbol{u}^d \mathcal{K}_{abcd}; \tag{28}$$

$$\tilde{\lambda}_{\max}(\mathcal{K}) = \sup_{\boldsymbol{u}:\|\boldsymbol{u}\|_2=1} \boldsymbol{u}^a \boldsymbol{u}^b \boldsymbol{u}^c \boldsymbol{u}^d \mathcal{K}_{abcd}. \tag{29}$$

Now We first prove the following lemma regarding the Z-eigenvalues.

**Lemma J.1.** *Suppose $\mathcal{K}$ is 4-dimensional tensor. Then we have*

$$\|\boldsymbol{v}\|_2^4 \cdot \tilde{\lambda}_{\min}(\mathcal{K}) \leq \boldsymbol{v}^a \boldsymbol{v}^b \boldsymbol{v}^c \boldsymbol{v}^d \mathcal{K}_{abcd} \leq \|\boldsymbol{v}\|_2^4 \cdot \tilde{\lambda}_{\max}(\mathcal{K}) \tag{30}$$

*Proof.* The proof follows similarly to Lemma I.1. We simple use the following calculation:

$$\boldsymbol{v}^a \boldsymbol{v}^b \boldsymbol{v}^c \boldsymbol{v}^d \mathcal{K}_{abcd}$$

$$= \|\boldsymbol{v}\|_2^4 \cdot (\boldsymbol{v}/\|\boldsymbol{v}\|_2)^a (\boldsymbol{v}/\|\boldsymbol{v}\|_2)^b (\boldsymbol{v}/\|\boldsymbol{v}\|_2)^c (\boldsymbol{v}/\|\boldsymbol{v}\|_2)^d \mathcal{K}_{abcd}$$

$$\leq \|\boldsymbol{v}\|_2^4 \cdot \sup_{\boldsymbol{u}:\|\boldsymbol{u}\|_2=1} \boldsymbol{u}^a \boldsymbol{u}^b \boldsymbol{u}^c \boldsymbol{u}^d \mathcal{K}_{abcd}$$

$$= \|\boldsymbol{v}\|_2^4 \cdot \tilde{\lambda}_{\max}(\mathcal{K}).$$

The minimum case is proven identically (with the opposite inequality). □

Now we can prove the bounds of Eq. (8)

*Proof.* From Lemma 3.1, we have that

$$\mathcal{V}_1(\theta_i \,|\, \boldsymbol{x}) = \frac{1}{N}\partial_i\boldsymbol{h}^a(\boldsymbol{x})\partial_i\boldsymbol{h}^b(\boldsymbol{x})\partial_i\boldsymbol{h}^c(\boldsymbol{x})\partial_i\boldsymbol{h}^d(\boldsymbol{x})\left[\mathcal{K}_{abcd} - \mathcal{I}_{ab}(\boldsymbol{h}\,|\,\boldsymbol{x})\cdot\mathcal{I}_{cd}(\boldsymbol{h}\,|\,\boldsymbol{x})\right],$$

where we shorthand $\mathcal{K}_{abcd} = \mathcal{K}^p_{abcd}(\boldsymbol{t}\,|\,\boldsymbol{x})$

We bound two terms.

$$\|\partial_i\boldsymbol{h}(\boldsymbol{x})\|_2^4 \cdot \tilde{\lambda}_{\min}(\mathcal{K}) \le \partial_i\boldsymbol{h}^a(\boldsymbol{x})\partial_i\boldsymbol{h}^b(\boldsymbol{x})\partial_i\boldsymbol{h}^c(\boldsymbol{x})\partial_i\boldsymbol{h}^d(\boldsymbol{x})\mathcal{K}_{abcd}$$
$$\le \|\partial_i\boldsymbol{h}(\boldsymbol{x})\|_2^4 \cdot \tilde{\lambda}_{\max}(\mathcal{K}),$$

which follows directly from Lemma J.1

We now bound the second term in a similar way, taking $v^a \doteq \partial_i\boldsymbol{h}^a(\boldsymbol{x})$ and noting that

$$v^a v^b v^c v^d \mathcal{I}_{ab}(\boldsymbol{h}\,|\,\boldsymbol{x})\mathcal{I}_{cd}(\boldsymbol{h}\,|\,\boldsymbol{x}) = (v^a v^b \mathcal{I}_{ab}(\boldsymbol{h}\,|\,\boldsymbol{x}))^2$$

which directly gives us,

$$\left(\|v\|_2^2 \cdot \lambda_{\min}(\mathcal{I}(\boldsymbol{h}\,|\,\boldsymbol{x}))\right)^2 \le v^a v^b v^c v^d \mathcal{I}_{ab}(\boldsymbol{h}\,|\,\boldsymbol{x})\mathcal{I}_{cd}(\boldsymbol{h}\,|\,\boldsymbol{x})$$
$$\le \left(\|v\|_2^2 \cdot \lambda_{\max}(\mathcal{I}(\boldsymbol{h}\,|\,\boldsymbol{x}))\right)^2.$$

which follows from Lemma I.1.

Thus, together these bounds prove Eq. (8). $\qquad\square$

## K  Proof of Eq. (9)

From Lemma 3.1 we have that,

$$\mathcal{V}_2^i = \frac{1}{N}\partial_i^2\boldsymbol{h}^a(\boldsymbol{x})\partial_i^2\boldsymbol{h}^b(\boldsymbol{x})\mathcal{I}_{ab}(\boldsymbol{h}_L).$$

Thus we get

$$\|\partial_i^2\boldsymbol{h}(\boldsymbol{x})\|_2^2 \cdot \lambda_{\min}\left(\mathcal{I}(\boldsymbol{h})\right) \le \partial_i^2\boldsymbol{h}^b(\boldsymbol{x})\cdot\partial_i^2\boldsymbol{h}^b(\boldsymbol{x})\cdot\mathcal{I}_{ab}(\boldsymbol{h}) \le \|\partial_i^2\boldsymbol{h}(\boldsymbol{x})\|_2^2 \cdot \lambda_{\max}\left(\mathcal{I}(\boldsymbol{h})\right),$$

which follows from Lemma I.1. This immediately gives the bound as required.

## L  Proof of Corollary 4.2

*Proof.* The corollary holds from distributing the inf or sup and examining how the variational definition of the generalized 'eigenvalue' simplifies under tensor products.

Indeed, for the minimum case,

$$\tilde{\lambda}_{\min}\left(\mathcal{K}^p(\boldsymbol{t}\,|\,\boldsymbol{x}) - \mathcal{I}(\boldsymbol{h}\,|\,\boldsymbol{x})\otimes\mathcal{I}(\boldsymbol{h}\,|\,\boldsymbol{x})\right)$$
$$= \inf_{\boldsymbol{u}:\|\boldsymbol{u}\|_2=1}\boldsymbol{u}^a\boldsymbol{u}^b\boldsymbol{u}^c\boldsymbol{u}^d\left(\mathcal{K}^p_{abcd}(\boldsymbol{t}\,|\,\boldsymbol{x}) - \mathcal{I}_{ab}(\boldsymbol{h}\,|\,\boldsymbol{x})\cdot\mathcal{I}_{cd}(\boldsymbol{h}\,|\,\boldsymbol{x})\right)$$
$$\ge \left(\inf_{\boldsymbol{u}:\|\boldsymbol{u}\|_2=1}\boldsymbol{u}^a\boldsymbol{u}^b\boldsymbol{u}^c\boldsymbol{u}^d\mathcal{K}^p_{abcd}(\boldsymbol{t}\,|\,\boldsymbol{x})\right) + \left(\inf_{\boldsymbol{u}:\|\boldsymbol{u}\|_2=1}\boldsymbol{u}^a\boldsymbol{u}^b\boldsymbol{u}^c\boldsymbol{u}^d(-\mathcal{I}_{ab}(\boldsymbol{h}\,|\,\boldsymbol{x})\cdot\mathcal{I}_{cd}(\boldsymbol{h}\,|\,\boldsymbol{x}))\right)$$
$$= \left(\inf_{\boldsymbol{u}:\|\boldsymbol{u}\|_2=1}\boldsymbol{u}^a\boldsymbol{u}^b\boldsymbol{u}^c\boldsymbol{u}^d\mathcal{K}^p_{abcd}(\boldsymbol{t}\,|\,\boldsymbol{x})\right) - \left(\sup_{\boldsymbol{u}:\|\boldsymbol{u}\|_2=1}\boldsymbol{u}^a\boldsymbol{u}^b\boldsymbol{u}^c\boldsymbol{u}^d(\mathcal{I}_{ab}(\boldsymbol{h}\,|\,\boldsymbol{x})\cdot\mathcal{I}_{cd}(\boldsymbol{h}\,|\,\boldsymbol{x}))\right)$$
$$= \left(\inf_{\boldsymbol{u}:\|\boldsymbol{u}\|_2=1}\boldsymbol{u}^a\boldsymbol{u}^b\boldsymbol{u}^c\boldsymbol{u}^d\mathcal{K}^p_{abcd}(\boldsymbol{t}\,|\,\boldsymbol{x})\right) - \left(\sup_{\boldsymbol{u}:\|\boldsymbol{u}\|_2=1}\left(\boldsymbol{u}^a\boldsymbol{u}^b\mathcal{I}_{ab}(\boldsymbol{h}\,|\,\boldsymbol{x})\right)^2\right)$$
$$\ge \left(\inf_{\boldsymbol{u}:\|\boldsymbol{u}\|_2=1}\boldsymbol{u}^a\boldsymbol{u}^b\boldsymbol{u}^c\boldsymbol{u}^d\mathcal{K}^p_{abcd}(\boldsymbol{t}\,|\,\boldsymbol{x})\right) - \left(\sup_{\boldsymbol{u}:\|\boldsymbol{u}\|_2=1}\boldsymbol{u}^a\boldsymbol{u}^b\mathcal{I}_{ab}(\boldsymbol{h}\,|\,\boldsymbol{x})\right)^2,$$

where the last line holds from the fact that $\mathcal{I}_{ab}(\boldsymbol{h} \,|\, \boldsymbol{x})$ is PSD (thus the inner Einstein summation is always positive).

Taking definitions of the types of eigenvalues, gives the statement.

We note that the 'max' case follows identically.

Additionally, for the lower bound, we can show the non-triviallity of the non-negativity of the minimum eigenvalue.

We note that $\mathcal{K}_{abcd}^p(\boldsymbol{t} \,|\, \boldsymbol{x}) = \mathbb{E}_p[\boldsymbol{v}_a \boldsymbol{v}_b \boldsymbol{v}_c \boldsymbol{v}_d]$, where $\boldsymbol{v} = \boldsymbol{t}(y) - \eta(\boldsymbol{x})$.

Thus we have that

$$
\begin{aligned}
\tilde{\lambda}_{\min} & \left( \mathcal{K}^p(\boldsymbol{t} \,|\, \boldsymbol{x}) - \mathcal{I}(\boldsymbol{h} \,|\, \boldsymbol{x}) \otimes \mathcal{I}(\boldsymbol{h} \,|\, \boldsymbol{x}) \right) \\
& = \inf_{\boldsymbol{u}: \|\boldsymbol{u}\|_2 = 1} \boldsymbol{u}^a \boldsymbol{u}^b \boldsymbol{u}^c \boldsymbol{u}^d \left( \mathcal{K}_{abcd}^p(\boldsymbol{t} \,|\, \boldsymbol{x}) - \mathcal{I}_{ab}(\boldsymbol{h} \,|\, \boldsymbol{x}) \cdot \mathcal{I}_{cd}(\boldsymbol{h} \,|\, \boldsymbol{x}) \right) \\
& = \inf_{\boldsymbol{u}: \|\boldsymbol{u}\|_2 = 1} \mathbb{E}_p \left[ \boldsymbol{u}^a \boldsymbol{u}^b \boldsymbol{u}^c \boldsymbol{u}^d \left( \boldsymbol{v}_a \boldsymbol{v}_b \boldsymbol{v}_c \boldsymbol{v}_d - \mathcal{I}_{ab}(\boldsymbol{h} \,|\, \boldsymbol{x}) \cdot \mathcal{I}_{cd}(\boldsymbol{h} \,|\, \boldsymbol{x}) \right) \right] \\
& = \inf_{\boldsymbol{u}: \|\boldsymbol{u}\|_2 = 1} \mathbb{E}_p \left[ \left( \boldsymbol{u}^a \boldsymbol{u}^b \left( \boldsymbol{v}_a \boldsymbol{v}_b - \mathcal{I}_{ab}(\boldsymbol{h} \,|\, \boldsymbol{x}) \right) \right)^2 \right] \geq 0.
\end{aligned}
$$

Equality holds from simply looking at the definition of $\mathcal{K}^p(\boldsymbol{t} \,|\, \boldsymbol{x})$ and $\mathcal{I}(\boldsymbol{h} \,|\, \boldsymbol{x})$ (as moments). $\qquad\square$

## M    Proof of Proposition 4.3

*Proof.* Letting $\boldsymbol{v} = \boldsymbol{t}(y) - \boldsymbol{\eta}(y)$, we note that the maximum eigenvalue is given by,

$$
\begin{aligned}
\tilde{\lambda}_{\max} \left( \mathcal{K}^p(\boldsymbol{t} \,|\, \boldsymbol{x}) \right) & = \sup_{\boldsymbol{u}: \|\boldsymbol{u}\|_2 = 1} \boldsymbol{u}^a \boldsymbol{u}^b \boldsymbol{u}^c \boldsymbol{u}^d \mathcal{K}_{abcd}^p(\boldsymbol{t} \,|\, \boldsymbol{x}) \\
& = \sup_{\boldsymbol{u}: \|\boldsymbol{u}\|_2 = 1} \mathbb{E}_p \left[ \boldsymbol{u}^a \boldsymbol{u}^b \boldsymbol{u}^c \boldsymbol{u}^d \boldsymbol{v}_a \boldsymbol{v}_b \boldsymbol{v}_c \boldsymbol{v}_d \right] \\
& = \sup_{\boldsymbol{u}: \|\boldsymbol{u}\|_2 = 1} \mathbb{E}_p \left[ (\boldsymbol{u}^\top \boldsymbol{v})^4 \right] \\
& = \sup_{\boldsymbol{u}: \|\boldsymbol{u}\|_2 = 1} \mathbb{E}_p \left[ (\boldsymbol{u}^\top \boldsymbol{v})^2 (\boldsymbol{u}^\top \boldsymbol{v})^2 \right] \\
& \leq \sup_{\boldsymbol{u}: \|\boldsymbol{u}\|_2 = 1} \mathbb{E}_p \left[ (\|\boldsymbol{u}\|_2 \cdot \|\boldsymbol{v}\|_2)^2 (\boldsymbol{u}^\top \boldsymbol{v})^2 \right] \\
& \leq B \cdot \sup_{\boldsymbol{u}: \|\boldsymbol{u}\|_2 = 1} \mathbb{E}_p \left[ (\boldsymbol{u}^\top \boldsymbol{v})^2 \right] \\
& = B \cdot \lambda_{\max} \left( \mathcal{I}(\boldsymbol{h} \,|\, \boldsymbol{x}) \right).
\end{aligned}
$$

$\qquad\square$

## N    Proof of Corollary 4.6

*Proof.* We split up the proof into the two arguments of the various $\min$-function.

**For the right term:**

Suppose that we have a bound such that $\mathcal{V}_j(\theta_i \,|\, \boldsymbol{x}) \leq \alpha \beta_i$. Then,

$$
\begin{aligned}
\|\mathcal{V}_j(\boldsymbol{\theta} \,|\, \boldsymbol{x})\|_2^2 & = \sum_{i=1}^{\dim(\boldsymbol{\theta})} (\mathcal{V}_j(\theta_i \,|\, \boldsymbol{x}))^2 \\
& \leq \sum_{i=1}^{\dim(\boldsymbol{\theta})} (\alpha \beta_i)^2 \\
& = \alpha^2 \sum_{i=1}^{\dim(\boldsymbol{\theta})} \beta_i^2.
\end{aligned}
$$

Thus we have,

$$\|\mathcal{V}_j(\boldsymbol{\theta}\,|\,\boldsymbol{x})\|_2 \le \alpha\sqrt{\sum_{i=1}^{\dim(\boldsymbol{\theta})} \beta_i^2}.$$

Taking the appropriate $\alpha$ and $\beta$ from Eqs. (8) and (9) proves the case for Eqs. (13) and (15). For Eq. (14), that is taking

$$\alpha = \frac{1}{N} \cdot \left(\tilde{\lambda}_{\max}(\mathcal{K}^p(\boldsymbol{t}\,|\,\boldsymbol{x})) - \lambda_{\min}^2\left(\mathcal{I}(\boldsymbol{h}\,|\,\boldsymbol{x})\right)\right);$$
$$\beta_i = \|\partial_i \boldsymbol{h}(\boldsymbol{x})\|_2^4.$$

Where we note that

$$\sqrt{\sum_{i=1}^{\dim(\boldsymbol{\theta})} \left(\|\partial_i \boldsymbol{h}(\boldsymbol{x})\|_2^4\right)^2} = \sqrt{\sum_{i=1}^{\dim(\boldsymbol{\theta})} \left[\left(\sum_{t=1}^{T} [\partial_i h_t(\boldsymbol{x})]^2\right)^2\right]^2}$$

$$= \sqrt{\sum_{i=1}^{\dim(\boldsymbol{\theta})} \left(\sum_{t=1}^{T} [\partial_i h_t(\boldsymbol{x})]^2\right)^4}$$

$$\le \sqrt{\left(\sum_{i=1}^{\dim(\boldsymbol{\theta})} \sum_{t=1}^{T} [\partial_i h_t(\boldsymbol{x})]^2\right)^4}$$

$$= \left(\sum_{i=1}^{\dim(\boldsymbol{\theta})} \sum_{t=1}^{T} [\partial_i h_t(\boldsymbol{x})]^2\right)^2$$

$$= \|\partial \boldsymbol{h}(\boldsymbol{x})\|_F^4.$$

For Eq. (15), that is taking

$$\alpha = \frac{1}{N} \cdot \lambda_{\max}(\mathcal{I}(\boldsymbol{h}\,|\,\boldsymbol{x}));$$
$$\beta_i = \|\partial_i^2 \boldsymbol{h}(\boldsymbol{x})\|_2^2.$$

Where we note that

$$\sqrt{\sum_{i=1}^{\dim(\boldsymbol{\theta})} \left(\|\partial_i^2 \boldsymbol{h}(\boldsymbol{x})\|_2^2\right)^2} = \sqrt{\sum_{i=1}^{\dim(\boldsymbol{\theta})} \left(\sum_{t=1}^{T} [\partial_i^2 h_t(\boldsymbol{x})]^2\right)^2}$$

$$\le \sqrt{\left(\sum_{i=1}^{\dim(\boldsymbol{\theta})} \sum_{t=1}^{T} [\partial_i^2 h_t(\boldsymbol{x})]^2\right)^2}$$

$$= \sum_{i=1}^{\dim(\boldsymbol{\theta})} \sum_{t=1}^{T} \left[\partial_i^2 h_t(\boldsymbol{x})\right]^2$$

$$= \|\mathrm{dHes}(\boldsymbol{h}\,|\,\boldsymbol{x})\|_F^2.$$

**For the left term:**

We take the largest singular value of the network derivative term. We then further notice that $s_{\max}(A) \le \|A\|_F$ from norm ordering (of the matrix 2-norm).

To further elaborate on the Eq. ([14](#)) case, we further need to simplify the following:

$$s_{\max}(\mathrm{vJac}) \leq \|\mathrm{vJac}(\boldsymbol{h}\,|\,\boldsymbol{x})\|_F$$

$$= \sqrt{\sum_{i=1}^{\dim(\boldsymbol{\theta})} \|\partial_i \boldsymbol{h}(\boldsymbol{x})\partial_i \boldsymbol{h}^\top(\boldsymbol{x})\|_F^2}$$

$$= \sqrt{\sum_{a,b=1}^{T} \sum_{i=1}^{\dim(\boldsymbol{\theta})} (\partial_i \boldsymbol{h}^a(\boldsymbol{x}))^2 (\partial_i \boldsymbol{h}^b(\boldsymbol{x}))^2}$$

$$\leq \sqrt{\sum_{a,b=1}^{T} \|(\partial \boldsymbol{h}^a(\boldsymbol{x}))^2\|_2 \cdot \|(\partial \boldsymbol{h}^b(\boldsymbol{x}))^2\|_2}$$

$$= \sqrt{\left(\sum_{a=1}^{T} \|(\partial \boldsymbol{h}^a(\boldsymbol{x}))^2\|_2\right)^2}$$

$$= \sum_{a=1}^{T} \|(\partial \boldsymbol{h}^a(\boldsymbol{x}))^2\|_2$$

$$= \sum_{a=1}^{T} \sqrt{\sum_{i=1}^{\dim(\boldsymbol{\theta})} (\partial_i \boldsymbol{h}^a(\boldsymbol{x}))^4}$$

$$\leq \sum_{a=1}^{T} \sum_{i=1}^{\dim(\boldsymbol{\theta})} |(\partial_i \boldsymbol{h}^a(\boldsymbol{x}))^2|$$

$$= \|\partial \boldsymbol{h}(\boldsymbol{x})\|_F^2,$$

where the last inequality follows from the norm ordering $\|\cdot\|_2 \leq \|\cdot\|_1$. $\qquad\square$

## O   Proof of Theorem [4.7](#)

To prove the Theorem, we will utilize the law of total variances. We note, that by the premise of the Theorem, we are sampling $N_x$ many samples from $q(\boldsymbol{x})$ and $N$ many samples from $q(\boldsymbol{y}\,|\,\boldsymbol{y})$ for each $\boldsymbol{y}$ initially sampled. To make this clear, the samples and sampling will be notated by:

$$\boldsymbol{x}_k \sim q(\boldsymbol{x})$$
$$\boldsymbol{y}_{l\,|\,\boldsymbol{x}_k} \sim p(\boldsymbol{y}\,|\,\boldsymbol{x}_k)$$

Note that using these samples, our empirical estimators for the FIM (for either estimator) will be of the form:

$$\hat{\mathcal{I}}_1(\theta_i) = \frac{1}{N_x} \sum_{\boldsymbol{x}_k} \left(\frac{1}{N} \sum_{\boldsymbol{y}_{l\,|\,\boldsymbol{x}_k}} f(\boldsymbol{x}_k, \boldsymbol{y}_{l\,|\,\boldsymbol{x}_k})\right),$$

for an appropriately chosen $f$.

This also gives:

$$\hat{\mathcal{I}}_j(\theta_i\,|\,\boldsymbol{x}) = \frac{1}{N} \sum_{\boldsymbol{y}_{l\,|\,\boldsymbol{x}}} f(\boldsymbol{x}, \boldsymbol{y}_{l\,|\,\boldsymbol{x}}).$$

Now, we simplify the variance as follows:

$$\mathrm{Var}\left[\frac{1}{N_x}\sum_{\bm{x}_k}\left(\frac{1}{N}\sum_{\bm{y}_{l\,|\,\bm{x}_k}}f(\bm{x}_k,\bm{y}_{l\,|\,\bm{x}_k})\right)\right]$$

$$=\frac{1}{N_x^2}\sum_{\bm{x}_k}\left(\mathrm{Var}\left[\frac{1}{N}\sum_{\bm{y}_{l\,|\,\bm{x}_k}}f(\bm{x}_k,\bm{y}_{l\,|\,\bm{x}_k})\right]\right)$$

$$=\frac{1}{N_x^2}\sum_{\bm{x}_k}\mathrm{Var}\left[\hat{\mathcal{I}}_j(\theta_i\,|\,\bm{x}_k)\right]$$

$$=\frac{1}{N_x^2}\sum_{\bm{x}_k}\left(\mathrm{Var}_{\bm{x}_k}\left[\underset{\bm{y}_{1\,|\,\bm{x}_k},\dots,\bm{y}_{N\,|\,\bm{x}_k}}{\mathbb{E}}\left[\hat{\mathcal{I}}_j(\theta_i\,|\,\bm{x}_k)\right]\right]+\underset{\bm{x}_k}{\mathbb{E}}\left[\mathrm{Var}_{\bm{y}_{1\,|\,\bm{x}_k},\dots,\bm{y}_{N\,|\,\bm{x}_k}}\left[\hat{\mathcal{I}}_j(\theta_i\,|\,\bm{x}_k)\right]\right]\right)$$

$$=\frac{1}{N_x^2}\sum_{\bm{x}_k}\left(\mathrm{Var}_{\bm{x}_k}\left[\mathcal{I}(\theta_i\,|\,\bm{x}_k)\right]+\underset{\bm{x}_k}{\mathbb{E}}\left[\mathcal{V}_j(\theta_i\,|\,\bm{x}_k)\right]\right)$$

$$=\frac{1}{N_x}\left(\mathrm{Var}_{\bm{x}}\left[\mathcal{I}(\theta_i\,|\,\bm{x})\right]+\underset{\bm{x}}{\mathbb{E}}\left[\mathcal{V}_j(\theta_i\,|\,\bm{x})\right]\right).$$

As required.

**For $\mathcal{V}_1(\bm{\theta})$**

*Proof.*

$$\mathcal{V}_1(\theta_i)=\frac{1}{N}\left(\underset{p(\bm{x},\bm{y})}{\mathbb{E}}\left[\left(\frac{\partial\log p(\bm{y}\,|\,\bm{x})}{\partial\theta_i}\right)^2\right]-\underset{p(\bm{x},\bm{y})}{\mathbb{E}}\left[\frac{\partial\log p(\bm{y}\,|\,\bm{x})}{\partial\theta_i}\right]^2\right).$$

Let $\bm{\delta}_a(\bm{x},\bm{y})\doteq(\bm{t}(\bm{y})-\bm{\eta}(\bm{x}))$.

$$\underset{p(\bm{x},\bm{y})}{\mathbb{E}}\left[\left(\frac{\partial\log p(\bm{y}\,|\,\bm{x})}{\partial\theta_i}\right)^2\right]$$

$$=\underset{p(\bm{x},\bm{y})}{\mathbb{E}}\left[\frac{\partial\bm{h}^a(\bm{x})}{\partial\theta_i}\frac{\partial\bm{h}^b(\bm{x})}{\partial\theta_i}\frac{\partial\bm{h}^c(\bm{x})}{\partial\theta_i}\frac{\partial\bm{h}^d(\bm{x})}{\partial\theta_i}\bm{\delta}_a(\bm{x},\bm{y})\bm{\delta}_b(\bm{x},\bm{y})\bm{\delta}_c(\bm{x},\bm{y})\bm{\delta}_d(\bm{x},\bm{y})\right]$$

$$=\underset{q(\bm{x})}{\mathbb{E}}\left[\frac{\partial\bm{h}^a(\bm{x})}{\partial\theta_i}\frac{\partial\bm{h}^b(\bm{x})}{\partial\theta_i}\frac{\partial\bm{h}^c(\bm{x})}{\partial\theta_i}\frac{\partial\bm{h}^d(\bm{x})}{\partial\theta_i}\underset{p(\bm{y}\,|\,\bm{x})}{\mathbb{E}}\left[\bm{\delta}_a(\bm{x},\bm{y})\bm{\delta}_b(\bm{x},\bm{y})\bm{\delta}_c(\bm{x},\bm{y})\bm{\delta}_d(\bm{x},\bm{y})\right]\right]$$

$$=\underset{q(\bm{x})}{\mathbb{E}}\left[\frac{\partial\bm{h}^a(\bm{x})}{\partial\theta_i}\frac{\partial\bm{h}^b(\bm{x})}{\partial\theta_i}\frac{\partial\bm{h}^c(\bm{x})}{\partial\theta_i}\frac{\partial\bm{h}^d(\bm{x})}{\partial\theta_i}\mathcal{K}_{abcd}^p(\bm{t}\,|\,\bm{x})\right]$$

And:

$$\mathbb{E}_{p(\boldsymbol{x},\boldsymbol{y})}\left[\frac{\partial \log p(\boldsymbol{y}\mid\boldsymbol{x})}{\partial \theta_i}\right]^2$$

$$= \mathbb{E}_{p(\boldsymbol{x},\boldsymbol{y})}\left[\frac{\partial \boldsymbol{h}^a(\boldsymbol{x})}{\partial \theta_i}\frac{\partial \boldsymbol{h}^b(\boldsymbol{x})}{\partial \theta_i}\boldsymbol{\delta}_a(\boldsymbol{x},\boldsymbol{y})\boldsymbol{\delta}_b(\boldsymbol{x},\boldsymbol{y})\right]^2$$

$$= \mathbb{E}_{q(\boldsymbol{x})}\left[\frac{\partial \boldsymbol{h}^a(\boldsymbol{x})}{\partial \theta_i}\frac{\partial \boldsymbol{h}^b(\boldsymbol{x})}{\partial \theta_i}\mathbb{E}_{p(\boldsymbol{y}\mid\boldsymbol{x})}\left[\boldsymbol{\delta}_a(\boldsymbol{x},\boldsymbol{y})\boldsymbol{\delta}_b(\boldsymbol{x},\boldsymbol{y})\right]\right]^2$$

$$= \mathbb{E}_{q(\boldsymbol{x})}\left[\frac{\partial \boldsymbol{h}^a(\boldsymbol{x})}{\partial \theta_i}\frac{\partial \boldsymbol{h}^b(\boldsymbol{x})}{\partial \theta_i}\mathcal{I}_{ab}(\boldsymbol{h}\mid\boldsymbol{x})\right]^2$$

$$= \mathbb{E}_{q(\boldsymbol{x})}\left[\left(\frac{\partial \boldsymbol{h}^a(\boldsymbol{x})}{\partial \theta_i}\frac{\partial \boldsymbol{h}^b(\boldsymbol{x})}{\partial \theta_i}\mathcal{I}_{ab}(\boldsymbol{h}\mid\boldsymbol{x})\right)^2\right] - \mathrm{Var}_{\mathsf{X}}\left(\left(\frac{\partial \boldsymbol{h}(\boldsymbol{x})}{\partial \theta_i}\right)^\top \mathcal{I}(\boldsymbol{h}\mid\boldsymbol{x})\frac{\partial \boldsymbol{h}(\boldsymbol{x})}{\partial \theta_i}\right)$$

$$= \mathbb{E}_{q(\boldsymbol{x})}\left[\left(\frac{\partial \boldsymbol{h}^a(\boldsymbol{x})}{\partial \theta_i}\frac{\partial \boldsymbol{h}^b(\boldsymbol{x})}{\partial \theta_i}\mathcal{I}_{ab}(\boldsymbol{h}\mid\boldsymbol{x})\right)^2\right] - \mathrm{Var}_{\mathsf{X}}\left(\mathcal{I}(\theta_i\mid\boldsymbol{x})\right).$$

Together:

$$\mathcal{V}_1(\theta_i) = \frac{1}{N}\mathrm{Var}_{\mathsf{X}}\left(\mathcal{I}(\theta_i\mid\boldsymbol{x})\right)$$

$$+ \frac{1}{N}\mathbb{E}_{q(\boldsymbol{x})}\left[\frac{\partial \boldsymbol{h}^a(\boldsymbol{x})}{\partial \theta_i}\frac{\partial \boldsymbol{h}^b(\boldsymbol{x})}{\partial \theta_i}\frac{\partial \boldsymbol{h}^c(\boldsymbol{x})}{\partial \theta_i}\frac{\partial \boldsymbol{h}^d(\boldsymbol{x})}{\partial \theta_i}\left[\mathcal{K}^p_{abcd}(\boldsymbol{t}\mid\boldsymbol{x}) - \mathcal{I}_{ab}(\boldsymbol{h}\mid\boldsymbol{x})\cdot\mathcal{I}_{cd}(\boldsymbol{h}\mid\boldsymbol{x})\right]\right]$$

$$\square$$

**For $\mathcal{V}_2(\boldsymbol{\theta})$**

*Proof.*

$$\mathcal{V}_2(\theta_i) = \frac{1}{N}\mathrm{Var}\left(\left(\boldsymbol{\eta}_a(\boldsymbol{x}) - \boldsymbol{t}_a(\boldsymbol{y})\right)\frac{\partial^2 \boldsymbol{h}^a(\boldsymbol{x})}{\partial \theta_i \partial \theta_i} + \mathcal{I}(\theta_i\mid\boldsymbol{x})\right)$$

$$= \frac{1}{N}\left[\underbrace{\mathrm{Var}\left(\left(\boldsymbol{\eta}_a(\boldsymbol{x}) - \boldsymbol{t}_a(\boldsymbol{y})\right)\frac{\partial^2 \boldsymbol{h}^a(\boldsymbol{x})}{\partial \theta_i \partial \theta_i}\right)}_{(a)} + \mathrm{Var}\left(\mathcal{I}(\theta_i\mid\boldsymbol{x})\right)\right.$$

$$\left.+ 2\underbrace{\mathrm{Cov}\left(\left(\boldsymbol{\eta}_a(\boldsymbol{x}) - \boldsymbol{t}_a(\boldsymbol{y})\right)\frac{\partial^2 \boldsymbol{h}^a(\boldsymbol{x})}{\partial \theta_i \partial \theta_i}, \mathcal{I}(\theta_i\mid\boldsymbol{x})\right)}_{(b)}\right].$$

$$(a) = \text{Var}\left((\boldsymbol{\eta}_a(\boldsymbol{x}) - \boldsymbol{t}_a(\boldsymbol{y})) \frac{\partial^2 \boldsymbol{h}^a(\boldsymbol{x})}{\partial \theta_i \partial \theta_i}\right)$$

$$= \mathbb{E}_{p(\boldsymbol{x},\boldsymbol{y})}\left[\left((\boldsymbol{\eta}_a(\boldsymbol{x}) - \boldsymbol{t}_a(\boldsymbol{y})) \frac{\partial^2 \boldsymbol{h}^a(\boldsymbol{x})}{\partial \theta_i \partial \theta_i}\right)^2\right]$$

$$- \mathbb{E}_{p(\boldsymbol{x},\boldsymbol{y})}\left[(\boldsymbol{\eta}_a(\boldsymbol{x}) - \boldsymbol{t}_a(\boldsymbol{y})) \frac{\partial^2 \boldsymbol{h}^a(\boldsymbol{x})}{\partial \theta_i \partial \theta_i}\right]^2$$

$$= \mathbb{E}_{p(\boldsymbol{x},\boldsymbol{y})}\left[\left((\boldsymbol{\eta}_a(\boldsymbol{x}) - \boldsymbol{t}_a(\boldsymbol{y})) \frac{\partial^2 \boldsymbol{h}^a(\boldsymbol{x})}{\partial \theta_i \partial \theta_i}\right)^2\right]$$

$$- \mathbb{E}_{q(\boldsymbol{x})}\left[\frac{\partial^2 \boldsymbol{h}^a(\boldsymbol{x})}{\partial \theta_i \partial \theta_i} \mathbb{E}_{p(\boldsymbol{y}\,|\,\boldsymbol{x})}\left[(\boldsymbol{\eta}_a(\boldsymbol{x}) - \boldsymbol{t}_a(\boldsymbol{y}))\right]\right]^2$$

$$= \mathbb{E}_{p(\boldsymbol{x},\boldsymbol{y})}\left[\left((\boldsymbol{\eta}_a(\boldsymbol{x}) - \boldsymbol{t}_a(\boldsymbol{y})) \frac{\partial^2 \boldsymbol{h}^a(\boldsymbol{x})}{\partial \theta_i \partial \theta_i}\right)^2\right] - 0$$

$$= \mathbb{E}_{q(\boldsymbol{x})}\left[\left(\frac{\partial^2 \boldsymbol{h}(\boldsymbol{x})}{\partial \theta_i \partial \theta_i}\right)^\top \mathbb{E}_{p(\boldsymbol{y}\,|\,\boldsymbol{x})}\left[(\boldsymbol{\eta}(\boldsymbol{x}) - \boldsymbol{t}(\boldsymbol{y}))(\boldsymbol{\eta}(\boldsymbol{x}) - \boldsymbol{t}(\boldsymbol{y}))^\top\right]\left(\frac{\partial^2 \boldsymbol{h}(\boldsymbol{x})}{\partial \theta_i \partial \theta_i}\right)\right]$$

$$= \mathbb{E}_{q(\boldsymbol{x})}\left[\left(\frac{\partial^2 \boldsymbol{h}(\boldsymbol{x})}{\partial \theta_i \partial \theta_i}\right)^\top \mathcal{I}(\boldsymbol{h}\,|\,\boldsymbol{x})\left(\frac{\partial^2 \boldsymbol{h}(\boldsymbol{x})}{\partial \theta_i \partial \theta_i}\right)\right].$$

$$(b) = \text{Cov}\left((\boldsymbol{\eta}_a(\boldsymbol{x}) - \boldsymbol{t}_a(\boldsymbol{y})) \frac{\partial^2 \boldsymbol{h}^a(\boldsymbol{x})}{\partial \theta_i \partial \theta_i}, \mathcal{I}(\theta_i\,|\,\boldsymbol{x})\right)$$

$$= \mathbb{E}_{p(\boldsymbol{x},\boldsymbol{y})}\left[(\boldsymbol{\eta}_a(\boldsymbol{x}) - \boldsymbol{t}_a(\boldsymbol{y})) \frac{\partial^2 \boldsymbol{h}^a(\boldsymbol{x})}{\partial \theta_i \partial \theta_i}\mathcal{I}(\theta_i\,|\,\boldsymbol{x})\right]$$

$$- \mathbb{E}_{p(\boldsymbol{x},\boldsymbol{y})}\left[(\boldsymbol{\eta}_a(\boldsymbol{x}) - \boldsymbol{t}_a(\boldsymbol{y})) \frac{\partial^2 \boldsymbol{h}^a(\boldsymbol{x})}{\partial \theta_i \partial \theta_i}\right]\mathbb{E}_{p(\boldsymbol{x},\boldsymbol{y})}\left[\mathcal{I}(\theta_i\,|\,\boldsymbol{x})\right]$$

$$= 0,$$

which follows by taking the 'partial' expectation $(\boldsymbol{y}\,|\,\boldsymbol{x})$ for both terms.

Thus together,

$$\mathcal{V}_2(\theta_i) = \frac{1}{N}\text{Var}\left(\mathcal{I}(\theta_i\,|\,\boldsymbol{x})\right) + \frac{1}{N}\mathbb{E}_{q(\boldsymbol{x})}\left[\left(\frac{\partial^2 \boldsymbol{h}(\boldsymbol{x})}{\partial \theta_i \partial \theta_i}\right)^\top \mathcal{I}(\boldsymbol{h}\,|\,\boldsymbol{x})\left(\frac{\partial^2 \boldsymbol{h}(\boldsymbol{x})}{\partial \theta_i \partial \theta_i}\right)\right].$$

$\square$

# P    Proof of Lemma 4.8

*Proof.* The lower bound holds from just considering the non-negativity of variance. For the upper bound, we utilize the bound directly consider the bounds of Eq. (7),

$$\text{Var}\left(\mathcal{I}(\theta_i\,|\,\boldsymbol{x})\right) = \mathbb{E}_{q(\boldsymbol{x})}\left[\mathcal{I}(\theta_i\,|\,\boldsymbol{x})^2\right] - \mathbb{E}_{q(\boldsymbol{x})}\left[\mathcal{I}(\theta_i\,|\,\boldsymbol{x})\right]^2$$

$$\leq \mathbb{E}_{q(\boldsymbol{x})}\left[\mathcal{I}(\theta_i\,|\,\boldsymbol{x})^2\right]$$

$$\leq \mathbb{E}_{q(\boldsymbol{x})}\left[\|\partial_i \boldsymbol{h}(\boldsymbol{x})\|_2^4 \cdot \lambda_{\max}^2(\mathcal{I}(\boldsymbol{h}\,|\,\boldsymbol{x}))\right].$$

$\square$

# Q   Proof of Proposition 5.1

We first derive the statistics $\mathcal{I}(\boldsymbol{h} \,|\, \boldsymbol{x})$ and $\mathcal{K}^p(\boldsymbol{t} \,|\, \boldsymbol{x})$ presented in "Regression: Isotropic Gaussian Distribution" Section 5. It follows that from the regression setting, we have that,

$$F(\boldsymbol{h}(\boldsymbol{x})) = \log \int \pi(\boldsymbol{y}) \cdot \exp(\boldsymbol{t}^\top(\boldsymbol{y})\boldsymbol{h}(\boldsymbol{x}))$$

$$= \log \int \pi(\boldsymbol{y}) \cdot \exp(\boldsymbol{y}^\top \boldsymbol{h}(\boldsymbol{x})),$$

where notably, by definition, $\pi(\boldsymbol{y})$ is independent of learned parameter $\boldsymbol{h}(\boldsymbol{x})$.

As such, we have that:

$$\frac{\partial}{\partial \boldsymbol{h}_i} F(\boldsymbol{h}) \bigg|_{\boldsymbol{h}=\boldsymbol{h}(\boldsymbol{x})} = \frac{1}{\int \pi(\boldsymbol{y}) \cdot \exp(\boldsymbol{t}^\top(\boldsymbol{y})\boldsymbol{h}(\boldsymbol{x}))} \cdot \int \pi(\boldsymbol{y}) \cdot \exp(\boldsymbol{t}^\top(\boldsymbol{y})\boldsymbol{h}(\boldsymbol{x})) \cdot \boldsymbol{h}_i(\boldsymbol{x}) = \underset{p(\boldsymbol{y}\,|\,\boldsymbol{x})}{\mathbb{E}} [\boldsymbol{y}_i].$$

Now we note that $\mathbb{E}_p(\boldsymbol{y} \,|\, \boldsymbol{x})[\boldsymbol{y}_i]$ is exactly $\boldsymbol{h}_i(\boldsymbol{x})$ as the parameter $\boldsymbol{h}(\boldsymbol{x})$ specifies the mean of the (isotropic) multivariate normal distribution. As such we have that,

$$\frac{\partial}{\partial \boldsymbol{h}_i} F(\boldsymbol{h}) \bigg|_{\boldsymbol{h}=\boldsymbol{h}(\boldsymbol{x})} = \boldsymbol{h}(\boldsymbol{x})$$

$$\mathcal{I}(\boldsymbol{h} \,|\, \boldsymbol{x}) = \frac{\partial^2}{\partial \boldsymbol{h} \partial \boldsymbol{h}^\top} F(\boldsymbol{h}) \bigg|_{\boldsymbol{h}=\boldsymbol{h}(\boldsymbol{x})} = I.$$

Furthermore, by [37, Lemma 5], we have that,

$$\mathcal{K}^p_{abcd}(\boldsymbol{t} \,|\, \boldsymbol{x}) = \frac{\partial^4 F(\boldsymbol{h})}{\partial \boldsymbol{h}_a \partial \boldsymbol{h}_b \partial \boldsymbol{h}_c \partial \boldsymbol{h}_d} \bigg|_{\boldsymbol{h}=\boldsymbol{h}(\boldsymbol{x})}$$
$$+ \mathcal{I}_{ab}(\boldsymbol{h} \,|\, \boldsymbol{x}) \cdot \mathcal{I}_{cd}(\boldsymbol{h} \,|\, \boldsymbol{x}) + \mathcal{I}_{ac}(\boldsymbol{h} \,|\, \boldsymbol{x}) \cdot \mathcal{I}_{bd}(\boldsymbol{h} \,|\, \boldsymbol{x}) + \mathcal{I}_{ad}(\boldsymbol{h} \,|\, \boldsymbol{x}) \cdot \mathcal{I}_{bc}(\boldsymbol{h} \,|\, \boldsymbol{x})$$
$$= 0 + \mathcal{I}_{ab}(\boldsymbol{h} \,|\, \boldsymbol{x}) \cdot \mathcal{I}_{cd}(\boldsymbol{h} \,|\, \boldsymbol{x}) + \mathcal{I}_{ac}(\boldsymbol{h} \,|\, \boldsymbol{x}) \cdot \mathcal{I}_{bd}(\boldsymbol{h} \,|\, \boldsymbol{x}) + \mathcal{I}_{ad}(\boldsymbol{h} \,|\, \boldsymbol{x}) \cdot \mathcal{I}_{bc}(\boldsymbol{h} \,|\, \boldsymbol{x})$$
$$= \mathcal{I}_{ab}(\boldsymbol{h} \,|\, \boldsymbol{x}) \cdot \mathcal{I}_{cd}(\boldsymbol{h} \,|\, \boldsymbol{x}) + \mathcal{I}_{ac}(\boldsymbol{h} \,|\, \boldsymbol{x}) \cdot \mathcal{I}_{bd}(\boldsymbol{h} \,|\, \boldsymbol{x}) + \mathcal{I}_{ad}(\boldsymbol{h} \,|\, \boldsymbol{x}) \cdot \mathcal{I}_{bc}(\boldsymbol{h} \,|\, \boldsymbol{x}).$$

In summary, we have,

$$\mathcal{K}^p_{abcd}(\boldsymbol{t} \,|\, \boldsymbol{x}) = \mathcal{I}_{ab}(\boldsymbol{h} \,|\, \boldsymbol{x}) \cdot \mathcal{I}_{cd}(\boldsymbol{h} \,|\, \boldsymbol{x}) + \mathcal{I}_{ac}(\boldsymbol{h} \,|\, \boldsymbol{x}) \cdot \mathcal{I}_{bd}(\boldsymbol{h} \,|\, \boldsymbol{x})$$
$$+ \mathcal{I}_{ad}(\boldsymbol{h} \,|\, \boldsymbol{x}) \cdot \mathcal{I}_{bc}(\boldsymbol{h} \,|\, \boldsymbol{x})$$
$$(\mathcal{K}^p(\boldsymbol{t} \,|\, \boldsymbol{x}) - \mathcal{I}(\boldsymbol{h} \,|\, \boldsymbol{x}) \otimes \mathcal{I}(\boldsymbol{h} \,|\, \boldsymbol{x}))_{abcd} = \mathcal{I}_{ac}(\boldsymbol{h} \,|\, \boldsymbol{x}) \cdot \mathcal{I}_{bd}(\boldsymbol{h} \,|\, \boldsymbol{x}) + \mathcal{I}_{ad}(\boldsymbol{h} \,|\, \boldsymbol{x}) \cdot \mathcal{I}_{bc}(\boldsymbol{h} \,|\, \boldsymbol{x}).$$

*Proof.* The minimum and maximum eigenvalues of $\mathcal{I}(\boldsymbol{h} \,|\, \boldsymbol{x})$ follows directly noting that the trace of a matrix is the sum of eigenvalues. As such, from the statistics presented above we have that the minimum and eigenvalue must be $1$.

The tensor eigenvalues of $\mathcal{K}^p(\boldsymbol{t} \,|\, \boldsymbol{x}) - \mathcal{I}(\boldsymbol{h} \,|\, \boldsymbol{x}) \otimes \mathcal{I}(\boldsymbol{h} \,|\, \boldsymbol{x}) = \mathcal{I}_{ac}(\boldsymbol{h} \,|\, \boldsymbol{x}) \cdot \mathcal{I}_{bd}(\boldsymbol{h} \,|\, \boldsymbol{x}) + \mathcal{I}_{ad}(\boldsymbol{h} \,|\, \boldsymbol{x}) \cdot \mathcal{I}_{bc}(\boldsymbol{h} \,|\, \boldsymbol{x})$ follows from the variational definition Eq. (10). For instance, for the minimum eigenvalue,

$$\inf_{\boldsymbol{u}:\|\boldsymbol{u}\|_2=1} \boldsymbol{u}^a \boldsymbol{u}^b \boldsymbol{u}^c \boldsymbol{u}^d \left( \mathcal{I}_{ac}(\boldsymbol{h} \,|\, \boldsymbol{x}) \cdot \mathcal{I}_{bd}(\boldsymbol{h} \,|\, \boldsymbol{x}) + \mathcal{I}_{ad}(\boldsymbol{h} \,|\, \boldsymbol{x}) \cdot \mathcal{I}_{bc}(\boldsymbol{h} \,|\, \boldsymbol{x}) \right)$$

$$= 2 \cdot \inf_{\boldsymbol{u}:\|\boldsymbol{u}\|_2=1} \|\boldsymbol{u}\|_2^2$$

$$= 2.$$

The maximum eigenvalue is proven identically. $\qquad\square$

# R   Proof of Theorem 5.2

We first prove the following corollary which connects the maximum eigenvalues of $\mathcal{K}(\boldsymbol{t} \,|\, \boldsymbol{x})$ to the maximum eigenvalues of $\mathcal{I}(\boldsymbol{h} \,|\, \boldsymbol{x})$.

**Corollary R.1.** *Suppose that the exponential family in Eq.* (1) *is specified by a categorical distribution. Then,*

$$\tilde{\lambda}_{\max}\left(\mathcal{K}(\boldsymbol{t}\,|\,\boldsymbol{x})\right) \leq 2 \cdot \lambda_{\max}(\mathcal{I}(\boldsymbol{h}\,|\,\boldsymbol{x})). \tag{31}$$

*Proof.* As we are consider a categorical distribution we have that

$$\boldsymbol{v}_i = \begin{cases} 1 - \sigma_i(\boldsymbol{h}) & \text{if } y = i \\ 0 - \sigma_i(\boldsymbol{h}) & \text{if } y \neq i \end{cases}.$$

Thus we have that $|\boldsymbol{v}_i| \leq 1$. Furthermore, note that the maximum $\ell_2$-norm that we can have is $\|\boldsymbol{v}\|_2 \leq \sqrt{2}$. Note that this is tight when the positive and negative mass are placed only two distinct coordinates, *i.e.*, $(-1, 1, \ldots)$.

Thus using Proposition 4.3, the result follows. □

Now by using Corollaries 4.2 and R.1, the remainder of the proof, all we require is the bounding of $\lambda_{\max}(\mathcal{I}(\boldsymbol{h}\,|\,\boldsymbol{x}))$.

*Proof.* The first term in the maximum eigenvalue follows from,

$$\begin{aligned}
\lambda_{\max}(\mathcal{I}(\boldsymbol{h}\,|\,\boldsymbol{x})) &= \lambda_{\max}\left(\mathrm{Diag}(\sigma(\boldsymbol{x})) - \sigma(\boldsymbol{x})\sigma(\boldsymbol{x})^{\top}\right) \\
&\leq \lambda_{\max}\left(\mathrm{Diag}(\sigma(\boldsymbol{x}))\right) - \lambda_{\min}\left(\sigma(\boldsymbol{x})\sigma(\boldsymbol{x})^{\top}\right) \\
&= \max_k \sigma_k(\boldsymbol{x}).
\end{aligned}$$

The second term in the maximum follows from the trace of $\mathcal{I}(\boldsymbol{h}\,|\,\boldsymbol{x})$ being the sum of total eigenvalues. □

## S   Proof of Lemma 6.1

*Proof.* The proof follows from the standard definition of covariance. Denoting $\hat{\boldsymbol{\eta}}(\boldsymbol{x}) \doteq \mathbb{E}_{q(\boldsymbol{y}\,|\,\boldsymbol{x})}[\boldsymbol{t}(\boldsymbol{y})]$, we have:

$$\mathrm{Cov}^q(\boldsymbol{t}\,|\,\boldsymbol{x}) = \underset{q(\boldsymbol{y}\,|\,\boldsymbol{x})}{\mathbb{E}}\left[\boldsymbol{t}(\boldsymbol{y})\boldsymbol{t}^{\top}(\boldsymbol{y})\right] - \hat{\boldsymbol{\eta}}(\boldsymbol{x})\hat{\boldsymbol{\eta}}^{\top}(\boldsymbol{x}).$$

Also expanding $\mathrm{I}(\boldsymbol{h}\,|\,\boldsymbol{x})$:

$$\begin{aligned}
\mathrm{I}(\boldsymbol{h}\,|\,\boldsymbol{x}) &= \underset{q(\boldsymbol{y}\,|\,\boldsymbol{x})}{\mathbb{E}}\left[(\boldsymbol{t}(\boldsymbol{y}) - \boldsymbol{\eta}(\boldsymbol{x}))(\boldsymbol{t}(\boldsymbol{y}) - \boldsymbol{\eta}(\boldsymbol{x}))^{\top}\right] \\
&= \underset{q(\boldsymbol{y}\,|\,\boldsymbol{x})}{\mathbb{E}}\left[\boldsymbol{t}(\boldsymbol{y})\boldsymbol{t}^{\top}(\boldsymbol{y})\right] - \boldsymbol{\eta}(\boldsymbol{x})\hat{\boldsymbol{\eta}}^{\top}(\boldsymbol{x}) - \hat{\boldsymbol{\eta}}^{\top}(\boldsymbol{x})\boldsymbol{\eta}^{\top}(\boldsymbol{x}) + \boldsymbol{\eta}(\boldsymbol{x})\boldsymbol{\eta}^{\top}(\boldsymbol{x}).
\end{aligned}$$

Thus we have

$$\begin{aligned}
\mathrm{Cov}^q(\boldsymbol{t}\,|\,\boldsymbol{x}) &= \mathrm{I}(\boldsymbol{h}\,|\,\boldsymbol{x}) + \boldsymbol{\eta}(\boldsymbol{x})\hat{\boldsymbol{\eta}}^{\top}(\boldsymbol{x}) + \hat{\boldsymbol{\eta}}^{\top}(\boldsymbol{x})\boldsymbol{\eta}^{\top}(\boldsymbol{x}) - \boldsymbol{\eta}(\boldsymbol{x})\boldsymbol{\eta}^{\top}(\boldsymbol{x}) - \hat{\boldsymbol{\eta}}(\boldsymbol{x})\hat{\boldsymbol{\eta}}^{\top}(\boldsymbol{x}) \\
&= \mathrm{I}(\boldsymbol{h}\,|\,\boldsymbol{x}) - (\boldsymbol{\eta}(\boldsymbol{x}) - \hat{\boldsymbol{\eta}}(\boldsymbol{x}))(\boldsymbol{\eta}(\boldsymbol{x}) - \hat{\boldsymbol{\eta}}(\boldsymbol{x}))^{\top}.
\end{aligned}$$

As required. □

## T   Proof of Corollary G.1

*Proof.* We calculate the variance:

$$\mathrm{V}(\theta_i) = \frac{1}{N}\left(\underset{q(\boldsymbol{y}\,|\,\boldsymbol{x})}{\mathbb{E}}\left[\left(\frac{\partial \log p(\boldsymbol{y}\,|\,\boldsymbol{x})}{\partial \theta_i}\right)^2\right] - \underset{q(\boldsymbol{x},\boldsymbol{y})}{\mathbb{E}}\left[\frac{\partial \log q(\boldsymbol{y}\,|\,\boldsymbol{x})}{\partial \theta_i}\right]^2\right).$$

Each of the terms can be calculated: Let $\boldsymbol{\delta}_a(\boldsymbol{x}, \boldsymbol{y}) \doteq (\boldsymbol{t}(\boldsymbol{y}) - \boldsymbol{\eta}(\boldsymbol{x}))$.

$$
\mathop{\mathbb{E}}_{p(\boldsymbol{x}, \boldsymbol{y})} \left[ \left( \frac{\partial \log p(\boldsymbol{y} \mid \boldsymbol{x})}{\partial \theta_i} \right)^2 \right]
$$

$$
= \mathop{\mathbb{E}}_{q(\boldsymbol{y} \mid \boldsymbol{x})} \left[ \frac{\partial \boldsymbol{h}^a(\boldsymbol{x})}{\partial \theta_i} \frac{\partial \boldsymbol{h}^b(\boldsymbol{x})}{\partial \theta_i} \frac{\partial \boldsymbol{h}^c(\boldsymbol{x})}{\partial \theta_i} \frac{\partial \boldsymbol{h}^d(\boldsymbol{x})}{\partial \theta_i} \boldsymbol{\delta}_a(\boldsymbol{x}, \boldsymbol{y}) \boldsymbol{\delta}_b(\boldsymbol{x}, \boldsymbol{y}) \boldsymbol{\delta}_c(\boldsymbol{x}, \boldsymbol{y}) \boldsymbol{\delta}_d(\boldsymbol{x}, \boldsymbol{y}) \right]
$$

$$
= \frac{\partial \boldsymbol{h}^a(\boldsymbol{x})}{\partial \theta_i} \frac{\partial \boldsymbol{h}^b(\boldsymbol{x})}{\partial \theta_i} \frac{\partial \boldsymbol{h}^c(\boldsymbol{x})}{\partial \theta_i} \frac{\partial \boldsymbol{h}^d(\boldsymbol{x})}{\partial \theta_i} \mathop{\mathbb{E}}_{q(\boldsymbol{y} \mid \boldsymbol{x})} \left[ \boldsymbol{\delta}_a(\boldsymbol{x}, \boldsymbol{y}) \boldsymbol{\delta}_b(\boldsymbol{x}, \boldsymbol{y}) \boldsymbol{\delta}_c(\boldsymbol{x}, \boldsymbol{y}) \boldsymbol{\delta}_d(\boldsymbol{x}, \boldsymbol{y}) \right]
$$

$$
= \frac{\partial \boldsymbol{h}^a(\boldsymbol{x})}{\partial \theta_i} \frac{\partial \boldsymbol{h}^b(\boldsymbol{x})}{\partial \theta_i} \frac{\partial \boldsymbol{h}^c(\boldsymbol{x})}{\partial \theta_i} \frac{\partial \boldsymbol{h}^d(\boldsymbol{x})}{\partial \theta_i} \mathrm{K}_{abcd}(\boldsymbol{t} \mid \boldsymbol{x}).
$$

And:

$$
\mathop{\mathbb{E}}_{p(\boldsymbol{y} \mid \boldsymbol{x})} \left[ \frac{\partial \log p(\boldsymbol{y} \mid \boldsymbol{x})}{\partial \theta_i} \right]^2
$$

$$
= \mathop{\mathbb{E}}_{p(\boldsymbol{y} \mid \boldsymbol{x})} \left[ \frac{\partial \boldsymbol{h}^a(\boldsymbol{x})}{\partial \theta_i} \frac{\partial \boldsymbol{h}^b(\boldsymbol{x})}{\partial \theta_i} \boldsymbol{\delta}_a(\boldsymbol{x}, \boldsymbol{y}) \boldsymbol{\delta}_b(\boldsymbol{x}, \boldsymbol{y}) \right]^2
$$

$$
= \left[ \frac{\partial \boldsymbol{h}^a(\boldsymbol{x})}{\partial \theta_i} \frac{\partial \boldsymbol{h}^b(\boldsymbol{x})}{\partial \theta_i} \mathop{\mathbb{E}}_{p(\boldsymbol{y} \mid \boldsymbol{x})} \left[ \boldsymbol{\delta}_a(\boldsymbol{x}, \boldsymbol{y}) \boldsymbol{\delta}_b(\boldsymbol{x}, \boldsymbol{y}) \right] \right]^2
$$

$$
= \left[ \frac{\partial \boldsymbol{h}^a(\boldsymbol{x})}{\partial \theta_i} \frac{\partial \boldsymbol{h}^b(\boldsymbol{x})}{\partial \theta_i} \mathrm{I}_{ab}(\boldsymbol{h} \mid x) \right]^2.
$$

Together with Lemma 6.1 proves the theorem. $\qquad \square$

## U   Proof of Corollary G.2

*Proof.* The proof follows identically to that of Theorem 4.7 with densities changed. $\qquad \square$

