# OpenReview forum: "Trade-Offs of Diagonal Fisher Information Matrix Estimators"
_NeurIPS.cc/2024/Conference — NeurIPS 2024 poster_

### Official Review · Reviewer_QMbw · 2024-07-03

**Soundness:** 4
**Presentation:** 3
**Contribution:** 2
**Rating:** 5
**Confidence:** 3

**Summary:**

The authors study two popular estimators of the Fisher information matrix with respect to neural network parameters. They derive upper and lower bounds for the variance of these estimators and showcase them in applications to regression and classification problems.

**Strengths:**

- Analyzing the convergence properties of Fisher information estimators is a timely and nontrivial endeavor, in part due to their use in natural gradient descent.
- The paper is very extensive and mathematically sound.

**Weaknesses:**

- The paper is highly technical and the numerical examples are fairly toy.
- The paper takes strong inspiration from a previous work [Soen, Alexander, and Ke Sun. "On the variance of the Fisher information for deep learning." Advances in Neural Information Processing Systems 34 (2021): 5708-5719] on the topic. As such, I question the value the present submission adds to this line of research. In particular, I think the submission does a poor job in highlighting the central novel aspects of their work and reviewing the field.

**Questions:**

- The authors should clarify clearly to what extent their submission improves upon previous works, particularly [Soen, Alexander, and Ke Sun. "On the variance of the Fisher information for deep learning." Advances in Neural Information Processing Systems 34 (2021): 5708-5719]. How do their bounds compare to the ones previously derived?

- In my opinion, the impact of the submission would be substantially improved if the authors state more clearly in what scenarios which of the two estimators will be preferred and provide a rough guideline for practitioners.

- Could the authors provide a numerical example for regression similar to Fig. 2?

**Limitations:**

There is no further need for authors to address limitations.

---

> ### Author Rebuttal · Authors · 2024-08-07
>
> Thank you for recognizing our contributions and their soundness.
>
> > Re: *Significance vs [37]*
>
> Please see our global response to all reviewers.
>
> > Re: *Better clarity on which estimator is preferred*
>
> Through our analysis, We have identified certain scenarios where one estimator
> is superior to the other.
>
> For example, ReLU networks can only apply $\\hat{\\mathcal{I}}_1$. As another
> example, $\\hat{\\mathcal{I}}_2$ has zero variance in the last layer and is
> always preferable than $\\hat{\\mathcal{I}}_1$. Similarly, in the second last
> layer, $\\hat{\\mathcal{I}}_2$ has a simple closed form and preferable for
> neurons in their linear regions (see Remark 4.5).
>
> In general, Theorem 4.1 suggests that the variance of both estimators depend on
> three factors: (1) the number of samples; (2) the derivatives of the neural
> network output wrt its parameters; (3) moments of the sufficient statistics
> $\\mathbf{t}(\\mathbf{y})$ (see Table 1). One has to incorporate these factors
> based on their specific neural network and settings, to decide which estimator
> to use. We agree with the reviewer to recall these points in the conclusion
> section.
>
> > Re: *Numerical examples are fairly toy*
>
> Our contributions are mainly theoretical and do not depend on the empirical
> results. In the paper and appendix, we use MNIST MLP with different activations
> (sigmoid and log-sigmoid) and different number of layers (4 and 5) to showcase
> the bounds of the variance and how the variance evolves over training time in
> different layers of the neural network. These numerical examples mainly serves
> the purpose of providing intuition. We do agree with the reviewer that more
> extensive empirical studies are meaningful as future work.
>
> > Re: *Numeric example for regression similar to Fig (2)*
>
> We thank the reviewer for this suggestion. As remarked by Proposition 5.1, for
> regression, all bounds in Theorem 4.1 becomes equalities. Therefore it is less
> meaningful to examine the case of regression, where all interested quantities
> $\\mathcal{I}$, $\\mathcal{V}_1$, and $\\mathcal{V}_2$ reduce to evaluation of the
> Jacobian/Hessian of the parameter-output mapping. The limited time for the
> rebuttal does not allow for redesigning the pipeline of FIM estimation on a new
> dataset and learning task. We therefore hesitate to overstate our empirical
> discoveries or to promise experimental extensions without clearly understanding
> their implications.
>
> We would also like to iterate that aim of this paper's contribution is
> primarily theoretical. All though additional numerical experiments would be
> nice, a thorough independent empirical study is out of this paper's scope and
> is arguably would make an independent piece of work (examining various
> settings, optimizers, and architectures etc).

---

> > ### Comment · Reviewer_QMbw · 2024-08-09
> >
> > I thank the authors for their concise rebuttal and have updated my score accordingly.

---

### Official Review · Reviewer_qMTe · 2024-07-09

**Soundness:** 4
**Presentation:** 3
**Contribution:** 3
**Rating:** 8
**Confidence:** 5

**Summary:**

The paper analyses two estimators of the Fisher Information matrix and specifically their variances, which (from reference 37) have closed-form but non-practical expressions. The authors show a sequence of inequalities and derive practical bounds for these variances, both element-wise and trace-wise, both conditioned to x and not. Several remarks and trade-offs on these bounds clarify when an estimator is preferable over the other.
The results hold in the general exponential family likelihood but are also further concretized for regression and classification.

**Strengths:**

The paper is overall very well written and structured.

Notation is clear, results are properly presented and every step is further clarified in the appendix. \
The alternation between formal statements and explanations is great and makes the paper vey pleasant to read. I particularly like the amount of remarks and observations that discuss specific terms of the equations, explains them and connect them to each other. And the same is true for all the highlighted trade-offs (for example line 212-215)

**Weaknesses:**

I have found no major weaknesses in the paper. I report some minor things.

I personally dislike the notation $\hat{\mathcal{I}}_1(\theta_i)$ for the i-th element of the diagonal introduced in line 87. This notation suggest that this term only depends on $\theta_i$, but it actually depends on the whole $\theta$ vector. On the other hand, this is quite a light easy notations, and I can't think of alternatives which are not much heavier to read.

Also, I find the definition in line 120 "is defined by $\hat{\mathcal{I}}_j(\theta_i)$ with $x=x_1=...=x_N$" to be yes understandable but not super clean. The alternatives are not as compact, but I'd recommend using the extra space to clarify this a bit better. At least you should write that $y_1,...,y_N$ are i.i.d. from $p(y|x;\theta)$ and not from $p(y,x;\theta)$ as in $\hat{\mathcal{I}}_j(\theta_i)$.

It would be nice to explicitly write the proof for the variance closed expression in Lemma B.1, yet clarifying that it's not a paper contribution. This is for two reasons: (1) it would make the paper self contained, without the need for the reader to read reference 37 and (2) the notation is slightly different and it would be easier for the reader to be consistent.

**Questions:**

In line 168. Shouldn't "of Eqs. (4) and (6)" instead be "of Eqs. (4) and (5)"? And consistently shouldn't "$\mathcal{V}_1(\theta_i|x)$" be "$\mathcal{V}_2(\theta_i|x)$"? \
I ask this because "small shifts in parameter space yield large changes in the output" to me refers to the magnitude of the network jacobian $||\partial_i h(x)||$, and that appear in the bound for $V_1$, not for $V_2$. Am I misunderstanding something or is this a typo?

In line 246. Isn't the bound sample complexity $\mathcal{O}(\frac{1}{N_x} + \frac{1}{N_y})$? I can't see why it should be the product, can you elaborate more on this derivation?

**Limitations:**

I think it would be nice to clarify better that the work is incremental on "Soen et all - On the variance of the Fisher information for deep learning" [reference 37] and their closed form expression reported in the appendix in Lemma B.1. (also it would be nice to move these closed form expressions to the main paper).

---

> ### Author Rebuttal · Authors · 2024-08-07
>
> Thank you for recognizing our contributions and praising our writing.
> We respond to your remarks and suggestions as below.
>
> > Re: *The notation* $\\hat{\\mathcal{I}}_{1}(\\theta_i)$
>
> We propose to explicitly define it as $\\hat{\\mathcal{I}}_{1}(\\theta_i) := (\\hat{\\mathcal{I}}_1(\\theta))\_{i i}$ at the first appearance of this notation in
> the beginning of Section 3, and add a sentence to remind the reader that
> $\\hat{\\mathcal{I}}_1(\\theta_i)$ is an abuse of notation and actually depends on
> the whole $\\mathbf{\\theta}$ vector.
>
> > Re: *The notation* $\\hat{\\mathcal{I}}_j(\\theta_i \\mid \\mathbf{x})$ *with*
> > $\\mathbf{x}=\\mathbf{x}_1=\\cdots=\\mathbf{x}_N$
>
> We will rewrite the sentence to clarify that $y_1,\\cdots,y_N$ are i.i.d. from
> $p(y \\mid \\mathbf{x}; \\mathbf{\\theta})$ as the reviewer has suggested.
>
> > Re: *Explicit proof of Lemma B.1 to make the paper self-contained*
>
> We agree to include such a proof that is consistent with our current notation.
>
> > Re: *Line 168, Eqs. (4) and (6), the scale of* $\\mathcal{I}(\\theta_i \\mid
> > \\mathbf{x})$ *and* $\\mathcal{V}_2(\\theta_i \\mid \\mathbf{x})$
>
> Yes, this is a typo and the corrections suggested by the reviewers are
> correct. Eqs. (4) and (5) are the bounds which depend on the network Jacobian.
>
> > Re: *Line 246, sample complexity should not be* $\\mathcal{O}(\\frac{1}{N_xN_y})$
>
> We thank the reviewer for pointing out this mistake. By the total variance
> decomposition in Eq. (13), the variance wrt $q(\\mathbf{x})$ scales inversely to
> $N_x$, the number of $\\mathbf{x}$-samples, and the variance wrt $p(y \\mid
> \\mathbf{x})$ scales inversely to $N_y$. Therefore the total variance is
> in the order of $\\mathcal{O}\\left(\\frac{1}{N_x}+\\frac{1}{N_y}\\right)$ as the
> reviewer suggested. We will make sure to correct this sentence and related
> places in this paragraph.
>
> > Re: *Relationship with reference [37]*
>
> Please see our global response. We shifted the closed-form expression of
> the variances to the appendix mainly due to space limitation. Should more space
> be permitted, we agree to improve our self-contained narrative by moving them
> back into the main paper.

---

> > ### Comment · Reviewer_qMTe · 2024-08-08
> > **Keep score**
> >
> > Thanks for the clarification. While I do agree with the other reviewers that the improvements made on top of [37] could have been clarified better, I do think that they are overall clear and valuable. Hence, I keep my score and strongly support acceptance.

---

### Official Review · Reviewer_wDHQ · 2024-07-10

**Soundness:** 2
**Presentation:** 2
**Contribution:** 2
**Rating:** 5
**Confidence:** 1

**Summary:**

In this paper, the authors analyzed two different estimators, $I_1(\theta)$ and  $I_2(\theta)$, for the diagonal elements
of the Fisher information matrix of a parametric neural net model. These diagonal elements are approximations for the entire matrix,
which is unfeasible to be calculated in real neural nets models. They identified the situations in which each of these estimators is preferable with respect to the other.

**Strengths:**

The Fisher information matrix is a major parameter driving the quality of the fit and highlighting important aspects of the interdependence between the parameters. It has been used in several papers of continual learning, the research area that I follow.
It focuses on the diagonal elements of the Fisher information matrix, offering approximations that are feasible to calculate in real neural net models, addressing a significant practical challenge given the unfeasibility of computing the entire matrix. he authors identified specific situations where each of the two estimators, $I_1(\theta)$ and  $I_2(\theta)$, is preferable. This practical guidance can help researchers and practitioners make informed decisions about which estimator to use in different scenarios.

**Weaknesses:**

This is a highly theoretical paper and I have difficulty following all the findings presented by the authors. One of the main problems with this paper is that it has a large intersection with Soen and Sun (2021) and this becomes clear while we read the paper. In the Related work, Soen and Sun (2021) is mentioned in passing and it is a major reference work for the current manuscript. It is worth to have a better explanation of the relative merits of these two papers. Both papers analyzed two different estimators, $I_1(\theta)$ and  $I_2(\theta)$, for the Fisher information matrix of a parametric neural net model. While the older paper focuses on the matrix as a whole, the present manuscript focuses on the diagonal elements. However, the work of Soen and Sun (2021) is repeatedly referred to during the technical development of the paper. The contribution of this paper seems marginal when compared to the previous paper but still relevant.

The results are derived assuming $\theta$ is equal to its true value. In practice, the maximum likelihood estimator (MLE) of
$\theta is used, which introduces additional variability. The paper does not adequately address the implications of this assumption or the consequences when using the MLE, which limits the practical applicability of the results.

The paper does not discuss the impact of model misspecification on the results. In real-world scenarios, the data might not follow the proposed model exactly, and the inverse-Fisher information matrix might not represent the correct asymptotic variance of MLE estimators. The authors should comment on how their findings would be affected in such cases.

**Questions:**

The results are all given using $\theta$ equal to its true value. In practice, we use the MLE estimator of $\theta$. The results
are not valid anymore as we have this additional source of variability. The authors may comment on the consequences of this?

**Limitations:**

This is a theoretical paper with no immediate connection to societal impact.

---

> ### Author Rebuttal · Authors · 2024-08-07
>
> Thank you for recognizing our strength and potential usefulness
> in application areas including continual learning.
>
> > Re: *Significance vs [37]*
>
> We kindly refer the reviewer to our global response.
>
> > Re: *Assumption of* $\\theta$ *being the true value*
>
> We clarify that our result holds for any $\\theta$ in the parameter space
> of neural networks, aka the neuromanifold. Our formal statements and
> their proofs do *not* depend on setting $\\theta$ to the true value or its MLE,
> nor to be "nearby" to the MLE in any sense.
> Similar to how the FIM $\\mathcal{I}(\\theta)$ is a PSD tensor when $\\theta$
> varies on the neuromanifold, our central subjects
> $\\hat{\\mathcal{I}}_1(\\theta)$, $\\hat{\\mathcal{I}}_2(\\theta)$,
> $\\mathcal{V}_1(\\theta)$, and $\\mathcal{V}_2(\\theta)$ are all tensors defined
> for any $\\theta$.
>
> > Re: *What are the effects of model miss-specification*
>
> Model miss-specification can be examined from the perspective of empirical data
> deviating from the joint model distribution $p(\\mathbf{x}, \\mathbf{y};
> \\mathbf{\\theta}) = q(\\mathbf{x})p(\\mathbf{y} \\mid \\mathbf{x}; \\mathbf{\\theta})$
> in Line 24. Our analysis does provide convenient tools to study this
> phenomenon.
>
> When the observed samples $\\mathbf{x}$ deviates from the "true"
> $q(\\mathbf{x})$, the error in estimating the FIM is described by the first term
> in Eq. (13): a high variance of the conditional FIM $\\mathcal{I}(\\theta_i \\mid
> \\mathbf{x})$ leads to a large error in estimating the FIM.
>
> On the other hand, when the observed $(\\mathbf{x},\\hat{\\mathbf{y}})$'s sampling
> deviates from the predictive model $p(\\mathbf{y} \\mid \\mathbf{x};
> \\mathbf{\\theta})$, data also plays a larger role.
> In this case, the empirical Fisher becomes a biased estimation of the FIM,
> which is described by Lemma 6.1 and its surrounding text. Our FIM estimators do
> not depends on the observed $\\hat{\\mathbf{y}}$ in the dataset and is not
> affected by such a bias. Note the difference in sampling of the labels
> $\\mathbf{y}$ / $\\hat{\\mathbf{y}}$ in the data FIM versus the FIM.
>
> When the model is well specified, the data FIM and FIM are equivalent (see Line
> 328).

---

> > ### Comment · Reviewer_wDHQ · 2024-08-13
> >
> > Thanks for responding to my questions. A better clarification of the contribution of this paper wrt [37] is appreciated.
> > I keep my positive evaluation and my score.

---

### Official Review · Reviewer_7Xmq · 2024-07-13

**Soundness:** 3
**Presentation:** 2
**Contribution:** 2
**Rating:** 6
**Confidence:** 2

**Summary:**

Summary
-------
The paper studies two estimators for estimating the diagonal of a Fisher information
matrix (FIM) of a parametric machine learning model. Both estimators are based on equivalent
expressions for the FIM. The first is the standard definition, expressed in terms of the
derivatives of the log likelihood, and the latter, expressed in terms of the second
derivative when the model is twice differentiable with respect to the parameters. The
estimators replace the expectation with an average over the data. The authors provide
bounds on the variance of both (unbiased) estimators in terms of the minimum/maximum
eigenvalues of the FIM, and then provide bounds on these eigenvalues under special cases.
They also provide a modest empirical evaluation.


Overall, while I think there is a fair amount of technical meat in the paper, I am
unconvinced about the problem. I will wait to hear from more expert reviewers in this
field before I make my decision.


Detailed comments:
------------------

While I am theoretically inclined, I am not an expert in this sub-field. Therefore, I will
keep my comments at a high level.

Significance of problem: I struggled to understand when estimating the FIM will be useful.
- One point raised by authors is that it is used in optimizing NNs. If this was true, it
  would have been nice to see the estimators used in NN optimization empirically, or a
  theoretical analysis showing how the bounds derived here percolate to bounds on
  optimization.
- Another explanation given is that it could be used to analyze the structure fo NNs.
  But this is stated vaguely, and it wasn't clear to me that such an analysis would help
  design better neural networks.

Significance of techniques: The proof in the appendix is substantial but the authors have
not stated which techniques used here are novel. Currently technical results are presented
one after the other. It might help the reader/reviewer
appreciate the contributions of the paper if the authors can highlight the novel tools
they used to derive these bounds.

Presentation and writing: The technical content was clear for the most part. However,
the writing was very equation-driven. I understand that this is in part due to the nature
of the paper, but the authors should simplify the exposition further. One option is to
cut down on some of the technical content.

**Strengths:**

See above

**Weaknesses:**

See above

**Questions:**

See above

**Limitations:**

See above

---

> ### Author Rebuttal · Authors · 2024-08-07
>
> We thank the reviewer for praising the technical developments of the paper.
>
> > Re: *Usefulness of FIM estimation, e.g. in NN optimization*
>
> As stated in the submitted paper, the FIM can be used (Line 117) "to study the
> singular structure of the neuromanifold [2,40], the curvature of the loss [8],
> to quantify model sensitivity [28], and to evaluate the quality of the local
> optimum [15,16]." Other applications include continual learning as pointed out
> by **Reviewer wDHQ**.
>
> We believe studying the estimation quality of the FIM in the general setting --
> without examining the specific application and implications of these use cases
> -- is meaningful on its own.
> First, the bias and variance always exists for tackling the computational
> cost of estimating the (diagonal) FIM, but has not been thoroughly analyzed.
> Second, such a study can be potentially applied to different scenarios.
> For example, in NN optimization, the bounds of the diagonal FIM leads to bounds
> of a learning step, which is determined by the product of the inverse of the
> diagonal FIM and the gradient vector. In our toy example in Figure 1, we
> demonstrated how optimization can be interfered with the variance of the FIM.
> To apply our results into specific optimizers would be another independent
> work.
>
> > Re: *FIM for analyzing NN structure*
>
> In the paper, we mentioned that one application of the FIM is to study the
> "intrinsic structure of the neuromanifold". Here, "structure" does not refers
> to the NN structure but the Riemannian geometry structure in the space of
> neural networks (aka the neuromanifold), where the FIM plays the role of a local
> metric tensor. Our main results are universal in the sense that they do not
> rely on specific NN structure. On the other hand, the FIM is indeed affected by
> the NN structure, which in the simplest case includes the depth and width of
> neural networks [15,16]. This is beyond the scope of the current paper.
>
> > Re: *Organization of technical contents*
>
> A main reason for the technical density is space limitation. We tried hard
> to layout the main results while providing enough examples with intuitions.
> To address the reviewer's concern, we propose to not further increase the
> density and use extra space, if provided, to remark on the connections between
> the equations -- to improve the paper's self-contained narrative (also
> suggested by **Reviewer qMTe**) -- and remark on takeaways in the conclusion
> (suggested by **Reviewer QMbw**).

---

> > ### Comment · Reviewer_7Xmq · 2024-08-12
> > **Response**
> >
> > Thanks for responding to my questions. Based on the reply, I have increased my score.

---

### Author Rebuttal · Authors · 2024-08-07

We extend our appreciation to all reviewers for their thoughtful reviews.
We are pleased to acknowledge the positive remarks on "[...] fair amount of technical
meat [...]" (**Reviewer 7Xmq**) and that the paper "[...] is very
extensive and mathematically sound" (**Reviewer QMbw**).
Despite the technical nature of the paper, we are glad to hear that the "[...]
alternation between formal statements and explanations is great and makes the
paper very pleasant to read" (**Reviewer qMTe**). We are also happy to learn about
other applications of the FIM, where it has been "[...] used in several papers of
continual learning [...]" (**Reviewer wDHQ**), which extends the potential
applicability of our results.

Please find the per-point rebuttals in each reviewer's response section.
Immediately below, we provide a shared response regarding our paper's relation to
reference [37]:

> [37] Soen and Sun. On the variance of the Fisher information for deep learning.
In Advances in Neural Information Processing Systems, 2021.


## Relationship to [37]
In response to **Reviewer wDHQ**, **Reviewer qMTe**, and **Reviewer QMbw**, we
hereby clarify our relationship with [37]. Indeed, both the current submission
and [37] study the same subject, and our developments rely on the closed-form
expression of the FIM variances, which is proved in [37]. Despite that, the
significance of the current work is explained as below.

- In [37], only the norm of the FIM and related variance tensors are bounded. By
  focusing on the diagonal elements, we derive novel bounds on the individual
  elements of these interested tensors, where the spectrum of sufficient
  statistics quantities naturally appear. In terms of proof techniques, [37]
  mostly utilize Holder's inequality whilst we utilize variational definitions /
  computations of eigenvalues.

- The main subjects in [37] are 4D tensors and their norms, which can not be
  easily computed. Their case studies are constrained to 1D exponential families.
  Our results leads to numerical algorithms that can be implemented through
  auto-differentiation. As a result, our bounds extend to typical learning
  settings (Section 5).

- We discussed not only variances but also bias in the estimation of the FIM,
  and clarified the relationship with empirical FIM (data FIM) that is widely
  used.

- [37] only considered the conditional FIM and associated objects in its study.
  Hence, their results only accounts for the sampling of $\\mathbf{y}$ with a
  fixed $\\mathbf{x}$.
  Our variance decomposition in Theorem 4.7 clarified how the variance is
  affected by both the sampling of $\\mathbf{x}$ and the sampling of
  $\\mathbf{y}$.

We are happy to include this discussion in Section 1 (around Line 42) and other relevant
places by utilizing additional space, if provided, to clarify the contributions
of our paper.

---

### Decision · Program_Chairs · 2024-09-25

**Decision:**

Accept (poster)

**Comment:**

All reviewers lean towards accept with at least one reviewer advocating for a strong accept. This is a highly theoretical paper that is likely to be of interest to others interested in theory.

However, there are several weaknesses discussed by the reviewers that should be addressed in the final version. In particular, more discussion on the differences between this work and [37] should be included. Also, the authors should work to better "remark on the connections between the equations" in the technical sections to make these sections more accessible and also further discuss when one estimator should be preferred over the other.